



# Nepal Ambient Monitoring and Source Testing Experiment (NAMaSTE): Emissions of trace gases and light-absorbing carbon from wood and dung cooking fires, garbage and crop residue burning, brick kilns, and other sources

Chelsea E. Stockwell[1], Ted J. Christian[1], J. Douglas Goetz[2], Thilina Jayarathne[3], Prakash V. Bhave[4], Puppala S. Praveen[4], Sagar Adhikari[5], Rashmi Maharjan[5], Peter F. DeCarlo[2], Elizabeth A. Stone[3], Eri Saikawa[6], Donald R. Blake[7], Isobel Simpson[7], Robert J. Yokelson[1], Arnico K. Panday[4]

[1]University of Montana, Department of Chemistry, Missoula, 59812, USA

[2]Drexel University, Departments of Chemistry and Civil, Architectural, and Environmental Engineering, Philadelphia, 19104,
USA

[3]University of Iowa, Department of Chemistry, Iowa City, 52242, USA

[4]International Centre for Integrated Mountain Development (ICIMOD), Khumaltar, 44700, Nepal

[5]MinErgy Pvt. Ltd, Lalitpur, 9354, Nepal

[6]Emory University, Department of Environmental Sciences, Atlanta, 30322, USA

[7]University of California-Irvine, Department of Chemistry, Irvine, 92697, USA

*Correspondence to*: R. J. Yokelson (bob.yokelson@umontana.edu)

**Abstract.** The Nepal Ambient Monitoring and Source Testing Experiment (NAMaSTE) campaign took place in and around the Kathmandu Valley and in the Indo-Gangetic plains (IGP) of southern Nepal during April 2015. The source characterization phase targeted numerous important but undersampled (and often inefficient) combustion sources that are widespread in the
developing world such as cooking with a variety of stoves and solid fuels, brick kilns, open burning of municipal solid waste (a.k.a. trash or garbage burning), crop-residue burning, generators, irrigation pumps, and motorcycles. NAMaSTE produced the first, or rare, measurements of aerosol optical properties and mass and detailed trace gas chemistry for the emissions from many of the sources. This paper reports the trace gas and aerosol measurements obtained by Fourier transform infrared (FTIR) spectroscopy, whole air sampling (WAS), and photoacoustic extinctiometers (PAX, 405 and 870 nm) based on field work with a
moveable lab sampling authentic sources. The primary aerosol optical properties reported include emission factors (EFs) for scattering and absorption coefficients (EF $B_{scat}$, EF $B_{abs}$, in $m^2$/kg fuel burned), single scattering albedo (SSA), and absorption Ångström exponents (AAE). From these data we estimate black and brown carbon (BC, BrC) emission factors (g/kg fuel burned). The trace gas measurements provide EFs (g/kg) for $CO_2$, CO, $CH_4$, selected nonmethane hydrocarbons up to $C_{10}$, a large suite of oxygenated organic compounds, $NH_3$, HCN, $NO_x$, $SO_2$, HCl, HF, etc.; up to ~ 80 gases in all.
The emissions varied significantly by source and light absorption by both BrC and BC was important for many sources. The AAE for dung-fuel cooking-fires (4.63 ± 0.68) was significantly higher than for wood-fuel cooking-fires (3.01 ± 0.10). Dung-fuel cooking-fires also emitted high levels of $NH_3$ (3.00 ± 1.33 g/kg), organic acids (7.66 ± 6.90 g/kg), and HCN (2.01 ± 1.25 g/kg), where the latter could contribute to satellite observations of high levels of HCN in the lower stratosphere above the Asian Monsoon. HCN was also emitted in significant quantities by several non-biomass burning sources. BTEX compounds (benzene,
toluene, ethylbenzene, xylenes) were major emissions from both dung- (~4.5 g/kg) and wood-fuel (~1.5 g/kg) cooking fires and a



simple method to estimate indoor exposure to the many measured important air toxics is described. Biogas emerged as the cleanest cooking technology of approximately a dozen stove/fuel combinations measured. Crop residue burning produced relatively high emissions of oxygenated organic compounds (~12 g/kg) and $SO_2$ (2.54 ± 1.09 g/kg). Two brick kilns co-firing different amounts of biomass with the primarily coal fuel produced contrasting results. A zig-zag kiln burning mostly coal at

high efficiency produced larger amounts of BC, HF, HCl, and $NO_x$, with the halogenated emissions likely coming from the clay. The clamp kiln (with relatively more biomass fuel) produced much greater quantities of most individual organic gases, about twice as much BrC, and significantly more known and likely organic aerosol precursors. Both kilns were significant $SO_2$ sources with their emission factors averaging (12.8 ± 0.2 g/kg). Mixed garbage burning produced significantly more BC (3.3 ± 3.88 g/kg) and BTEX (~4.5 g/kg) emissions than in previous measurements. For all fossil fuel sources, diesel burned more efficiently than

gasoline, but produced larger $NO_x$ and aerosol emission factors. Among the least efficient sources sampled were gasoline-fueled motorcycles during start-up and idling for which the CO EF was on the order of ~700 g/kg – or about ten times that of a typical biomass fire. Minor motorcycle servicing led to minimal if any reduction of gaseous pollutants, but reduced particulate emissions as detailed in a companion paper (Jayarathne et al., 2016). A small gasoline-powered generator and an "insect repellent fire" were also among the sources with the highest emission factors for pollutants.

**1 Introduction**

Several major atmospheric sources such as temperate forest biogenic emissions (e.g. Ortega et al., 2014), developed-world pollution from fossil fuel use (e.g. Ryerson et al., 2013), and laboratory-simulated biomass burning (BB) (e.g Stockwell et al., 2014) have been sampled extensively with a wide range of instrumentation; but many important emission sources remain unsampled, or rarely sampled, by reasonably comprehensive efforts (Akagi et al., 2011). As the emissions of greenhouse gases

and other air pollutants from developing countries have grown in importance for air quality and regional-global climate studies, the need for a more detailed understanding of these emissions has increased. For example, the diverse and loosely-regulated combustion sources of South Asia are poorly characterized and greatly undersampled relative to their proportion of global emissions (Akagi et al., 2011). These sources include industrial and domestic biofuel use (e.g. cooking fires), brick kilns, poorly-maintained vehicles, open burning of garbage and crop-residue, diesel and gasoline generators, and irrigation pumps.

Approximately 2.8 billion people worldwide burn solid fuels (e.g. wood, dung, charcoal, coal, etc.) for domestic (household) cooking and heating (Smith et al., 2013) with the largest share in Asia. Cooking fires are the largest source of soot in South Asia (Ramanathan and Carmichael, 2008). Industrial solid fuel use (e.g. brick kilns) is ubiquitous, but difficult to quantify in the developing world as it is not highly regulated or adequately-inventoried and can involve a variety of fuels (e.g. coal, sawdust, wood, garbage, tires, crop residue, etc.) (Christian et al., 2010). Along with industrial and domestic solid fuel use, open burning

of agricultural waste and garbage, gasoline and diesel-powered generators, and many examples of high-emitting vehicles are prevalent, but grossly undersampled in the developing world with previous field emissions characterization usually limited to a few trace gases and a few particulate species such as black carbon (BC) mass (Bertschi et al., 2003; Christian et al., 2010; Akagi et al., 2011; Bond et al., 2013).

Understanding the local through global impacts of these sources is vital to modeling atmospheric chemistry, climate, and,

notably, air quality as these sources most commonly occur indoors or near or within population centers. Aerosols directly affect climate through both absorption and scattering of solar radiation and indirectly effect climate by modifying clouds (Bond and Bergstrom, 2006). Therefore, global modeling of radiative forcing requires (among other things) accurate information on the amount and optical properties of aerosol emissions (Reid et al., 2005). BB is a major source of BC in the atmosphere, but it also dominates the global emissions of weakly-absorbing organic aerosol known as brown carbon (BrC). BrC has a contribution to





total absorption of BB aerosol that is poorly constrained, but critical to determining whether the net forcing of BB aerosol is positive or negative (Feng et al., 2003; Chen and Bond, 2010). Open burning of biomass and household-level consumption of biofuel account for a majority of BC emissions in important regions including Asia, but data are limited about how much BrC is emitted from biofuel and biomass combustion (Kirchstetter et al., 2004; Chen and Bond, 2010; Hecobian et al., 2010; Arola et

al., 2011). In general, there is significant uncertainty in emissions inventories since BrC is rarely tabulated as a separate species though the scattering and absorption of both BC and BrC are necessary to model radiative transfer (Clark et al., 1987). Additionally, the secondary formation of organic aerosol and ozone as well as the evolution of the BC and BrC optical properties are strongly influenced by the co-emitted gases and particles via processes such as coagulation, evaporation, oxidation, condensation, etc. (Alvarado et al., 2015; May et al., 2015). Near-source measurements of light absorption and scattering by BC

and BrC and their emission factors (EFs), along with the suite of co-emitted gas-phase precursors are needed to better estimate the impacts of these undersampled sources on climate, chemistry, and local-global air quality, especially in regions that lack comprehensive sampling.

Current reviews of global BC emissions note that global models likely underestimate BC absorption in several important regions including South Asia (Bond et al., 2013), making this an important region where undersampled emission sources have critical

climate and chemistry impacts. BC emissions from South Asia may negatively impact important regional water resources (Menon et al., 2010), contribute significantly to the warming of the Arctic (Allen et al., 2012; Sand et al., 2013), and emissions of volatile organic compounds (VOCs) and nitrogen oxides ($NO_x$) in this region were estimated to influence global warming more significantly than similar emissions from other Northern hemisphere regions (Collins et al., 2013). Thus, these sources contribute significantly to the local-global burden of primary aerosol, greenhouse gases, and reactive trace gases. Crudely

estimating their activity and the composition of their emissions can lead to significant errors and uncertainties in regional and global atmospheric models (Dickerson et al., 2002; Venkataraman et al., 2005; Adhikary et al., 2007, 2010; Akagi et al., 2011; Bond et al., 2013; Wiedinmyer et al., 2014).

The Nepal Air Monitoring and Source Testing Experiment (NAMaSTE) was a collaborative effort with multiple goals: (1) providing detailed chemistry, physical properties, and EFs for the trace gases and aerosols produced by many undersampled BB

sources, a poorly maintained transport sector, brick kilns, etc.; (2) using these new emissions data to expand and update emissions inventories including the Nepal national inventory; (3) supporting a source apportionment for Kathmandu, Nepal; (4) enhancing regional air quality and climate modeling; and (5) informing mitigation strategies. The project involved the International Centre for Integrated Mountain Development (ICIMOD, the in-country lead), MinErgy (a local contractor to ICIMOD), the Institute for Advanced Sustainability Studies (IASS, fixed site support), and the universities of Drexel, Emory,

Iowa (UI), California, Irvine (UCI), Montana (UM), and Virginia (UVA) in the U.S.

NAMaSTE employed two strategies simultaneously in the first measurement phase. A temporary supersite was set up at a representative suburban Kathmandu location (Bode) to augment the ongoing monitoring that was initiated there in 2012 (Chen et al., 2015; Lüthi et al., 2015; Putero et al., 2015; Sarkar et al., 2015) and to provide a target receptor for the source apportionment. Simultaneously, a well-equipped mobile team investigated numerous undersampled emissions sources in and around the

Kathmandu Valley and in the rural Terai region in the Indo-Gangetic plains (IGP) of southern Nepal. The sources represented authentic, common practices, but were usually not random and were arranged by the MinErgy and ICIMOD team before the campaign. The source and fixed site measurements commenced on April 11 of 2015 but were cut short by the Gorkha earthquake on April 25. The early termination prevented sampling of on-road mobile sources including heavy duty diesel trucks, which is now planned for phase two. Additional measurements of cooking fires and other sources planned in the Makwanpur District in

the foothills south of Kathmandu were also canceled, but many valuable data on similar sources had already been gathered. In





this paper we present a brief summary of the source sampling campaign and the details of the trace gas measurements of fresh emissions obtained by Fourier transform infrared (FTIR) spectroscopy and whole air sampling (WAS). We also present photoacoustic extinctiometer (PAX) data co-collected at 405 and 870 nm to measure the optical properties and estimate the mass of the fresh BC and BrC emissions. Substantial additional source characterization data based on sampling with Teflon and quartz

filters and a suite of other real-time aerosol instruments will be presented separately (Jayarathne et al., 2016; Goetz et al., 2016). Several weeks of high quality filter, WAS, aerosol mass spectrometer, and other real-time data from the supersite at Bode will also be presented/discussed separately. Taken together, the NAMaSTE efforts reduce the information gap for these important undersampled sources.

## 2 Experimental details

### 2.1 Source types and site descriptions

Nepal has variable terrain ranging from high mountains to the low elevation plains in the Terai. Our team was based out of the major population center of Kathmandu and we traveled by truck to various locations in and around the Kathmandu Valley while also traveling south to the Terai region. The Terai sits on the southern edge of Nepal in the IGP with intensive agriculture, terrain, and other similarities to the heavily populated region of northern India. The emissions data we present were obtained

from many sources including two-wheeled vehicles (motorcycles and scooters), diesel- and gasoline-powered generators, agricultural pumps, garbage fires, cooking fires, crop residue burning, and brick kilns. This section briefly summarizes the significance of each source and how they were sampled in our study.

### 2.1.1 Motorcycles and scooters

Mobile emissions are extremely important in urban areas as they contribute significantly to degradation of air quality on local to

20 regional scales (Molina and Molina, 2002, 2004; Molina et al., 2007; Dunmore et al., 2015). In the Kathmandu Valley, approximately 80% of registered vehicles are motorcycles or scooters and this is the fastest growing portion of the transport sector in Kathmandu and nationally (MOPIT, 2014). Motorcycles are generally larger with larger engines than scooters and in Nepal both now burn unleaded Euro-3 gasoline. Together, nationally, these two-wheeled vehicles consume about one-third of the gasoline and ~10% of total fuel used for on-road transport (WECS, 2014), with total sales of diesel and gasoline approaching

1Tg in 2015 (Nepal Oil Corporation Limited, 2015). Vehicle EFs are commonly obtained from bulk exhaust measurements (USEPA, 2015) and the International Vehicle Emissions (IVE) model specifically generates EF for mobile sources in the developing world (Shrestha et al., 2013). However, the detailed source chemistry (e.g. specific air toxics) is poorly known, especially for the developing world, as most studies focus only on CO, $NO_x$, $PM_{2.5}$, and a few hydrocarbons or total VOC in developed countries (e.g. Zhang et al., 1993; Pang et al., 2014).

There are a number of approaches to measure vehicular emissions that include in-use sampling while driving as well as more controlled dynamometer studies (Yanowitz et al., 1999; Pelkmans and Debal, 2006). Franco et al. (2013) outline the advantages and drawbacks to these various sampling techniques, though we will not discuss them further here. We were able to measure the emissions exhaust of five motorcycles and one scooter during start-up and idling, which are considered common traffic situations in the Kathmandu Valley. On 13 April 2015, we set up the NAMaSTE emissions measuring equipment next to a motorcycle

repair shop and to limit sampling bias, we deliberately tested every motorcycle/scooter that entered the shop for servicing that day. Each motorcycle and scooter was sampled (start-up and idling) pre- and post-servicing (one motorcycle was not sampled post-service). The motorcycle/scooter brand, model, etc. are shown in Table S1. The maintenance routine included an oil change, cleaning the air filters and spark plugs, and adjusting the carburetor.





### 2.1.2 Generators

Nepal has no significant fossil fuel resources and insufficient hydropower. As a result, load-shedding for many hours per day is common nationally and diesel or gasoline powered generators (a.k.a. gensets) are critical infrastructure for industrial, commercial, institutional, and household use, consuming about 57,000 Mg of fuel per year (World Bank, 2014). Based on fuel use, the emissions from generators could be about six percent of those from the transport sector. A large variety of generators are deployed to meet various size, power, and load capacity needs. In this study we sampled exhaust emissions from one small diesel generator with 5 kVa capacity (Chanqta, CED6500s) and a much larger diesel generator, located on the ICIMOD campus, with 100 kVa capacity running at 1518 RPM, 85% of full load. In addition to the two diesel generators, we sampled the exhaust emissions from one gasoline-fueled generator (Yeeda, Y-113(1133106)) that had a similar capacity (4 kVa) to the smaller diesel generator. Most pollutants from these engines are emitted through the exhaust, though some fraction likely escapes from fuel evaporation.

### 2.1.3 Agricultural water pumps

The use of diesel-powered agricultural pumps to extract groundwater for irrigation is rapidly rising in rural regions of Nepal and India with few to no operational regulations (Barker and Molle, 2004). The dependence on diesel operated pumps is likely to rise in South Asia as crop production rises with population demands. Although massive groundwater extraction has aided agricultural productivity in the region, the environmental impacts are seldom investigated (Shah et al., 2000). The pumps are estimated to consume ~1.3 Tg/yr of diesel fuel, over the entire IGP. Diesel-powered engine emissions can cause adverse health effects and unfavorably impact air quality, climate, crops, and soils (Lloyd and Cackette, 2001). We sampled the exhaust from two smaller diesel pumps (Kirloskar, 4.6 kVa and Field Marshall R170a, 5 kVa) in the Terai. We also sampled the exhaust opportunistically from a much larger irrigation pump (Shineray) in suburban Kathmandu. We were unable to confirm fuel type, but suspect it was gasoline based on the emissions chemistry.

### 2.1.4 Garbage burning

Open burning of garbage is poorly characterized even in the most "developed" countries where it occurs with minimal oversight mostly in rural areas (USEPA, 2006). In developing countries open burning of garbage is much more prevalent, poorly characterized, and much less regulated if at all. In Nepal, as throughout the developing world, open burning of garbage is ubiquitous at a range of scales. Small, meter-scale piles of burning trash are seen along roads and in uncultivated fields. Approximately 10-20 times larger areas of burning trash are also common at landfills, along roadsides and riverbanks, and basically many accessible, uncultivated open spaces; with these areas evidently serving as an informal public resource. Given the large amount of refuse generated and the lack of economically viable alternatives to burning (Pokhrel and Viraraghavan, 2005), garbage burning is estimated to consume about 644,000 Mg of municipal solid waste (MSW) annually in Nepal (Wiedinmyer et al., 2014) and have a major impact on air quality, health, and atmospheric chemistry. The few available previous measurements of garbage burning suggest it is particularly important as a source of BC, hydrogen chloride, particulate chloride, several ozone precursors, and air toxics such as dioxins (Costner, 2005; Christian et al., 2010; Li et al., 2012; Lei et al., 2013; Wiedinmyer et al., 2014; Stockwell et al., 2014, 2015). To our knowledge only one study reports reasonably comprehensive EFs for authentic open burning of garbage in the developing world, namely the landfill fire sampling in Mexico of Christian et al. (2010). Several lab studies have measured the emissions from garbage burning under controlled conditions in great chemical detail (Yokelson et al., 2013; Stockwell et al., 2014, 2015), but the relevance of these lab experiments needs further evaluation against a better





picture of real-world garbage burning. More real-world data are also needed to evaluate and update the garbage burning global inventory mentioned above (Wiedinmyer et al., 2014).

During NAMaSTE, we were able to contribute a modest but important expansion of the real world garbage-burning sampling. We sampled mixed-garbage burning on 6 occasions and we conducted 3 experiments burning segregated trash since some

processing of garbage before combustion is common. The segregated trash experiments isolated plastics and foil-lined bags in separate individual burns. The components in each garbage burn are summarized in Supplemental Table S2. The overall carbon fraction for mixed waste was calculated in Stockwell et al. (2014) by estimating the carbon content of each component in the mixture and the value for overall carbon content calculated there-in is assumed in our mixed garbage EF calculations (0.50). Polyethylene terephthalate (PET) is the most common plastic used in metallized packaging, such as the case for chip and other

foil-lined bags, and has a carbon fraction of 0.63 (USEPA, 2010). Most plastic bags are composed of high- and low-polyethylene (HDPE, LDPE) mixed with PET, and thus we estimate a carbon content of 0.745 in this study (USEPA, 2010).

### 2.1.5 Cooking stoves

Most global estimates of domestic biofuel consumption (~3000 Tg/yr) designate domestic biofuel burning as the second largest BB source behind savanna fires (Akagi et al., 2011). In the developing world, it is estimated that the majority of biomass fuel is

burned in Asia (~66%; Yevich and Logan, 2003). The solid fuels regularly burned include wood-derived fuels (e.g. hardwood, twigs, sawdust, charcoal) and agricultural residues (e.g. crop waste, livestock dung, etc.) though the fuel choice depends on availability, local customs, and the season. Yevich and Logan (2003) estimate residential wood fuel use for Nepal in 1985 as 9.8 Tg/yr. They do not estimate dung fuel use in Nepal, but the data they provide for Indian states with populations similar to Nepal suggests that about 1-2 Tg/yr of dung is combusted residentially in Nepal.

The cooking fire measurements in this study were conducted in two phases. First measurements were conducted by simulating field cooking in a laboratory to capture emissions from a wide range of stove- and fuel-types. Fuels for the lab tests included wood, dung, mixed wood and dung, biobriquettes, and biogas. Stove types included traditional single-pot mudstove, open 3-stone, bhuse chulo (insulated vertical combustion chamber), rocket stove, chimney stove, and forced draft stove. In the second phase, cooking emissions were sampled from authentic cooking fires in the kitchens of several rural Nepali homes and one

restaurant operated out of a personal kitchen. The two kitchens that utilized the traditional 1-pot clay stove were separated from the main dwelling by a mud wall. The ventilation for all cases was by passive draft through the door, open windows, and gaps between the walls and roof. Smoke samples were taken from the upper corner of the kitchen where the inflow and outflow of emissions were somewhat balanced and we were able to grab representative samples of accumulated emissions not needing weighting by the fire-driven flow. Several biofuels are available to the home and restaurant owners including twigs and larger

pieces of hardwood (*Shorea robusta* and *Melia azedarach* [Bakaino]) and dung shaped into logs or cakes sometimes containing minor amounts of straw. Different fuels or a combination of fuels were consumed depending on cooking preference. Our study was designed to bring more comprehensive trace gas and aerosol field sampling to the effort to understand cooking fires. We note that the women tending to the cookstoves were in and out of the kitchens with their children during food preparation so exposure is also a concern. While our concentration data could be used directly for indoor exposure estimates, a better approach

for estimating exposure to the air toxics we report is via our ratios to commonly measured species in the available studies more focused on representative exposure.





### 2.1.6 Crop Residue

Crop residue burning is ubiquitous during the dry season in the Kathmandu Valley and rural Nepal. Globally, burning crop residue post-harvest is widely practiced to enable faster crop rotation; reduce weeds, disease, and pests; and return nutrients to the soil. Alternatives to crop residue burning such as plowing residue into the soil or use as livestock feed have drawbacks including increased risk of wind-erosion of top soil and poor "feed" nutritional quality (Owen and Jayasuriya, 1989). Thus, open burning of crop residue is a prevalent activity in both developing and developed countries and it has important atmospheric impacts, but the emissions are not well characterized (Yevich and Logan, 2003; Streets et al., 2013; Sinha et al., 2014). Data for Indian states with similar population to Nepal suggest that total annual crop residue burning in Nepal is on the order of 6-7 Tg/yr (Yevich and Logan, 2003).

The land-use in southern Nepal is representative of the much larger Indo-Gangetic plains, which are inhabited by nearly a billion people. Crop residue types may impact emissions significantly, thus, mostly in the Terai, we characterized emissions from two regionally important crop types separately: rice straw and wheat. Additionally, we sampled the emissions from the burning of other crop types important in this region including mustard residue, grass, and a mixture of these residues. The carbon fractions assumed in this study were taken from previous analyses of similar fuels compiled in Table 1 of Stockwell et al. (2014).

### 2.1.7 Brick kilns

Brick production is an important industry in South Asia and the number of brick kilns in Nepal and India combined likely exceeds 100,000 (Maithel et al., 2012) with perhaps ~1000 kilns in Nepal that would likely require ~1-2 Tg of fuel per year. However, the industry is neither unambiguously inventoried nor strongly regulated. The previous trace gas and particulate emissions data available on brick kilns are very limited (Christian et al., 2010; Weyant et al., 2014). We were not able to sample a large number of kilns in Nepal, but we were able to greatly expand the number of important trace gas and aerosol species/properties quantified.

During NAMaSTE, we sampled two brick kilns just outside the Kathmandu Valley that employed different common and regionally important technologies. The first kiln sampled was a zig-zag kiln, which is considered relatively advanced due to an airflow system that efficiently transfers heat to multiple brick chambers. We note that many zig-zag kilns in the Kathmandu valley have a chimney upwards of ten meters high to minimize impacts on immediate neighbors. The tall stacks are most easily sampled from a port on the side, but this raises uncertainties due to possible condensation after sampling hot/moist exhaust or losses on stack walls past the sampling point. Therefore we elected to sample the zig-zag emissions from a kiln outside the valley with a shorter chimney and where our inlet could be within several meters of the chimney where emissions had cooled to near ambient temperature. This approach was followed to reliably sample the "real" emissions. The zig-zag chimney emissions were sampled for five hours (9 a.m.-2 p.m.), which captured several firing/feeding cycles lasting about one hour each. By cycles we refer to the periodic addition of a primarily coal/bagasse mix during the day through multiple feeding orifices (a.k.a. stoke holes) above the firing chamber that were moved as the firing progressed. We also occasionally diverted the sampling to capture the emissions from these stoke holes. The smoke emitted from both the chimney and stoke holes mostly appeared white with occasional puffs of brown smoke when coal was added through the stoke holes.

The second kiln was a common batch-type clamp kiln. In clamp kilns green unfired bricks are stacked and brick walls are built up to surround the unfired bricks. Each batch is stacked, fired, cooled and must be unloaded before firing the next batch. There is no chimney to vent emissions as the kiln ventilates freely through the sides and roof. The naturally escaping emissions were sampled at or near ambient temperature about a meter from the roof throughout the day. The clamp kiln smoke always appeared white with no apparent periods of black smoke.



Generally the cheapest type of coal available is used in south Asian kilns. Bricks are typically fired to 700-1100℃ and consume significant amounts of coal and biomass as detailed elsewhere (Maithel et al., 2012). The practice of biomass co-firing to reduce the use of coal is common as it reduces expense, but co-firing in general is also known to reduce fossil-$CO_2$ emissions and some criteria pollutants such as $NO_x$ and $SO_2$ (e.g. Al-Naiema et al., 2015). We expect that the emissions change depending on the

biomass to coal blending ratios in South Asia and that the blend likely varies considerably between kilns. In the two kilns we measured the primary fuel was coal, however, the clamp kiln was more substantially co-fired with biomass. The coal piles next to the clamp kiln were adjacent to large piles of cut hardwood, thus, the coal was likely co-fired with a substantial amount of hardwood and the emissions data confirms that. We note that we were not on site long enough to measure the emissions from the entire kiln lifetime. Thus, we cannot probe seasonal variation in brick kiln emissions. However, we did capture 4-5 entire firing

cycles from each kiln that should represent the emissions near the end of the dry season production period. Some kiln operators suspect that these emissions may reflect more efficient combustion (and more bricks per kg fuel) than when the kilns are first started up in January under conditions of lower ambient temperature.

## 2.2 Instrument details

### 2.2.1 Land-based Fourier transform infrared (LA-FTIR) spectrometer

A rugged, cart-based, mobile FTIR (MIDAC, Inc.) designed to access remote sampling locations (Christian et al., 2007) was used for trace gas measurements. The system can run on battery or generator power. The vibration-isolated optical bench consists of a MIDAC spectrometer with a Stirling cycle cooled mercury-cadmium-telluride (MCT) detector (Ricor, Inc.) interfaced with a closed multipass White cell (Infrared Analysis, Inc.) that is coated with a halocarbon wax (1500 Grade, Halocarbon Products Corp.) to minimize surface losses (Yokelson et al., 2003). In the grab sampling mode used for the FTIR trace gas data reported in

this paper, air samples are drawn into the cell by a downstream pump through several meters of 0.635 cm o.d. corrugated Teflon tubing. The air samples are then trapped in the closed cell by Teflon valves and held for several minutes for signal averaging to increase sensitivity. Once the IR spectra of a grab sample are logged on the system computer a new grab sample can be obtained. This facilitates collecting many grab samples. Cell temperature and pressure are also logged on the system computer (Minco TT176 RTD, MKS Baratron 722A). Spectra were collected at a resolution of 0.50 cm$^{-1}$ covering a frequency range of 600-4200

25    cm$^{-1}$. Since the last report of the use of this system (Akagi et al., 2013), several upgrades were made: (1) addition of a retroreflector to the White cell mirrors increased the optical pathlength from 11 m to 17.2 m, lowering previous instrument detection limits, (2) replacing the Teflon cell coating with halocarbon wax for better measurements of ammonia ($NH_3$), hydrogen chloride (HCl), hydrogen fluoride (HF), and other species prone to absorption on surfaces, (3) mounting the mirrors to a stable carriage rather than the previous method of gluing them to the cell walls, (4) the above mentioned Stirling cycle detector, which

gave the same performance as a liquid-nitrogen-cooled detector without the need for cryogens, (5) the addition of two logged flow meters (APEX, Inc.) and filter holders to enable the system to collect particulate matter on Teflon and quartz filters for subsequent laboratory analyses. The new lower detection limits vary by gas from less than 1 ppb to ~100 ppb and are more than sufficient for near-source ground-based sampling as concentrations are much higher (e.g. ppm range) than in lofted smoke (Burling et al., 2011). Gas-phase species including carbon dioxide ($CO_2$), carbon monoxide (CO), methane ($CH_4$), acetylene

($C_2H_2$), ethylene ($C_2H_4$), propylene ($C_3H_6$), formaldehyde (HCHO), formic acid (HCOOH), methanol ($CH_3OH$), acetic acid ($CH_3COOH$), furan ($C_4H_4O$), hydroxyacetone ($C_3H_6O_2$), phenol ($C_6H_5OH$), 1,3-butadiene ($C_4H_6$), nitric oxide (NO), nitrogen dioxide ($NO_2$), nitrous acid (HONO), $NH_3$, hydrogen cyanide (HCN), HCl, sulfur dioxide ($SO_2$), and HF were quantified by fitting selected regions of the mid-IR transmission spectra with a synthetic calibration non-linear least-squares method (Griffith, 1996; Yokelson et al., 2007). HF and HCl were the only gases observed to decay during the several minutes of sample storage in





the multipass cell. Thus for these species, the results are based on retrievals applied separately to the first ten seconds of data in the cell (Yokelson et al., 2003). An upper limit 1σ uncertainty for most mixing ratios is ±10%. Post-mission calibrations with NIST-traceable standards indicated that CO, $CO_2$, and $CH_4$ had an uncertainty between 1-2%, suggesting an upper limit on the field measurement uncertainties for CO, $CO_2$, and $CH_4$ of 3-5%. The $NO_x$ species have the highest interference from water lines

under the humid conditions in Nepal and the uncertainty for $NO_x$ species is ~25%.

In addition to the primary grab sample mode, the FTIR system was also used in a real-time mode to support filter sampling when grab samples were not being obtained. Side by side Teflon and quartz fiber filters preceded by cyclones to reject particles with an aerodynamic diameter > 2.5 microns were followed by logged flow meters. The flow meter output was then combined and directed to the multipass cell where IR spectra are recorded at ~1.1 second time resolution. In real-time/filter mode we did not

employ signal averaging of multiple scans and the signal to noise is lower at high time resolution. In addition, there could be sampling losses of sticky species such as $NH_3$ on the filters. However, the data quality is still excellent for $CO_2$, CO, and $CH_4$. This allowed the time-integrated mass of particle species to be compared to the simultaneously sampled time-integrated mass of CO and other gases and provided additional measurements of the emissions for these three gases as described in detail in the filter sampling companion paper (Jayarathne et al., 2016).

**2.2.2 Whole air sampling (WAS) in canisters**

Whole air samples were collected in evacuated 2 L stainless steel canisters equipped with a bellows valve that were pre-conditioned by pump-and-flush procedures (Simpson et al., 2006). The canisters were filled to ambient pressure directly in plumes (alternately from the FTIR cell for the zig-zag kiln) to enable subsequent measurement and analysis of a large number of gases at UCI (Simpson et al., 2006). Species quantified included $CO_2$, CO, $CH_4$ and 93 non-methane organic compounds

(NMOCs) by gas chromatography coupled with flame ionization detection, electron capture detection, and quadrupole mass spectrometer detection as discussed in greater detail by Simpson et al. (2011). Peaks of interest in the chromatograms were individually inspected and manually integrated. The limit of detection for most NMOCs that were sampled was 20 pptv, which was well below the observed levels. Typically ~60 WAS NMOCs were enhanced in the source plumes and we do not report the results for most multiply-halogenated species and the higher alkyl nitrates, which are mostly secondary photochemical products.

The species we do not report were not correlated with CO and are generally not emitted directly by combustion (Simpson et al., 2011). Styrene is known to decay in canisters and the styrene data may be lower limits. 96 WAS canisters were sent to Nepal to support the source characterization and ambient monitoring site. Because we anticipated needing canisters for a longer campaign, typically only one emissions sample and one background sample were collected for each source on each day. 48 WAS canisters were filled in all, mostly in April, along with FTIR and other instruments, but some additional source and background

measurements were conducted by WAS alone in June after the main campaign. The trace gas measurement techniques used for the reported EFs are indicated in the "method" row near the top of the supplemental and main tables.

**2.2.3 Photoacoustic extinctiometers (PAX) at 405 nm and 870 nm**

Particle absorption and scattering coefficients ($B_{abs}$, $B_{scat}$), single scattering albedo (SSA), and absorption Ångström exponent (AAE) at 405 nm and 870 nm were measured directly at 1 s time resolution using two photoacoustic extinctiometers (PAX,

Droplet Measurement Technologies, Inc., CO). This monitored the real-time absorption and scattering resulting from BC and (indirectly) BrC. The two units were mounted with AC/DC power options, a common inlet, desiccator (Silica Gel), and gas scrubber (Purafil) in rugged, shock-mounted, Pelican military-style hard cases. Air samples were drawn in through conductive tubing equipped with 1.0 μm size-cutoff cyclones (URG) at 1 L/min. The continuously sampled air is split between a





nephelometer and photoacoustic resonator enabling simultaneous measurements of scattering and absorption at high time resolution. Once drawn into the acoustic section, modulated laser radiation is passed through the aerosol stream and absorbed by particles in the sample of air. The energy of the absorbed radiation is transferred to the surrounding air as heat and the resulting pressure changes are detected by a sensitive microphone. Scattering coefficients at each wavelength were measured by a wide-
angle integrating reciprocal nephelometer, using photodiodes to detect the scattering of the laser light. The estimated uncertainty in absorption and scattering measurements is ~4-11% (Nakayama et al., 2015). Additional details on the PAX instrument can be found elsewhere (Arnott et al., 2006; Nakayama et al., 2015). Due to damage during shipping the PAXs were not available until repaired part-way thru the campaign and PAX data are therefore not available for a few sources.

       Calibrations of the two PAXs were performed frequently during the deployment using the manufacturer recommended scattering
and absorption calibration procedures utilizing ammonium sulfate particles and a kerosene lamp to generate pure scattering and strongly absorbing aerosols, respectively. The calibrations of scattering and absorption of light were directly compared to measured extinction by applying the Beer-Lambert Law to laser intensity attenuation in the optical cavity (Arnott et al., 2000). As a quality control measure, we frequently compared the measured total light extinction ($B_{abs} + B_{scat}$) to the independently measured laser attenuation. For nearly all the 1-s data checked, the agreement was within 10% with no statistically significant
bias; consistent with (though not proof of) the error estimates in Nakayama et al. (2015).

### 2.2.4 Other measurements

Two instruments provided $CO_2$ data that was used in the analysis of the PAX data. An ICIMOD Picarro (G2401) cavity ring-down spectrometer measured $CO_2$, CO, $CH_4$, and $H_2O$ in real-time. A Drexel LI-COR (LI-820) that was factory calibrated immediately before the campaign also measured $CO_2$ in real time. The sampling inlet of the Picarro and/or LI-COR was co-
located with the PAX inlets so that the time-integrated PAX particle data were easily ratioed to time-integrated $CO_2$ allowing straightforward, accurate synthesis of the PAX data with the mobile FTIR and WAS grab sample measurements as described below. A suite of other instruments (mini-aerosol mass spectrometer; seven wavelength, dual spot aethalometer (model AE33); etc. from Drexel) and the filters employed during the source sampling for subsequent analysis at UI will be described in more detail in companion papers (Jayarathne et al., 2016; Goetz et al., 2016).

### 2.3 Emission ratio and emission factor determination

The excess mixing ratios above the background level (denoted ΔX for each gas-phase species "X") were calculated for all gas-phase species. The molar emission ratio (ER) for each gaseous species X relative to CO or $CO_2$ was calculated for the FTIR and WAS species. For the single WAS sample of any source the ER was simply ΔX/ΔCO or $ΔCO_2$. The source-average ER for each FTIR species, typically measured in multiple grab samples, was estimated from the slope of the linear least-square line (with the
intercept forced to zero) when plotting ΔX versus ΔCO or $ΔCO_2$ for all samples of the source (Yokelson et al., 2009; Christian et al., 2010). Forcing the intercept effectively weights the points obtained at higher concentrations that reflect more emissions and have greater signal to noise. Alternate data reduction methods usually have little effect on the results as discussed elsewhere (Yokelson et al., 1999). For a handful of species measured by both FTIR and WAS it is possible to average the ERs from each instrument for a source together as in Yokelson et al., (2009). However, in this study, due to the large number of FTIR samples
(~5-30) and small number of WAS samples (typically one) of each source we simply used the FTIR ER for "overlap species" (primarily $CH_3OH$, $C_2H_4$, $C_2H_2$, and $CH_4$).



From the ERs, emission factors (EFs) were derived in units of grams of species X emitted per kilogram of dry biomass burned by the carbon mass balance method, which assumes all of the major carbon-containing emissions have been measured (Ward and Radke, 1993; Yokelson et al., 1996, 1999):

$$EF\left(X\right)\left(g\,kg^{-1}\right)= F_C \times 1000 \times \frac{MM_x}{AM_C} \times \frac{\dfrac{\Delta X}{\Delta CO}}{\sum_{j=1}^{n}\left( NC_j \times \dfrac{\Delta C_j}{\Delta CO}\right)} \qquad (1)$$

where $F_C$ is the measured carbon mass fraction of the fuel; $MM_x$ is the molar mass of species X; $AM_C$ is the atomic mass of carbon (12 g mol$^{-1}$); $NC_j$ is the number of carbon atoms in species j; $\Delta C_j$ or $\Delta X$ referenced to $\Delta CO$ are the source-average molar emission ratios for the respective species. The carbon fraction was either measured directly (ALS Analytics, Tucson, Table S3) or assumed based on measurements of similar fuel types (Stockwell et al., 2014). The denominator of the last term in Eq. (1) estimates total carbon. Based on many combustion sources measured in the past, the species $CO_2$, CO, and $CH_4$ usually comprise

97-99% of the total carbon emissions (Akagi et al., 2011; Stockwell et al., 2015). Our total carbon estimate includes all the gases measured by both FTIR and WAS in grab samples of a source and we include the carbon in elemental and organic carbon (ratioed to CO) measured during filter sampling. Ignoring the carbon emissions not measureable by our suite of instrumentation (typically higher molecular weight oxygenated organic gases) likely inflates the EF estimates by less than ~1-2% (Andreae and Merlet, 2001; Yokelson et al., 2013; Stockwell et al., 2015).

Biomass fire emissions vary naturally as the mix of combustion processes varies. The relative amount of smoldering and flaming combustion during a fire can be roughly estimated from the modified combustion efficiency (MCE). MCE is defined as the ratio $\Delta CO_2/(\Delta CO_2+\Delta CO)$ and is mathematically equivalent to $(1/(1+\Delta CO/\Delta CO_2))$ (Yokelson et al., 1996). Flaming and smoldering combustion often occur simultaneously during biomass fires, but a very high MCE (~0.99) designates nearly pure flaming (more complete oxidation) while a lower MCE (~0.75-0.84 for biomass fuels) designates pure smoldering. Source-averaged MCE was

computed for all sources using the source average $\Delta CO/\Delta CO_2$ ratio as above. In the context of biomass or other solid fuels, smoldering refers to a mix of solid-fuel pyrolysis and gasification (Yokelson et al., 1997) that does not occur in the liquid fuel sources we sampled (e.g. motorcycles, generators, pumps). However, given the large difference in the heat of formation for $CO_2$ and CO (283 kJ/mol) and CO being the most abundant carbon-containing emission from incomplete combustion, MCE and $\Delta CO/\Delta CO_2$ were useful qualitative probes of their general operating efficiency.

The time-integrated excess $B_{abs}$ and $B_{scat}$ from the PAXs were used to directly calculate the source average single scattering albedo (SSA, defined as $B_{scat}/( B_{scat} + B_{abs})$ at both 870 and 405 nm for each source). The PAX time-integrated excess $B_{abs}$ at 870 and 405 nm were used directly to calculate each source-average AAE.

$$AAE = -\frac{\log\left(\dfrac{B_{abs,1}}{B_{abs,2}}\right)}{\log\left(\dfrac{\lambda_1}{\lambda_2}\right)} \qquad (2)$$

Emissions factors for BC and BrC were calculated from the light absorption measurements made by PAXs at 870 and 405 nm

(described in section 2.2.3). Aerosol absorption is a key parameter in climate models, however, inferring absorption from total attenuation of light by particles trapped on a filter, or from the assumed optical properties of a mass measured by thermal/optical





processing, incandescence, etc. can sometimes suffer from artifacts (Subramanian et al., 2007). In the PAX, the 870 nm laser is absorbed in-situ by black carbon containing particles only without filter or filter-loading effects that can be difficult to correct. We directly measured aerosol absorption ($B_{abs}$, Mm$^{-1}$) and used the literature-recommended mass absorption coefficient (MAC) ($4.74 \pm 0.63$ m$^2$/g at 870 nm) to calculate the BC concentration ($\mu$g/m$^3$) (Bond and Bergstrom, 2006). To a good approximation,

5    sp2-hybridized carbon has an AAE of $1.0 \pm 0.2$ and absorbs light proportional to frequency. Thus, $B_{abs}$ due only to BC at 405 nm would be expected to equal $2.148 \times B_{abs}$ at 870 nm. This assumes any coating effects are similar at both wavelengths and has other assumptions considered reasonably valid, especially in biomass burning plumes by Lack and Langridge (2013). Following these authors, we assumed that excess absorption at 405 nm, above the projected amount, is associated with BrC absorption and the BrC ($\mu$g/m$^3$) concentration was calculated using a literature-recommended brown carbon MAC of $0.98 \pm 0.45$ m$^2$/g at 404

10    nm (Lack and Langridge, 2013). The BrC mass calculated this way is considered roughly equivalent to the total organic aerosol (OA) mass, which as a whole weakly absorbs UV light, and not the mass of the actual chromophores. The MAC of bulk OA varies substantially and the BrC mass we calculate with the single average MAC we used is only qualitatively similar to bulk OA mass for "average" aerosol and even less similar to bulk OA for non-average aerosol (Saleh et al., 2014). The BrC mass estimated by PAX in this way was independently sampled and worth reporting, but the filters and mini-AMS provide additional

samples of the mass of organic aerosol emissions that have lower per-sample uncertainty for mass. However, the optical properties from the PAX (SSA, AAE, and absorption EFs detailed below) are not impacted by MAC variability or filter artifacts. As mentioned above, the PAXs were run in series or parallel with a $CO_2$ monitor. The mass ratio of BC and BrC to the simultaneous co-located $CO_2$, measured by either the Picarro or LI-COR, was multiplied by the FTIR-WAS grab sample EF for $CO_2$ to determine mass EFs for BC and BrC in g/kg. From the measured ratios of $B_{abs}$ and $B_{scat}$ to $CO_2$, the EFs for scattering and

absorption at 870 and 405 nm (EF $B_{abs}$, EF $B_{scat}$) were calculated and reported in units of m$^2$ emitted per kg of dry fuel burned. The absorption and scattering EFs do not depend on assumptions about the AAE of BC or MAC values. Both the $CO_2$ and PAX sample were often diluted by using a Dekati Ltd. Axial Diluter (DAD-100), which was factory calibrated to deliver 15.87 SLPM of dilution air at an atmospheric pressure of 1004.6 mbar. Since both instruments samples were diluted by the same amount the dilution factor does not impact the calculation of PAX/$CO_2$ ratios. On the other hand, the dilution could have some impact on

gas-particle partitioning and the mass of BrC measured. More on the dilution system (and additional aerosol measurements) will be in a forthcoming companion paper (Goetz et al., 2016). Related measurements of elemental and organic carbon on the filters will be discussed by Jayarathne et al., (2016).

## 2.4 Emission factors for sources with mixed fuels

Several of the cooking fires burned a mix of wood and dung, mixed garbage was burned, and the brick kilns co-fired some

biomass with the dominant coal fuel. It is not possible to quantify the exact contribution of each fuel to the overall fuel consumption during a specific measurement period or even in total. Thus for the mixed-fuel cooking fires, we simply assumed an equal amount of wood and dung burned and used the average carbon fraction for the two fuels (0.40) (Stockwell et al., 2014; Table S3). For mixed garbage we used a rigorous laboratory carbon content determination (0.50, Stockwell et al., 2014) as opposed to a field determination that relied in part on visual estimates of the amount of components (0.40, Christian et al., 2010).

For the zig-zag kiln, we used the measured carbon content of the coal (0.722). For the clamp kiln, which likely had more co-fired biomass, we used a weighted carbon content assuming 10% biomass (at a generic 0.50 carbon content) and 90% coal (measured carbon content 0.660). The weighted average carbon content for the clamp kiln is about 2.5% lower than for the pure coal. The correction is speculative, but in the appropriate direction. The assumed carbon fractions are indicated in each table and the new fuel analyses performed for NAMaSTE for several fuel types are compiled in Table S3.





There are a few unavoidable additional uncertainties in assigning EFs to specific fuels for the brick kilns due to the possibility of emissions from the clay during firing. An estimate of the impact can be made from literature data. Clay typically contains well under one percent organic material and some can be lost during firing though residual C can increase the strength of the fired product and limited permeability makes complete combustion of the C in the clay difficult to achieve (Wattel-Koekkoek et al.,
2001; Organic Matter in Clay, 2015). For a generous exploratory estimate, we can assume the green bricks are 1% by mass organic matter that is all C. The brick/coal mass ratio reported by Weyant et al. (2014) is 6-26 and we take 15 as an average. 15 kg of clay at 1% C would have 150 g of C and one kg of coal at 70% C would have 700 g C. Thus, if all of the C in the clay was emitted it would cause about 18% of the total C emissions from the production process as an upper limit. The impact on the EF per kg coal-fuel that we calculated by the carbon mass balance (CMB) method depends on the species-specific ER to $CO_2$ in the
emissions from the clay C. If the ER for a species due to heating clay-C is the same as burning coal-C then there is no effect on the EF computed by the CMB per kg coal even though some of the species is actually coming from the clay. If the ER for "heating" clay-C is much higher or lower than the ER for burning coal-C (e.g. a factor ten), then for some non-$CO_2$ species, we would calculate increases or decreases in the CMB-calculated EFs relative to what actually is produced from the coal fuel. These are only large if a species is emitted mostly from clay combustion (vide infra).

## 3 Results and Discussion

### 3.1 Overview of aerosol optical properties

As mentioned above, we measured absorption and scattering coefficients as well as single scattering albedo directly at 405 and 870 nm. One wavelength-independent SSA value is often assumed for BB aerosol, but we find, as seen previously, that the SSA varies by wavelength for each source (Liu et al., 2014; McMeeking et al., 2014). The AAE is related to the shape of the absorption cross-section. The AAE for pure BC is assumed to be ~1 while higher values of AAE indicate relatively more UV
absorption and the presence of BrC. Figure 1 plots the source-average AAE versus the source-average SSA at 405 nm showing that high AAE is associated with high SSA. In Fig. 1 we show source-averaged AAEs ranging from ~1-5 and SSA values at 405 nm ranging from 0.37-0.95 for the sources tested in this study. The "high-AAE" sources appearing toward the upper left-hand corner (e.g. dung and open wood cooking, clamp kiln) are associated with significant light absorption that would be overlooked
by consideration of BC alone. We note that both PAXs were not operational during the generator and motorcycle sampling days and the PAX 870 was not operational during the irrigation pump sampling and for several garbage burns. We assumed that the pumps emitted only BC (this assumption is supported by the very low SSA) and used the MAC of BC at 405 nm (10.19 $m^2$/g) to calculate BC for this one source (Bond and Bergstrom, 2006). Both PAXs were operational for only one garbage burn, which had a low AAE near 1. Additional data from the aethalometer and filters, including for tests where one or both PAXs were not
operational, will be presented in companion papers (Jayarathne et al., 2016; Goetz et al., 2016).

It is important to consider the differences in optical properties for the aerosol emitted by the various biofuel/stove combinations used in this understudied region with high levels of biofuel use. Dung-fired cooking had a significantly higher AAE (4.63 ± 0.68) than cooking with hardwood (3.01 ± 0.10). The AAE is also generally lower for improved stove types (1.68 ± 0.47) when compared to traditional open cooking (i.e. without an insulated combustion chamber) (Fig. 1). In general, the optical properties
vary significantly by fuel type and the mix of combustion processes. As established in previous studies (e.g. Christian et al., 2003; Liu et al., 2014), BC is emitted by flaming combustion and BrC is emitted primarily during smoldering combustion and both can contribute strongly to the total overall absorption. Thus, the fuels that burned at a higher average MCE usually produced relatively more BC, which is also reflected in lower AAE and SSA values. These trends are similar to those observed during the





third and fourth Fire Lab at Missoula Experiment (FLAME-3, -4) (Lewis et al., 2008; McMeeking et al., 2014; Liu et al., 2014). Additional PAX results will be discussed by fuel type along with the trace gas results in the following sections.

### 3.2 Motorcycle emissions

The average EFs (g/kg) based on FTIR and WAS for the pre- and post-service fleet are shown in Table 1 and bike-specific
pre/post results are included in Supplemental Table S4. As a fleet, we found that after servicing MCE, $NO_x$, and most NMOCs were slightly reduced and CO slightly increased, however, these fleet-average changes are not statistically significant given the high variability in EF. Interestingly, for individual motorbike-specific comparisons (Table S4), in four out of five bikes, the MCE actually decreased after servicing indicating less efficient (though not necessarily less "clean") combustion, but this result is not statistically significant. To ensure that effects such as background drift did not cause this result we verified that the same results
occur when obtaining slopes from plots using absolute (i.e. not background corrected) mixing ratios. A similar lack of reduction in gas-phase pollutants has been reported in the literature following repair and maintenance (Chiang et al., 2008) and has been attributed to the complexity in adjusting carburetors to optimal combustion conditions (Escalambre, 1995). Our high CO emissions did not always correlate with high hydrocarbon emissions. While we do not know the exact cause of this, this effect has been seen in other vehicle studies with a variety of explanations (Beaton et al., 1992; Zhang et al 1995). While the gaseous
pollutants were not significantly reduced post-service, the fleet's total particulate emissions did decrease significantly and we refer to Jayarathne et al. (2016) for a detailed comparison.

CO had the highest emissions of any gas after $CO_2$ and the FTIR-measured average EFs pre- and post- service over 700 g/kg are about ten times the typical EF for CO observed in BB. The FTIR-measured average MCE for the post service motorcycles was ~0.60, equivalent to a $CO/CO_2$ molar ER of ~0.66, dramatically highlighting the poor efficiency of the engines. We were initially
surprised by this result, but it is confirmed by WAS in that the one WAS sample of start-up/idling emissions returned a $CO/CO_2$ ER (0.789) that is within the FTIR-sample range. In fact, even higher $CO/CO_2$ ERs (3.2 – 4.2) are generated for start-up of motorcycles in the IVE model, which is based on sampling in developing countries (Oanh et al., 2012; Shrestha et al., 2013). Of 11227 vehicles of all types tested by remote sensing during on-road use in Kathmandu in 1993, about 2000 had a $CO/CO_2$ ER higher than 0.66 (fleet average 0.39, range 0 - 3.8, Zhang et al., 1995).
The next most abundant emissions after CO were: $C_2$ hydrocarbons (~24 g/kg), "BTEX" (benzene, toluene, ethylbenzene, and xylenes) compounds (~15 g/kg), and then the sum of measured oxygenated volatile organic compounds (OVOCs) and $CH_4$ each at ~7 g/kg. The OVOC from this source were mostly phenol, hydroxyacetone, and acetone (Tables 1 and S4). The BTEX and acetone data are from the one motorcycle that was analyzed by WAS pre-service. The WAS provided several overlap species with the FTIR and many additional non-methane hydrocarbons (NMHCs) not measured by FTIR. First we note, in agreement
with the FTIR, ethylene and acetylene were the most abundant WAS NMHC species and they accounted for ~38% of the total WAS NMHC emissions. The acetylene to ethylene ratio in this sample was 0.45 which is similar to previous roadside studies of all traffic (Tsai et al., 2006; Ho et al., 2009). Significantly, the WAS sample showed high concentrations of BTEX compounds, some of which are important carcinogens and all of which can lead to significant secondary organic aerosol (SOA) production (Platt et al., 2014). Toluene is a common gasoline additive and is sometimes used as a tracer for gasoline evaporation (Tsai et al.,
2006). However, in our motorcycle data, aromatics account for ~31% of the NMHC in the exhaust emissions with toluene being the most abundant aromatic. Platt et al. (2014) measured BTEX emission factors from about 10-100 g/kg (a range for driving to idling) for two-stroke motor scooter exhaust; also finding that toluene was the most abundant aromatic and with the BTEX accounting for ~40% of VOC. The combustion process in motorcycle engines is generally less efficient than in automobile engines (Platt et al., 2014) and the incomplete combustion can lead to emissions of many NMHC components in the gasoline.





For instance, the exhaust emissions of branched $C_5$-$C_6$ alkanes, including 2-methylpentane and *i*-pentane (sometimes a tracer for gasoline evaporation (Morikawa et al., 1998; Guo et al., 2004)) were also significant in the motorcycle exhaust. Previous studies also found that the VOC emission profile from motorcycle exhaust was similar to gasoline headspace analysis (Liu et al., 2008). In summary inefficient motorcycle engines produce exhaust containing a suite of NMHCs that overlaps with those produced by

fuel evaporation. However, there may be significant variability in headspace and exhaust measurements as observed by Lyu et al. (2015).

The air toxin and common BB tracer HCN was emitted by the motorcycles at about a tenth the ER to CO typically measured for BB. However, because of the very high motorcycle CO emissions, the EF for HCN for motorcycles was actually similar to that for BB. This is of importance for health effects and the use of HCN as a BB tracer in urban areas of developing countries where

motorcycles are prevalent (Yokelson et al., 2007; Crounse et al., 2009). A few other emissions stood out in the dataset including high emissions of 1,3-butadiene (~1.3 g/kg). While 1,3-butadiene is not a component of gasoline, it is a known component of vehicle exhaust (e.g. Duffy and Nelson, 1996) and is believed to originate from the combustion of olefins (Perry and Gee, 1995). The EPA has highlighted 1,3-butadiene as having the highest cancer risk of air toxics emitted by U.S. motor vehicles (USEPA, 1993) and exposure in densely populated urban centers can have significant negative health impacts.

One scooter was sampled by FTIR during this campaign and the CO emissions of the smaller scooter engine were only one-fourth to one-half those of the motorcycles (Table S4). The scooter exhaust emissions were also significantly lower for most other species captured by FTIR. The scooter, however, was the only motorbike sampled that produced detectable formaldehyde, a known carcinogen, irritant, and important radical precursor in urban atmospheres (Vaughan et al., 1986; Volkamer et al., 2010).

It is important to note that the average EFs from this study are not intended to represent the entire Kathmandu fleet of vehicles (or even all motorcycle use) as there is significant emissions variability between vehicles depending on running conditions: road conditions, driving patterns, maintenance, emissions control technology (Holmén and Niemeier, 1998; Popp et al., 1999) and engine specifics: model, size, age, power, fuel composition, combustion temperature and pressure, etc. (Zachariadis et al., 2001; Zavala et al., 2006). Larger studies similar to Zhang et al. (1995) are needed to get fleet averages. However, motorcycles and

motor scooters have been identified as major contributors to transport sector pollution in Kathmandu (Shrestha et al., 2013) and elsewhere (Oanh et al., 2012; Platt et al., 2014) and we provide chemically detailed real-world EFs for motorcycles under some common operating conditions that were previously unmeasured in Kathmandu.

Because of the diversity in fleet characteristics and how operating conditions are subdivided it is difficult to compare to other studies, but some of the species we measured are explicitly provided in other vehicle emissions estimates (Oanh et al., 2012;

Shrestha et al., 2013; Platt et al., 2014). Probably the most direct comparison is with Oanh et al. (2012) who reported EFs (in g/km) specifically for motorcycles for both start-up and running for the Hanoi 2008 average fleet based on the IVE that included some overlap species with our study ($NO_x$, $CH_4$, acetaldehyde, formaldehyde, benzene, and 1, 3-butadiene). Except for 1,3-butadiene our average ratios to CO for these species for start-up and idling are only 3-26% of theirs for start-up or running. Zhang et al. (1995) noted that partially functional catalytic converters convert VOC to CO (rather than $CO_2$) lowering the

VOC/CO ratio and also that these devices were becoming more common in the overall Kathmandu fleet, which points to emission control technology as a source of variability. The motorcycles we tested were all four-stroke and built by some of the world's largest manufacturers in India where catalytic converters are required on two-stroke vehicles, but are not required for four-stroke bikes until 2015. The Indian motorcycle emissions standards are based on an idling test and become increasingly stringent every five years (factor of 14.25 reduction for CO from 1991 to 2010). In response, a variety of emission control

measures are incorporated in the motorcycle engines to reduce "engine out" emissions as opposed to "after treatment." Some of





these measures are described in detail by Iyer (2012) while others are proprietary. The durability of many of these measures is very low (Ntziachristos et al., 2006, 2009) meaning they deteriorate with age despite minor service. Fuel quality (adulteration) is also noted as a widespread issue for emissions control (Iyer, 2012). In summary, it is quite possible that our VOC/CO ratios are lower than Oanh et al. (2012) mostly because of increased prevalence of emissions control technology (although poorly

maintained) in Kathmandu in 2015 compared to Hanoi in 2008.

In general our emission ratios can be used with e.g. CO EFs from other studies to roughly estimate additional chemical details for operating conditions we did not sample. It is also interesting that we observed that the emitted gases did not change significantly after servicing. It is possible that gas-phase pollutants would have decreased post-service under "cruising" conditions, but we were limited to testing start-up and idling emissions. A study in Hong Kong found that replacing old catalytic

converters had a large impact on emissions, but minor servicing did not (Lyu et al., 2015). Thus, major servicing might be required to mitigate gas-phase pollutants in general. Finally, our filter results suggested that the particulate matter (PM) emissions were reduced post-service (Jayarathne et al., 2016). Therefore, it is likely that minor servicing of motorcycles is beneficial if it reduces the PM without making the vast majority of the gases significantly worse. The EFs (in g/kg) here could theoretically be converted to fuel based EFs (g/km) using a conversion factor based on motorcycle fuel economy. However, this

is a complex process in practice (Clairotte et al., 2012) and it would probably be more meaningful to combine our ER to CO with fuel-based CO emission factors measured under the appropriate conditions.

### 3.3 Generator emissions

Three generators (two diesel and one gasoline) were sampled about a meter downstream of the exhaust manifold and the EFs are shown in Table S5. The larger diesel generator located on the ICIMOD campus is professionally maintained and had a much

smaller EF CO (4.10 g/kg) and a higher MCE (0.998) than the smaller (rented) diesel generator (MCE 0.962, EF CO 76.1 g/kg). The smaller rented diesel generator had 18-150 times higher emissions for the five non-$CO_2$ gases measured from both sources. The one gasoline generator we sampled had much higher CO emissions (> 1000 g/kg) and was much less efficient (MCE 0.437) than both diesel generators. The gasoline-powered motorcycles discussed in Sect. 3.2 also had high EF CO (> 700 g/kg) with generally low MCEs.

Not surprisingly, the one diesel generator sampled by FTIR (the small rental) did emit high concentrations of $NO_x$ (~24 g/kg), while $NO_x$ emissions remained below the detection limit for the gasoline-powered generator sampled by FTIR (Vestreng et al., 2009). The gasoline-powered generator emitted more NMHCs than both diesel generators and likely produces high secondary aerosol that has been observed in gasoline vehicle emission studies (Platt et al., 2013). We measured gasoline generator BTEX emissions that were ~20 times greater than those from the large diesel generator and note that the SOA yields from

photooxidation of *m*-xylene, toluene, and benzene are significant (Ng et al., 2007). We were able to measure HCHO emissions by FTIR from the small diesel generator (2.75 g/kg) and the gasoline-generator (0.61 g/kg). Even though the diesel generator ran much cleaner overall (for gas-phase pollutants) it produced significantly more HCHO than the gasoline generator and we recall that HCHO was below the detection limit for gasoline-powered motorcycles we measured. This suggests diesel may tend to produce higher HCHO emissions than gasoline. As mentioned in Sect. 3.2, HCHO is an air toxin and is important in atmospheric

chemistry. Overall, OVOCs were not clearly associated with either fuel with the gasoline generator having higher EFs for acetaldehyde, acetone, phenol, and furan, but lower EFs for HCHO and organic acids.

Other evident differences between the generators were potentially based on fuel. The large well-maintained diesel generator emitted more of the heavier NMHCs including heptane, octane, nonane, decane, and methylcyclohexane than the lesser maintained gasoline generator. The gasoline generator had much higher EFs for the smaller-chain NMHCs ($C_2H_2$, $C_2H_4$, $C_2H_6$,





$C_3H_6$, etc.). While the diesel fuel generators we sampled burned cleaner overall in terms of gas-phase pollutants, diesel is normally considered a much dirtier fuel in terms of soot production. The two PAX instruments were not operational for sampling generators, but filters were collected and demonstrated a higher EF PM for the small diesel generator than the gasoline generator as will be highlighted by Jayarathne et al. (2016).

We were able to sample both the smaller diesel generator and the gasoline generator during both start-up and free-running conditions. The diesel generator produced concentrations about twice as high for most measured species during start-up as opposed to free-running conditions, while the gasoline-fueled generator did not show these start-up concentration spikes. Sharp emission spikes peaking during both cold- and hot-startups of diesel engines have been observed previously (Gullet et al., 2006). This is often attributed to periods of incomplete combustion during ignition, and could have significant impacts on air quality as

power-cuts are a frequent, intermittent occurrence throughout the valley.

In summary, the well-maintained diesel generator had much lower EFs for overlapping measured gases (except large alkanes, which were a minor overall component), but gasoline could have advantages in terms of $NO_x$ and PM emissions at the cost of increases in most other pollutants unless they could be reduced by better maintenance. Although vehicular emissions are most commonly reported, emissions from gasoline and diesel powered generators can also have large impacts in urban regions subject

to significant load-shedding, which is relevant throughout Nepal and especially in the Kathmandu Valley (World Bank, 2014).

### 3.4 Agricultural diesel pump emissions

In this study, two groundwater irrigation diesel pumps were sampled by FTIR and the EFs are reported in Table 2. In addition, a surface-water irrigation pump was sampled by WAS canisters only and showed massively higher CO emissions than the two other pumps in our study indicating it was probably gasoline-powered. The WAS data may be mainly of interest to characterize

old or poorly maintained pumps and the EFs are included in Supplemental Table S6.

For the two pumps sampled by FTIR, the grab samples during cold startup differed from the samples during regular continuous operation by a much larger degree than the variability in grab samples for the other sources so we computed EF by two methods. Method one is our standard approach based on the ER plot using all the samples. The startup emissions can be outliers in this approach and get lower weight accordingly. Thus, we also computed ERs from the sum of the individual ERs and used those to

generate a second set of EF that weights the startup emissions more. Our standard approach yields the EFs shown in Table 2, columns 2 and 4, with an average of those two columns in column 6. We have included columns 3 and 5 with EFs calculated from the sum of excess emissions that emphasizes startup more. The alternate EF calculation reflects the increased emission of hydrocarbon species during ignition. CO also increases substantially while $NO_x$ decreases slightly. We believe the most representative EFs for model input are taken from the standard approach that does not add weight to the start-up conditions, as

most pumps are likely operated over longer periods of time. However, all the data are provided should a user prefer a different approach.

Although the 870 nm PAX was not operational on this day, the EFs ($m^2/kg$) of $B_{abs}$ and $B_{scat}$ for aerosols measured at 405 nm and the SSA are reported in Table 2 for the complete sampling cycle. The SSA at 405 nm ($0.405 \pm 0.137$) indicates that the diesel pump emissions were dominated by strongly-absorbing aerosols and if we assume there are no BrC emissions from this source, a

reasonable assumption supported by the AE-33 data, the absorption at 405 nm can be used to get a rough estimate for EF BC. The average EF BC ($5.72 \pm 0.58$ g/kg) is very high compared to typical values closer to 1 g/kg for most sources.

From the average emissions in Table 2 we see that the two pumps sampled by FTIR were not as prolific emitters for most pollutants as many other sources sampled in this study. However, the emissions of $NO_x$ and absorbing aerosol were



comparatively high. Especially taken together, the emissions from diesel powered generators and agricultural water pumps are likely significant in both urban and rural regions of Kathmandu and should be included in updated emissions inventories.

### 3.5 Garbage burning emissions

For an overview of our Nepal garbage burning (GB) data that also allows us to compare to authentic field and lab measured GB,
we tabulated (Table S7) our study-average Nepal mixed GB EFs along with mixed GB EFs from two lab studies (Yokelson et al., 2013; Stockwell et al., 2015), field measurements of open GB in Mexican landfills (Christian et al., 2010), and a single airborne sample of a Mexican dump fire (Yokelson et al., 2011). Figure 2 displays the major emissions from these studies in order of their abundance in the NAMaSTE data. We observe an interesting mix of compounds usually associated with burning biomass (OVOCs) and fossil fuels (NMHC and BTEX) as well as nitrogen and chlorine compounds. Even though the
methodology and locales varied considerably, the EFs reported in each study show reasonable agreement for most overlap compounds (Fig. 2). The average EFs of smoldering compounds for mixed garbage burns in Nepal were generally slightly higher than the other studies and the average MCE was lower (0.923, range in MCE 0.864-0.980). This is consistent with observations by several co-authors that flaming dominated GB is more common in winter months in Nepal when GB also provides heat. The comparison also suggests that the lab results for compounds not measured in the field (e.g. Yokelson et al., 2013; Stockwell et
al., 2015) could be used if scaled with caution. The NAMaSTE-specific EFs for garbage burning are reported for each fire in Table 3 along with our study-average for mixed GB EFs and we discuss some emissions next.

The laboratory mixed garbage-burning experiments during FLAME-4 were the first to yield a glycolaldehyde EF (0.658 g/kg) for trash burning. Our 14 April fire burning "mostly plastics" in Nepal produced a very high glycolaldehyde EF (4.56 g/kg). In both cases, the actual glycolaldehyde source is probably paper products, since glycolaldehyde is a product of cellulose pyrolysis
(Richards et al., 1987). Glycolaldehyde in our first Nepal segregated plastics burn likely resulted from newspaper used as kindling for ignition. This burn also had high EFs for a few other OVOCs, especially formic and acetic acid and formaldehyde (5.30, 2.22, and 5.23 g/kg). The high EFs in this study indicate that garbage burning may be an important source of these aldehydes and acids. Co-firing paper with plastics is also the likely reason our 14 April "mostly plastics" simulation burned at a significantly lower MCE than the pure plastic shopping bags that were burned during the FLAME-4 campaign. Most garbage is a
more complex mixture than just paper and plastic so our average EFs for garbage burning in Nepal in Table 3 are based on only the results from sampling mixed garbage burns.

NMHCs were major emissions with ethylene and acetylene always important for both the mixed garbage and the mostly plastic burns. Interestingly, benzene (a carcinogen) was just below ethylene as the most abundant NMHC in mixed garbage burning emissions overall (Fig. 2). Estimates of waste burning by country for all countries are presented in Wiedinmyer et al. (2014). For
Nepal, the estimated amount of waste burned is 644 Gg per year. Based on our average benzene EF for garbage burning (2.61 ± 1.85 g/kg), we estimate that trash-burning in Nepal produces ~ 1.68 Gg benzene (range 0.490 – 2.87 Gg) annually. The central estimate of Wiedinmyer et al. (2014) is 0.580 Gg/yr of benzene emitted from Nepali garbage burning; at the lower end of our range, but only 34% of our mean.

As observed in Fig. 2, EF HCl varies significantly between experiments and within the same study. Yokelson et al. (2013)
reported a lab-measured EF HCl of 10.1 g/kg, whereas Stockwell et al. (2014) reported their highest lab-measured EF HCl at 1.52 g/kg. These values are close to the upper and lower end of EF HCl for authentic Mexican landfill fires (1.65-9.8 g/kg) (Christian et al., 2010). HCl fell below the detection limit in some FTIR grab samples collected during NAMaSTE, indicating that GB emissions can differ depending on which components are burning during a particular grab sample. Our 14 April burn with fuels that were mostly plastics had extremely high EF HCl (77.9 g/kg), suggesting that many of the bags burned were made





from polyvinyl chloride (PVC). Our average EF for HCl for mixed GB was 2.32 ± 1.01 well within the range for Mexican GB. The other major halogenated emission detected from mixed GB was chloromethane (by WAS) at an EF up to 1.59 g/kg (average 0.702 ± 0.648 g/kg).

HCN is considered useful as a biomass burning tracer (Li et al. 2000), but was emitted by the mixed garbage and mostly plastic burns with an EF HCN that is similar to BB. We did not collect data in Nepal for acetonitrile, which is also used as a BB tracer, but the high $CH_3CN$/HCN ratios in Stockwell et al. (2015) for laboratory garbage burning suggests a similar issue may occur. This should be factored into any source apportionment based on using these compounds as tracers in regions where the emission sources include BB and either or both of garbage burning and motorcycles (e.g. Sect. 3.2).

Carbonyl sulfide (OCS) is emitted by natural (oceans, volcanoes, etc.), BB, and anthropogenic (automobiles, fossil fuel combustion) sources (Kettle et al., 2002). Two of our mixed garbage burns had high EF OCS (> 0.1 g/kg) and these are the first measurements reporting an EF OCS for GB. Burns 1 and 5 (Table 3) both had high OCS and both had a higher percentage of food waste. Because OCS is relatively inert in the troposphere, it freely transports into the stratosphere where it photodissociates and oxidizes and can ultimately condense into particles. The other S-species we could measure remained low (DMS) or below detection ($SO_2$).

The global garbage burning inventory of Wiedinmyer et al. (2014) had to rely on the EF BC (actually a filter-based EC measurement) from just one study (0.65 g/kg, Christian et al., 2010). Both PAXs were operational during one mixed garbage burn and we measured an EF BC of 6.04 g/kg (with an AAE ~1) almost ten times larger than the previously measured EF for BC suggesting a strongly BC-dominated aerosol. In addition, we can estimate an upper limit for EF BC for some of the other trash fires by assuming all 405 nm absorption is due to BC while the 870 PAX was not operational. This provides our 405-estimated values in Table 3 and they range from ~0.561-1.69 g/kg. Thus, our EF of 6.04 g/kg is likely a high end value from a flaming dominated garbage fire (MCE 0.980) while our lower values come from fires with more smoldering (MCE ~0.96) that are probably more common. Overall our PAX data suggests an upward revision for the literature-average garbage burning EF BC to something above 1 g/kg. However, with only one robust PAX-based EF BC determination, we will rely on the detailed EC/OC particulate analysis from NAMaSTE to better characterize this source in Jayarathne et al. (2016).

### 3.6 Cooking fire emissions

There were two main goals of our cooking fire measurements. One was to increase the amount of chemically- and optically-detailed trace gas and aerosol information that has been quantified in the field to allow more comprehensive assessments of the atmospheric and health impacts. The second was to obtain this type of detailed information for cooking fires that represent the most common global practice (open hardwood-fuel cooking fires); a major undersampled regional cooking practice (dung-fueled cooking fires); and, in exploratory fashion, a diverse range of stove/fuel combinations being considered as mitigation strategies. First, we illustrate the range of cooking technologies that we sampled and support some basic observations by plotting the MCE of all the stove/fuel combinations that we tested in decreasing order in Fig. 3. Several things stand out. Firstly, the biogas, the bhuse chulo sawdust, and biobriquette-fueled stoves had the highest MCE in our (limited) testing out of the wide range of possibilities and generally had smaller gas-phase EFs. The two measurements for biogas varied substantially and the differences could be a gas leak through the supply line and/or lingering BB emissions present in the laboratory room, thus we favor the field values. Biogas has proven to be a viable alternative to traditional wood sources especially in rural Nepal where agriculture and animal husbandry are the main sources of income (Katuwal and Bohara, 2009), however, biogas stoves remain unaffordable for poorer households. The higher MCEs in our emissions survey study suggest more extensive testing of biogas or the bhuse chulo could be warranted. The complete individual emissions for all stoves/fuels measured during NAMaSTE are included in



Supplement Table S8. Another apparent feature of Fig. 3 is the sharp drop off in MCE for the tests on the right side of the figure, which were mostly field measurements as opposed to the generally higher MCE in lab measurements. This suggests that "lower" MCE near 0.92 for wood and 0.90 for dung are apparently representative of real world use. More field tests were planned but were not completed due to the earthquake. However, lower stove MCE in the field compared to lab testing has been reported

previously (Bertschi et al., 2003; Roden et al., 2008; Stockwell et al., 2014) and the literature average MCE for field use is close to 0.92 (Akagi et al., 2011). Thus, we are fairly confident in adjusting the lab data for open cooking to reflect lower efficiency to use the lab tests to augment the field data. The straightforward adjustment procedure is described next.

As in Stockwell et al. (2014, 2015) we obtained field representative values from the lab data by multiplying the lab ER-to-$CH_4$ (measured by FTIR or WAS) for smoldering compounds and the lab ER-to-$CO_2$ (measured by FTIR or WAS) for flaming

compounds by the field EF for $CH_4$ and $CO_2$, respectively. Our full original NAMaSTE data are in Table S8 and the adjusted laboratory data for gases for traditional open hardwood and dung cooking-fires were averaged together with our authentic field values to estimate our NAMaSTE-average EF for open wood and dung cooking-fires. Those estimates along with values from a few other studies that reported a reasonably large number of EFs for cooking fires burning wood and dung are shown in Table 4 and form the basis for much of the ensuing discussion.

We focus next on dung cooking-fires, which are prevalent in South Asia. To our knowledge, there are very few studies that report any EFs for dung burning (Akagi et al., 2011) and this work significantly expands the gas-phase emissions data. The NAMaSTE-derived dung cooking-fire average in Table 4 includes 4 traditional dung cooking-fires (1-pot mud stoves and 3-stone) and an open fire intended to represent an authentic open warming fire outside a rural home. The open warming fire had a lower MCE (0.876) than our two field dung cooking-fires (0.910 ± 0.003) that was slightly closer to the low MCE (0.839)

average value reported in Akagi et al. (2011) based on open pasture burning of dung in Brazil (Christian et al., 2007) and laboratory burns of Indian dung (Keene et al., 2006).

As shown for dung-fuel cooking-fires in Table 4, our EFs for $CH_4$ (6.65 ±0.46 g/kg) are lower than the literature average reported in Akagi et al. (2011) (11 g/kg), although both are within the range (3-18 g/kg) reported by Smith et al. (2000) for simulated rural cooking in India. OVOCs were major emissions and we provide the first EFs for many OVOCs (e.g.

formaldehyde, acetone, glycolaldehyde, acetaldehyde, etc.). Acetic acid and hydroxyacetone were the most abundant OVOCs, though the Nepal EFs (7.32 and 3.19 g/kg) are lower than the Brazil EFs (14.3 and 9.6 g/kg) reported by Christian et al. (2007) at a lower MCE. This work considerably expands our knowledge of NMHCs from this source and reports a much higher EF for $C_2H_4$ (4.23 g/kg) and also many previously unobserved NMHCs at high levels. In particular, our new NMHC data include high emissions for BTEX compounds, especially benzene and toluene (1.96, 1.26 g/kg). Other notable compounds with high

emissions that were previously unobserved include chloromethane (1.60 g/kg) and carbonyl sulfide (0.148 g/kg). This is consistent with the elevated Cl and S content in the dung sample from MT (0.19 % S, 0.05 % Cl; Table S3). Chloromethane is the main form of organic chlorine in the atmosphere (Lobert et al., 1999) and is discussed more below.

As expected, the high N-content of dung (1.9% Table S3) led to high emissions for N-containing gases including $NH_3$ (3 g/kg), $NO_x$ (~3 g/kg), and HCN (~2 g/kg). Our $NO_x$ EF is higher than previously reported and this is an EPA regulated criteria pollutant

that is an important precursor to ozone, acid rain, and nitrate aerosols. The high $NH_3$ (3.00 ±1.33 g/kg) and acetic acid (7.32 ± 6.59 g/kg) emissions we observed, also previously observed in Brazil dung-fire emissions, might lead to ammonium acetate in secondary aerosol. Laboratory measurements during FLAME-4 were the first to report HCN from wood cooking fires (Stockwell et al., 2014), though the ERs to CO were about 5 times lower than what is typically observed for other BB fuels. The NAMaSTE real-world wood cooking fires had higher HCN EFs (0.557 ± 0.247 g/kg) than in the lab (0.221 g/kg); however, our HCN to CO

ratio for dung burning is 3.5 times higher than for wood. Despite the lower ER for wood, its dominance as a fuel mean both





should be considered an important source of HCN in the atmosphere. The cooking source continues during the monsoon, when open burning is reduced, and likely contributes to the large HCN anomaly observed by satellite in the anticyclone over the Asian monsoon (Park et al., 2008; Randel et al., 2010; Glatthor et al., 2015). The NAMaSTE ΔHCN/ΔCO ratios should be considered when using HCN in any source apportionment of pollution sources in areas subject to biomass burning and dung cooking along

with the motorcycles and garbage burning mentioned above.

Yevich and Logan (2003) estimated annual Asian use of dung as a biofuel in 1985 at 123 (±50%) Tg, with India accounting for 93 Tg. The NAMaSTE field measurements of dung burning were conducted in the Terai region that makes up the southern part of Nepal and likely represents similar cooking conditions as those in northern India. Fernandes et al. (2007) estimated that only 75 Tg/yr of dung is burned globally while Yevich and Logan (2003) estimated a slightly higher global value (136 Tg). If we take

the average of these two studies as an estimate of dung biofuel use (106 Tg), then we estimate from our EFs that 0.78 Tg acetic acid, 0.21 Tg HCN, and 0.17 Tg $CH_3Cl$ are emitted from dung burning each year. This accounts for ~33, 51, and over 100% of the previously estimated total biofuel burning emissions for these species in the late 1990s (Andreae and Merlet, 2001). Our estimate of HCN emitted solely from burning dung accounts for ~4-8% of HCN thought to be emitted by total BB annually in earlier work (Li et al., 2000). Our estimate of $CH_3Cl$ emitted by dung burning alone is ~18% of the total global $CH_3Cl$ emitted by

BB in the inventory of Lobert et al. (1999). They also cited a high Cl content of dung (4360 mg/kg) and concluded BB was the largest source of $CH_3Cl$ in the atmosphere. The contribution of dung burning to acetic acid, HCN, $CH_3Cl$, and other species should be included in updated inventories of global BB and biofuel emissions.

We report the first BrC emissions data from dung burning (to our knowledge) in Table 4 based on our NAMaSTE field-measured values only. Our EF BrC of 5.54 ± 1.66 g/kg is qualitative, but substantial and our more rigorously measured AAE (4.63±0.68) is

higher than our NAMaSTE value for wood cooking (3.01 ± 0.10). Expressed in terms of light absorption, BrC accounted for ~93% of aerosol absorption at 405 nm for dung burning and 79% for wood burning. In addition, for dung burning the BC absorption EF at 870 nm was only 3.5% of the "BrC-only" absorption EF at 405 nm. Even for wood burning, the BC absorption EF at 870 nm was just 12% of the BrC absorption EF at 405 nm. From these values we see that dung cooking fires are an important BrC source in South Asia and that BrC from cooking fires in general is of great importance for understanding their

climate impacts. Our EF BC (0.04 g/kg) for dung is lower than the suggested EF reported in Venkataraman et al. (2005) (0.12g/kg) for lab-burned cattle dung, though it is within the low end of the range estimated by Xiao et al. (2015) (0.03-0.3 g/kg) for dung cooking-fires. The sum of our BC and BrC emissions (~5.5 g/kg) is significantly lower than total carbon (EC+OC, 22 g/kg) reported for lab measurements of dung cooking-fires in Keene et al. (2006), but the methods used are difficult to compare. Both studies highlight the need for more measurements of this source. The SSA for dung cooking-fires is statistically higher at

both wavelengths than for wood cooking, but both sources produced fresh smoke with SSA < 0.9 indicating it would (initially) warm the atmosphere and cool the surface, impacting climate (Praveen et al., 2012). Our values of EF $B_{abs}$, EF $B_{scat}$, AAE, and SSA at 405 and 870 nm shown in Table 4 for dung and wood burning are independent of MAC estimates and can be used in models directly to estimate the optical properties, forcing, etc.

Open cooking fires using hardwood fuel are the most common cooking technology globally. Our NAMaSTE measurements

significantly increase the number of gases that have been measured in hardwood open cooking-fire emissions in the field. We report a few new OVOCs with high EF such as acetone (0.524 g/kg) and many new EFs for NMHCs (Table 4). The NAMaSTE results include lower emissions of total BTEX compounds from wood cooking fires (~1.5 g/kg) than dung cooking fires (~5.5 g/kg) but confirm the high EF for these species previously reported in lab studies (~3.2 g/kg, Stockwell et al., 2015). DMS emissions have not been reported previously for open cooking, and the EF is relatively high (0.255 g/kg) for a BB source

(Simpson et al., 2011). Rather than walk the reader through all the data in Table 4 we reiterate the main result, which is that



models can now use much improved speciation of the trace gases emitted by cooking fires. This can be seen by comparing columns 2 and 4 (the literature average) in Table 4. The agreement is good for most species previously measured in the field. For example, the NAMaSTE-average MCE (0.923) is very close to the Akagi et al. (2011) field average MCE (0.927). In addition, NAMaSTE provides data in column 2 for about 70 gases not previously measured in field work to our knowledge. The data will
be used to update the tables in Akagi et al., (2011) creating a new literature average.

The numerous trace gas EFs we measured for open-hardwood cooking-fires in Nepal also present an important validation opportunity for cooking-fire trace gas measurements made on simulated cooking fires in a lab study that featured many advanced instruments mostly never deployed on field cooking-fires. In FLAME-4, the lab cooking-fire EF for trace gases were adjusted to the field average MCE (0.927) and reported in Table S3 of Stockwell et al. (2015). In Table 4 we show the overlap species
between NAMaSTE and FLAME-4. There are a few noticeable deviations between the lab and NAMaSTE EF for NMOC. The lab/field EF ratios are shown in parentheses for acetic acid (2.8), hydroxyacetone (0.38), BTEX (2), and HCN (0.40). However, comparing columns 2 and 5 shows agreement within one standard deviation of the mean for more than 70% of the ~26 overlap species. Fuel S and N content differences may explain the EF differences for $SO_2$ and $NO_x$. In general the agreement suggests the FLAME-4 trace gas EF are useful, especially for the > 100 species that study measured that were not measured in the field
(Stockwell et al., 2015; Hatch et al., 2015).

As noted earlier, aerosol emissions from wood cooking-fires are a major global issue. Our EF BC (0.221±0.127 g/kg) for hardwood cooking fires is significantly lower than the Akagi et al. (2011) literature average (0.833 ±0.025 g/kg) based on EC measurements, but was within the range reported in Christian et al. (2010) (0.205-0.674 g/kg). Our BC and BrC combine to ~9 g/kg which is ~40 % larger than the typical value for $PM_{2.5}$ from biofuel sources (~7 g/kg, Akagi et al., 2011). To our knowledge
we report the first field-measured EF $B_{abs}$ and EF $B_{scat}$ for wood cooking-fires at 405 and 870 nm (Table 4), which can be used in models without MAC assumptions. We also provide rare measurements of SSA and AAE for fresh cooking fire aerosol in Tables 4 and S8. Our AAE for hardwood cooking-fires (3.01) is higher than Praveen et al. (2012) measured in hardwood cooking-fire smoke (2.2) in the IGP in northern India. More work is required to examine how methodological differences, aging, and sample size vs real regional variability affect measurements of regional averages. Our hardwood cooking SSAs (0.794, 870 nm; 0.605,
405 nm) indicate an absorbing fresh aerosol, but SSA has been seen to increase rapidly with aging in BB plumes (Abel et al., 2003; Yokelson et al., 2009; Akagi et al., 2012). In summary, our PAX data from Nepal increases the total amount of sampling and approaches used to estimate regional average cooking-fire aerosol properties. Incorporating our data would nudge the regional average for hardwood cooking-fires towards higher BrC/BC ratios and we show that dung cooking-fires are also an important BrC source. Additional NAMaSTE aerosol data will be reported in companion papers (Jayarathne et al., 2016; Goetz
et al., 2016).

Health impacts of indoor cooking-fire emissions are a major global concern (Davidson et al., 1986; Fullerton et al., 2008, etc.). We did not target exposure assessment in NAMaSTE, but our data can be used in a piggy-back approach with studies focused on longer-term exposure to a key indoor air pollutant to estimate exposure to other air toxic gases not measured in those exposure studies following Akagi et al. (2014). We give one example. Based on our measurements it is possible to extrapolate
concentrations of trace species not measured in previous studies. For example, assuming similar emission profiles, we can scale indoor CO measured by Davidson et al. (1986) to estimate indoor benzene concentrations and exposures. In their study indoor concentrations of CO were 21 ppm, which would equate to 183 ppb benzene using the ER (benzene / CO) from our study for dung cooking. The same approach can be extended to any of the gases we measured for any of the stove and fuel types. Overall, we were able to survey a very large variety of cooking technologies, practices, and fuel options representative of a diverse region
and identify candidate technologies for further testing and possible wider use. The large amount of new gas and aerosol data



from NAMaSTE as a whole should improve model representation and help to better understand the local and regional climate, chemistry, and health impacts of domestic and industrial biofuel use.

**3.7 Crop residue fire emissions**

We present the first detailed measurements of trace gas chemistry and aerosol properties for burning authentic Nepali crop
residues and we also significantly expand the field emissions characterization for global agricultural residue fires. The EFs for each fire are compiled in Supplemental Table S9. We examine the representativeness of our trace gas grab sampling, justify a small adjustment to the trace gas data, and then discuss the implications of the trace gas and aerosol results.

A detailed suite of EF for several crop residues commonly burned in the U.S. and globally that is based on continuous lab measurements over the course of whole fires is reported in Stockwell et al. (2014, 2015). A few fuels they measured overlap with
our Nepal study, including wheat and rice straw. The average MCE (0.954) for our Nepal grab samples burning wheat varieties is very close to the lab measured wheat straw burning MCE (0.956), though other crop types do not compare as well. When we compare our Nepal-average MCE for all our crop residue fire grab samples (0.952) to earlier field measurements we find that the MCEs reported in Mexico (0.925) by Yokelson et al. (2011) and in the U.S. (0.930) by Liu et al. (2016) are significantly lower. In addition, the previous field studies obtained more grab samples of a larger number of fires and sampled from the air, which is
unlikely to return too low an MCE. The MCE that we obtained from the real-time FTIR CO and $CO_2$ measurements that supported filter collection was also lower (e.g. ~0.933) and closer to the above-mentioned field MCE values. Thus, we believe our Nepal-average MCE based on grab samples is likely biased upwards. Thus, to make our Nepal EFs more representative of the likely Nepal (and regional) average, we have adjusted to the average airborne-measured field MCE (0.925) observed for crop residue burning in another developing country (Mexico) according to procedures originally established in Stockwell et al. (2014)
and also described in Sect. 3.6 above. These adjusted EFs for selected compounds are included in Table 5 along with values from selected other previous studies. Additional compounds measured in this study (both original and adjusted) are included in Supplemental Table S9.

Figure 4 shows the top OVOCs, NMHCs, and S- or N-containing compounds emitted and shows good agreement with literature values for overlap species. As noted in Stockwell et al. (2014), glycolaldehyde (the simplest "sugar-like" molecule) is a major
emission from crop residue fires and Fig. 4 shows that glycolaldehyde is the dominant NMOC by mass from the NAMaSTE crop residue fires. When we compare to other fuel-types, the EFs of glycolaldehyde from our study, smoldering Indonesian rice straw (Christian et al., 2003), and an assortment of U.S. crop residue fuels (Stockwell et al., 2014) are significantly higher than from other BB sources (Burling et al., 2011; Johnson et al., 2013; Akagi et al., 2013). Glycolaldehyde was below the detection limit for one NAMaSTE crop-type (mustard residue), suggesting emissions variability by fuel-type and/or fuel-properties. Our average
glycolaldehyde EF (4.07 g/kg ± 4.03) is similar to typical EFs for total PM from BB and glycolaldehyde has also been shown to be an efficient aqueous phase SOA precursor (Ortiz-Montalvo et al., 2012). Other oxygenated species emitted in large amounts by the crop residues burned in NAMaSTE include butanone (methyl ethyl ketone) (1.93 ± 2.41 g/kg) and hydroxyacetone (1.48 ± 0.62 g/kg). The Nepal data are higher or similar to previous data for many OVOC, but noticeably lower for methanol, formaldehyde, and organic acids. As expected the emissions of OVOCs were greater than NMHCs, though there are also large
emissions of $C_2$ NMHCs and BTEX compounds.

Figure 4 shows several major S- and N-containing compounds including significant $SO_2$ emissions (2.54 g/kg). While the $SO_2$ emissions are large compared to most BB types, the emissions from other S-containing compounds (OCS, DMS) are limited. $SO_2$ is an important precursor of sulfate aerosols and was also a significant emission from grasses and crop residue in Stockwell et al. (2014). This update is important to include in emissions inventories as many global and regional estimates rely on the much





smaller value (0.4 g/kg) reported by Andreae and Merlet (2001) (Streets et al., 2003). Yokelson et al. (2011) noted high emissions of $NO_x$ from crop residue fires sampled near the beginning of the Mexican dry season when plant N content may be higher. Our Nepal $NO_x$ (~2.5 g/kg) emissions for this fire type were measured in April, 6 months after the dry season started in October and may reflect lower fuel N content. The higher $NO_x$ emissions (4.65 g/kg) in Mexico may have also reflected higher

wind speed as an important mechanism, but one that requires airborne sampling to probe.

Unlike U.S. crop residue fires (Stockwell et al., 2014), HCl remained below the detection limit in nearly every crop residue burn. As a landlocked country these crops are not as influenced by chlorine-rich maritime air. Additionally, in comparison to U.S. crops, most rural agriculture in Nepal may be less augmented by chemical pesticides. There are, however, detectable emissions of $CH_3Cl$, which have not been measured previously in the field for crop residue burning. This new information for $CH_3Cl$

should be considered when assessing global emissions of reactive chlorine (Lobert et al., 1999).

The absorption and scattering coefficients at 405 and 870 nm were measured for 5 of the 6 crop residue fires. The fire-average SSA at 870 nm and AAE for these crop residue fires span a wide range. SSA (870) ranges from 0.579-0.981 (average 0.82 for both 870 and 405 nm) and AAE ranges from ~1.58-3.53 (average near 2). The AAE as a function of SSA colored by MCE is shown in Fig. 5 for all the real-time 1 s data collected during crop residue fires. The AAE increases sharply at high SSA, while

the MCE distinctly decreases at increasing SSA. These observations support previous interpretations that BrC is produced primarily by smoldering combustion at lower MCEs for most BB fuel-types (Liu et al., 2014; McMeeking et al., 2014). Similar trends were observed for all other fuel-types except for the zig-zag brick kiln, which will be discussed in the next section. The BC and OC literature average for crop residue fires reported by Akagi et al. (2011) were based on only two fires. Our average EF BC (0.831 ± 0.497 g/kg) from 5 crop residue fires is similar to the literature value (0.75 g/kg), while we report the EF for BrC for

the first time (10.9 ± 6.5 g/kg), which is considerably larger than the global average OC reported in Akagi et al., (2011), but in good agreement with the NAMaSTE, simultaneously-measured filter organic mass (~10 g/kg) (Jayarathne et al., 2016). More importantly from an absorption standpoint, we report EFs for $B_{abs}$ and $B_{scat}$ at both wavelengths for this fuel-type in column 4 of Table 5.

### 3.8 Brick kiln emissions

Very little is known about the chemical composition of brick kiln emissions. There are very few studies and most of what is reported focuses on a few key pollutants including CO, PM, and BC (Weyant et al., 2014). A previous study measured a larger suite of emissions from authentic brick kilns in Mexico (Christian et al., 2010), however, the fuel burned in those kilns was primarily biomass and the NMOC emissions were somewhat comparable to those from biomass burns. Coal is the main fuel used in brick kilns globally and to our knowledge NAMaSTE produced the first quantitative emissions data for numerous

atmospherically-significant species from authentic coal-fired brick kilns in a region heavily influenced by this source. The individual EFs for both brick kilns sampled in this study are reported in Table 6. There are large differences between the two kilns types that stand out in Table 6 despite our lack of opportunity to measure inherent kiln variability. We will first discuss the kiln emissions individually and then follow with a detailed kiln comparison.

### 3.8.1 Zig-zag emissions

The zig-zag kiln emissions had a very high average MCE (0.994) and the EFs for most smoldering compounds (e.g. most NMOC) were much reduced. Not surprisingly, the EFs for flaming compounds including HCl, HF, $NO_x$, and $SO_2$ were high. High emissions of $NO_x$ and S-containing gases are important as ozone and aerosol precursors and because they can enhance deposition and $O_3$ impacts on nearby crops and negatively impact crop yield. The latter issue is especially relevant since brick



kilns are commonly seasonal and located on land leased from farmers, where the depletion of the soil to collect clay for bricks is already another agricultural productivity issue.

The zig-zag kiln was the only source in our NAMaSTE study that emitted detectable quantities of HF. It has been suspected that brick kilns are an important source of atmospheric fluorides since fluorine is typically present in raw brick materials (USEPA, 1997). We found HF was a major emission from the zig-zag brick kiln with an average EF of 0.629 g/kg and a peak concentration of ~13 ppm. HF is a phytotoxic air pollutant and agricultural areas with visible foliar damage in Pakistan were suspected to be impacted by HF emissions from nearby brick kilns (Ahmad et al., 2012). While HF is rapidly transformed to particulate fluoride, much previous work confirms adverse effects of HF or particulate fluoride from various sources on crops (Haidouti, 1993; Ahmad et al., 2012). Since many brick kilns are present in agricultural regions, this first confirmation of high HF emissions is an important finding and should also be included in assessments of kiln impacts on agriculture. HF emissions from brick kilns likely vary considerably depending on the F-content of the clay (and possibly the coal) being fired (as discussed further below). HF is also very reactive, but perhaps particle fluoride could serve as a regional indicator for brick kilns with more work.

Because of the large number of FTIR grab samples over the sampling day, which lasted approximately 5 hours, we can construct a rough time series of the kiln emissions with resolution averaging about 12 minutes. To emphasize chemistry, normalize for fuel consumption rates, and account for somewhat arbitrary grab sample dilution, in Fig. S1 we plot selected ERs to $CO_2$. The ERs of HCl and HF to $CO_2$ rise first and track together over time. The ERs of NO and $SO_2$ rise next and their observed peak is about 2 hours after the halogens. This is consistent with the halogens being driven from clay at 500-600 °C (USEPA, 1997). The halogen peaks are then followed by a peak in the $NO_x$ and $SO_2$ emissions likely from the coal fuel.

As noted in Sect. 2.1.7, in hopes of obtaining representative emissions from this particular brick kiln, we sampled the smoke coming out of the top of the chimney stack, but we also sampled the lesser amount of emissions escaping the coal-feeder stoke holes located on the "roof" of the kiln. Table 6 also includes the EFs specific to the emissions from the stoke holes. The MCE is significantly reduced (0.861), consequently the EFs of smoldering compounds are much higher with e.g. high EF CO (230 g/kg). Oddly, the stoke hole smoke also had higher EFs for HF, HCl, $NO_x$, and $SO_2$; compounds normally emitted during flaming combustion. This is probably because the stoke holes are much closer to the combustion zone and many internally generated species are scavenged in the kiln and stack walls before being emitted from the stack. Some kilns have internal water reservoirs below the stack to scavenge the smoke as rudimentary emissions control. However, these stoke hole emissions do not need to be weighted much if at all in an assessment of overall emissions as the vents are normally closed.

Table 6 includes the EFs for BC and BrC, and the EFs for scattering and absorption at 405 and 870 nm calculated from all the real-time PAX (and co-located $CO_2$) data above background for separate plumes throughout the sampling day, that we then averaged together. The SSA at 870 nm (0.779 ± 0.103) indicates that BC contributes to the absorption in the fresh emissions while the AAE (1.92 ± 0.50) implies that the emissions are not pure BC. The PAX data suggest that a little under half the absorption at 405 nm is due to BrC. Weyant et al. (2014) reported a range of EFs for EC for South Asian brick kilns (0.01-3.7 g/kg) and our EF BC (0.112 g/kg) falls within the range they report. We note that for all the other sources sampled in NAMaSTE and in the BB literature, high values of SSA and AAE are mostly associated with a low MCE (smoldering) and low SSA/AAE is associated with high MCE (flaming). This is illustrated for crop residues in Fig. 5. For the zig-zag kiln this pattern is less pronounced. In the zig-zag kiln, the highest MCE values are not clustered at the lowest SSA/AAE (Fig. 6). Nearly all the real time data from the zig-zag kilns was at high MCE (>0.95), but accompanied by some evidence for BrC emissions. Given the plethora of possible UV-absorbing compounds in OA, characterizing the variety of primary and secondary "BrC types" with different absorption intensities, abundances, and lifetimes is an important area for future research (Saleh et al., 2014).




### 3.8.2 Clamp kiln emissions

The clamp kiln emissions had a lower average MCE (0.950) than the zig-zag kiln (though still reflecting primarily efficient combustion), which is not surprising since we estimate the fuel had a larger component of biomass. Consequently the EFs for most products of incomplete combustion are ~5-3000 times higher than those from the zig-zag kiln and also higher than values

reported for a clamp kiln in Mexico that burned mostly sawdust at an average MCE of 0.968 (Christian et al., 2010). Even though the MCE was lower, the clamp kiln EF $SO_2$ (13.0 g/kg) was almost the same as the zig-zag kiln. This is most likely rationalized at least in part by the higher sulfate emission factors for the zig-zag kiln (Jayarathne et al., 2016). For all grab samples of the clamp kiln, the NO remained below the detection limit while $NO_2$ only had detectable quantities for three grab samples near the end of the day. HCl and HF probably remained below the detection limit because of lower halogen content in

the clay (vide infra and Table S3).

If we convert and sum the $NO_2$, NO, and HONO emissions to "$NO_x$ as NO" this quantity is more than 3.5 times higher from the zig-zag kiln. The coal from both kilns had similar N content so the difference in $NO_x$ emissions is most likely traced to the higher MCE in the zig-zag kiln. However, we cannot completely rule out a different contribution of "thermal $NO_x$" between the kilns. Co-firing coal with biomass is a common practice in power plants as it has been shown to decrease combustion zone temperature

and thermally-dependent $NO_x$ formation, thereby reducing several criteria pollutants including $NO_x$ (USEPA, 2007; Al-Naiema et al., 2015). Thus, the lower $NO_x$ EFs from the clamp kiln could be partly due to co-firing with more biomass.

The differences in NMOC emissions for the two kiln types were dramatic. We simply list some common pollutants/precursors of concern and include the approximate $EF_{CK}/EF_{ZZ}$ ratio in parentheses after each: CO (7), $CH_4$ (223), ethane (2604), ethylene (30), benzene (203), methanol (16), phenol (28). In addition, many species were emitted at high levels from the clamp kiln but were

below the detection limit from the zig-zag kiln including: formaldehyde, furan, hydroxyacetone, and ammonia. The main emissions overall from the clamp kiln in order of mass were: $CO_2$, CO, $CH_4$, $SO_2$, ethane, propane, hydroxyacetone, BrC, methanol, and benzene. Methane is an important short-lived climate pollutant and the $CH_4$ EF for the clamp kiln (19.5 g/kg) is among the highest seen for any combustion source. The other alkanes were also extremely enhanced all the way through $n$-decane which had an EF of 0.428 g/kg. These enormous EFs for alkanes are not typical for BB and might reflect burning coal

inefficiently. Another possible explanation is that used motor oil is reportedly sometimes disposed of as fuel in brick kilns or added to the fuel to impart color to bricks (USEPA, 1997; Christian et al., 2010). The enhancement observed for the alkanes throughout the $C_1$-$C_{10}$ size range that we could measure suggests that even larger alkanes are also enhanced. Large alkanes have recently attracted attention as important SOA precursors (Presto et al., 2010). In our clamp kiln data the sum of the EFs for NMOCs we measured that are known to have high yields for SOA (BTEX plus phenol) is ~5 g/kg, which is already much larger

than the initial EF OA as crudely approximated from the EF BrC (~2.0 ± 0.4 g/kg).

The EF BC (0.02 g/kg) for the clamp kiln was much lower than for the zig-zag kiln and the co-collected filter data are consistent with this result. Weyant et al. (2014) also noted similar "low" EFs for EC for several brick kilns measured in that study. The EF BrC was greater for the clamp kiln than the zig-zag kiln, which is consistent with the filter OC and an expected result given a more significant biomass contribution to overall fuel. The AAE and SSA were slightly greater for the clamp kiln than the zig-zag

kiln (Table 6).

We had only one sample of the coal from each kiln and the elemental analysis is shown in Table S3. The likely higher fuel variability for the non-C trace substances limits us to a few general comments. The measured emissions of the sulfur species from both kilns (including stoke holes) accounted for about 60-111% of the nominal S in the coal, which is a good match given experimental uncertainty. The measured emissions of N-containing species from both kilns were significantly lower than the

nominal coal N. Much of the missing N was likely emitted as $N_2$, especially at high MCE (Kuhlbusch et al., 1991; Burling et al.,



2010). Finally, the zig-zag kiln emissions had significantly higher halogen content than the 0.3 g/kg upper limit for the zig-zag coal. This is consistent with our speculation above that much of the halogen emissions come from the clay and that this is a source of kiln to kiln variability.

This is by no means an exhaustive evaluation of South Asian brick kiln emissions. However, because there are so few studies detailing the chemical composition of brick kiln emissions, this is a valuable addition to the current body of measurements. In terms of comparative pollution between the two technologies, there are some trade-offs. The clamp kiln we sampled produced far more of BrC and a large suite of NMOC pollutants and precursors typically associated with inefficient combustion of biomass (e.g. HCHO and benzene) or (likely) inefficient combustion of motor oil or coal (e.g. alkanes). The zig-zag kiln we sampled produced significantly more BC, $NO_x$, HCl, and HF; where the latter two could be larger because of the clay and not the kiln design. For $SO_2$ the kilns were not significantly different. Ultimately, since the zig-zag kiln is thought to produce significantly more bricks per unit fuel use than the clamp kiln (e.g. Weyant et al., 2014), this ratio should be further investigated for scaling emissions (on a per brick basis). The zig-zag kiln is very likely preferred from the standpoint of pollutants emitted per brick produced, which is a major factor in selecting mitigation strategies. More measurement and modeling studies will clearly be needed to fully assess the impact of brick kiln emissions and subsequent atmospheric chemistry in the region.

## 4 Conclusions

We investigated the trace gas and aerosol emissions from a large suite of major undersampled sources around Kathmandu and the Indo-Gangetic plain of southern Nepal. Our source characterization included motorcycles, kilns, wood and dung cooking-fires, crop-residue burning, diesel and gasoline generators, agricultural pumps, and open garbage burning. We report the emission factors (grams of compound emitted per kilogram of dry fuel burned) for ~80 important trace gases measured by FTIR and WAS, including important NMHCs up to $C_{10}$ and many oxygenated organic compounds. We also measured aerosol mass and optical properties using two PAX systems at 405 and 870 nm. We report important aerosol optical properties that include emission factors (in $m^2$/kg) for scattering and absorption at 405 and 870 nm, single scattering albedo, and absorption Ångström exponent. From the direct measurements of absorption we estimated black and brown carbon emission factors (in g/kg).

Although we were not able to sample the transport sector extensively due to the Gorkha earthquake, we were able to measure several motorbikes pre- and post-service. The minor maintenance led to minimal if any reduction of gaseous pollutants consistent with the idea that more major servicing is needed to reduce gas-phase pollutants. Motorcycles were in general among the least efficient sources sampled and the CO EF was on the order of ~700 g/kg, about ten times that of a typical biomass fire. For most fossil fuel sources, including generators and agricultural pumps, diesel burned more efficiently than gas, but produced more $NO_x$, HCHO, and aerosol.

Numerous trace gas emissions (many for the first time in the field) were quantified for open cooking fires and several improved cooking stoves with several fuel variations. Authentic open dung cooking-fires emitted high levels of BrC (5.54 ± 1.66 g/kg), $NH_3$ (3.00 ± 1.33 g/kg), organic acids (7.66 ± 6.90 g/kg), and HCN (2.01 ± 1.25 g/kg), where the latter could contribute to space-based observations of high levels of HCN in the lower stratosphere above the Asian Monsoon. HCN and some alkynes > $C_2$ (previously linked to BB) were also observed from several non-biomass burning sources. BTEX compounds were major emissions of both dung (~4.5 g/kg) and wood (~1.5 g/kg) cooking-fires and a simple method to estimate indoor exposure to the many important air toxics we measured in the emissions is described. Our PAX data suggest relatively more absorption by BrC as opposed to BC from cooking fires than may be currently recognized; especially for dung burning. Biogas, as expected, emerged as the most efficient and least polluting cooking technology out of approximately a dozen types subjected to limited testing.





The first global garbage burning inventory relied on measurements from very few studies and information for many compounds is often limited to laboratory simulations (Wiedinmyer et al., 2014). Our authentic Nepali garbage burning data shift the global average observed for this source to lower MCE and significantly more BC and BTEX emissions than in previous measurements while supporting previous measurements of high HCl. Crop residue burning produced EFs in good agreement with literature

values with relatively high emissions of oxygenated organic compounds (~12 g/kg) and $SO_2$ (2.54 ± 1.09 g/kg). We observed an EF for BrC of ~11 g/kg or about 4 times higher than the previous organic carbon literature average, which was based on less data. Our EF BrC is qualitative, but in agreement with our absorption data and SSA in showing that BrC absorption is important for this major global BB type.

There are very few studies detailing the chemical emissions from brick kilns. While we were only able to sample two brick kilns

in this study, we present a significant expansion in chemical speciation data. The two brick kilns sampled had different designs and utilized different clay, coal, and amounts of biomass for co-firing with the main coal fuel. Consequently the two kilns produced very different emissions. A zig-zag kiln burning primarily coal at high efficiency produced larger amounts of BC, $NO_x$, HF, and HCl, (the halogen compounds most likely from the clay) while the clamp kiln (with relatively more biomass fuel) produced dramatically more organic gases, organic aerosol (BrC), and aerosol precursors including large alkanes. Both kilns

were significant $SO_2$ sources with their emission factors averaging ~13 g/kg.

Overall, we report the first, or rare, optically- and chemically-detailed emissions data for many undersampled biomass burning sources and other undersampled sources in developing countries. Companion papers will report results from other co-deployed techniques such as filter sampling and mini-AMS, a source apportionment for a fixed supersite, and model interpretation as guidance for mitigation strategies. Future measurements and modeling are also needed to better understand the evolution of the

emissions we report here.

**Acknowledgements**

C. S., T. C., R. Y., purchase of the PAXs, and many other NAMaSTE-associated expenses were supported by NSF grant AGS-1349976. T. J. and E. A. S. were supported by NSF grant AGS-1351616. J. D. G. and P. D. were supported by NSF grant AGS-1461458. We thank G. McMeeking, J. Walker, and S. Murphy for helpful discussions on the PAX instruments and data.



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





Table 1. Fleet average emission factors (g/kg) and standard deviation for two-wheeled vehicle measurements.

| Compound (Formula) | EF Pre-service fleet avg (stdev) | EF Post-service fleet avg (stdev) |
|---|---|---|
| Method | FTIR | FTIR |
| MCE | 0.619 | 0.601 |
| Carbon Dioxide ($CO_2$) | 1846(690) | 1816(562) |
| Carbon Monoxide (CO) | 710(389) | 761(327) |
| Methane ($CH_4$) | 7.60(7.24) | 6.74(4.54) |
| Acetylene ($C_2H_2$) | 11.7(11.1) | 7.89(5.83) |
| Ethylene ($C_2H_4$) | 13.2(3.9) | 11.4(4.2) |
| Propylene ($C_3H_6$) | 3.32(0.75) | 2.58(1.03) |
| Formaldehyde (HCHO) | 0.548 | 0.535 |
| Methanol ($CH_3OH$) | bdl | bdl |
| Formic Acid (HCOOH) | 9.57E-2(3.57E-2) | 5.95E-2(1.84E-2) |
| Acetic Acid ($CH_3COOH$) | bdl | bdl |
| Glycolaldehyde ($C_2H_4O_2$) | bdl | bdl |
| Furan ($C_4H_4O$) | bdl | bdl |
| Hydroxyacetone ($C_3H_6O_2$) | 2.10(3.18) | 2.41(0.99) |
| Phenol ($C_6H_5OH$) | 4.84(3.55) | 3.02(2.29) |
| 1,3-Butadiene ($C_4H_6$) | 1.30(0.51) | 1.19(0.56) |
| Isoprene ($C_5H_8$) | bdl | bdl |
| Ammonia ($NH_3$) | 0.113(0.034) | 0.032(0.023) |
| Hydrogen Cyanide (HCN) | 0.841(0.428) | 0.678(0.174) |
| Nitrous Acid (HONO) | bdl | bdl |
| Sulfur Dioxide ($SO_2$) | bdl | bdl |
| Hydrogen Fluoride (HF) | bdl | bdl |
| Hydrogen chloride (HCl) | bdl | bdl |
| Nitric Oxide (NO) | 2.94(2.39) | 1.89(0.81) |
| Nitrogen Dioxide ($NO_2$) | bdl | bdl |

Note: "bdl" indicates below the detection limit;

C-fraction: 0.85- source is *Kirchstetter et al.* (1999)



Table 2. Emission factors (g/kg) for agricultural diesel irrigation pumps including EFs weighting only startup emissions.

| Compound (Formula) | EF Ag Pump 1 | EF Ag Pump 1 emphasize startup | EF Ag Pump 2 | EF Ag Pump 2 emphasize startup | EF Ag pumps Avg (stdev) |
|---|---|---|---|---|---|
| Method | FTIR | FTIR | FTIR | FTIR | - |
| MCE | 0.987 | 0.974 | 0.996 | 0.990 | 0.992 |
| Carbon Dioxide ($CO_2$) | 3103 | 3038 | 3161 | 3133 | 3132(41) |
| Carbon Monoxide (CO) | 26.0 | 51.3 | 7.36 | 20.2 | 16.7(13.2) |
| Methane ($CH_4$) | 3.80 | 6.14 | 1.41 | 2.85 | 2.61(1.69) |
| Acetylene ($C_2H_2$) | 0.413 | 2.18 | 0.08 | 0.748 | 0.246(0.237) |
| Ethylene ($C_2H_4$) | 5.37 | 9.15 | 1.47 | 3.04 | 3.42(2.75) |
| Propylene ($C_3H_6$) | 1.85 | 3.26 | 0.424 | 0.894 | 1.14(1.01) |
| Formaldehyde (HCHO) | 0.506 | 1.23 | 5.29E-02 | 0.175 | 0.280(0.320) |
| Methanol ($CH_3OH$) | 3.59E-02 | 0.119 | 5.77E-03 | 1.33E-02 | 2.08E-2(2.13E-2) |
| Formic Acid (HCOOH) | bdl | bdl | bdl | bdl | bdl |
| Acetic Acid ($CH_3COOH$) | bdl | bdl | bdl | bdl | bdl |
| Glycolaldehyde ($C_2H_4O_2$) | bdl | bdl | bdl | bdl | bdl |
| Furan ($C_4H_4O$) | bdl | bdl | bdl | bdl | bdl |
| Hydroxyacetone ($C_3H_6O_2$) | bdl | bdl | bdl | bdl | bdl |
| Phenol ($C_6H_5OH$) | 0.449 | 0.583 | 0.117 | 0.258 | 0.283(0.235) |
| 1,3-Butadiene ($C_4H_6$) | 0.809 | 1.47 | 0.194 | 0.399 | 0.501(0.435) |
| Isoprene ($C_5H_8$) | 1.55E-02 | 7.20E-02 | 1.93E-02 | 2.30E-02 | 1.74E-2(2.69E-3) |
| Ammonia ($NH_3$) | 9.27E-03 | 6.42E-02 | 1.32E-03 | 1.32E-03 | 5.29E-3(5.62E-3) |
| Hydrogen Cyanide (HCN) | 0.188 | 0.458 | 4.77E-02 | 0.282 | 0.118(0.099) |
| Nitrous Acid (HONO) | 0.348 | 0.307 | 0.346 | 0.373 | 0.347(0.001) |
| Sulfur Dioxide ($SO_2$) | bdl | bdl | bdl | bdl | bdl |
| Hydrogen Fluoride (HF) | bdl | bdl | bdl | bdl | bdl |
| Hydrogen chloride (HCl) | bdl | bdl | bdl | bdl | bdl |
| Nitric Oxide (NO) | 5.31 | 5.09 | 15.9 | 15.7 | 10.6(7.5) |
| Nitrogen Dioxide ($NO_2$) | 2.19 | 1.86 | 1.20 | 1.15 | 1.69(0.70) |
| EF Black Carbon (BC) | 6.13 | - | 5.31 | - | 5.72(0.58) |
| EF $B_{abs}$ 405 nm ($m^2$/kg) | 62.4 | - | 54.1 | - | 58.3(5.9) |
| EF $B_{scat}$ 405 nm ($m^2$/kg) | 62.9 | - | 24.0 | - | 43.4(27.5) |
| SSA 405 nm | 0.502 | - | 0.307 | - | 0.405(0.137) |

Note: "bdl" indicates below the detection limit; C-fraction: 0.85- source is *Kirchstetter et al.* (1999)



Atmospheric Chemistry and Physics Discussions — Open Access — EGU

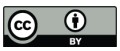

Table 3. Emission factors (g/kg) for individual garbage burns sampled during NAMaSTE and average EFs and one standard deviation for mixed garbage burning.

| Compound (Formula) | EF Mixed garbage 1 | EF Mixed garbage 2 | EF Mixed garbage 3 | EF Mixed garbage 4 | EF Mixed garbage 5 | EF Mixed garbage 6 | EF Mixed Chip bags | EF Plastics burn 1 | EF Plastics burn 2 | EF Mixed garbage avg (stdev) |
|---|---|---|---|---|---|---|---|---|---|---|
| Method | FTIR+WAS | FTIR+WAS | WAS | WAS | WAS | WAS | FTIR | FTIR | WAS | - |
| MCE | 0.937 | 0.980 | 0.926 | 0.863 | 0.864 | 0.967 | 0.989 | 0.962 | 0.990 | 0.923(0.050) |
| Carbon Dioxide ($CO_2$) | 1446 | 1773 | 1641 | 1498 | 1498 | 1756 | 2249 | 2473 | 2695 | 1602(142) |
| Carbon Monoxide (CO) | 61.5 | 22.8 | 84 | 152 | 151 | 38.0 | 15.9 | 62.2 | 16.6 | 84.7(55.5) |
| Methane ($CH_4$) | 2.22 | 0.531 | 4.15 | 12.5 | 3.82 | 0.542 | 0.279 | 2.04 | 0.684 | 3.97(4.47) |
| Acetylene ($C_2H_2$) | 1.49 | 0.261 | 0.269 | 0.101 | 0.674 | 1.18 | 0.434 | 2.23 | 0.298 | 0.662(0.562) |
| Ethylene ($C_2H_4$) | 9.33 | 0.768 | 2.05 | 1.72 | 3.725 | 0.578 | 1.85 | 9.36 | 0.477 | 3.03(3.29) |
| Propylene ($C_3H_6$) | 1.98 | 0.426 | 1.940 | 1.999 | 3.884 | 0.167 | 0.520 | 3.53 | 0.150 | 1.73(1.34) |
| Formaldehyde (HCHO) | 4.15 | 0.507 | nm | nm | nm | nm | 0.475 | 5.23 | nm | 2.33(2.57) |
| Methanol ($CH_3OH$) | 1.23 | 0.146 | 0.271 | 2.429 | 0.590 | 3.38E-02 | 3.43E-02 | 0.98 | bdl | 0.783(0.914) |
| Formic Acid (HCOOH) | 0.585 | 0.323 | nm | nm | nm | nm | 0.126 | 5.30 | nm | 0.454(0.185) |
| Acetic Acid ($CH_3COOH$) | 1.63 | 0.118 | nm | nm | nm | nm | 4.42E-02 | 2.22 | nm | 0.872(1.066) |
| Glycolaldehyde ($C_2H_4O_2$) | 2.41 | bdl | nm | nm | nm | nm | 2.44E-02 | 4.56 | nm | 2.41(-) |
| Furan ($C_4H_4O$) | 0.349 | 7.77E-02 | nm | nm | nm | nm | bdl | 0.234 | nm | 0.213(0.192) |
| Hydroxyacetone ($C_3H_6O_2$) | 2.70 | 0.664 | nm | nm | nm | nm | bdl | 2.59 | nm | 1.68(1.44) |
| Phenol ($C_6H_5OH$) | 0.776 | 5.09E-02 | nm | nm | nm | nm | 0.127 | 1.42 | nm | 0.414(0.513) |
| 1,3-Butadiene ($C_4H_6$) | 0.930 | 0.127 | 0.205 | 0.177 | 0.116 | 4.86E-02 | 0.192 | 1.07 | 3.41E-04 | 0.267(0.329) |
| Isoprene ($C_5H_8$) | 0.145 | bdl | 1.84E-02 | 0.103 | bdl | 6.80E-04 | 9.59E-02 | 0.226 | bdl | 6.67E-2(6.86E-2) |
| Ammonia ($NH_3$) | bdl | 0.761 | nm | nm | nm | nm | bdl | 5.66E-02 | nm | 0.761(-) |
| Hydrogen Cyanide (HCN) | 0.551 | 0.312 | nm | nm | nm | nm | 0.374 | 0.955 | nm | 0.432(0.169) |
| Nitrous Acid (HONO) | 0.564 | 0.422 | nm | nm | nm | nm | 0.164 | 2.50 | nm | 0.493(0.100) |
| Sulfur Dioxide ($SO_2$) | bdl | bdl | nm | nm | nm | nm | bdl | bdl | nm | bdl |
| Hydrogen Fluoride (HF) | bdl | bdl | nm | nm | nm | nm | bdl | bdl | nm | bdl |
| Hydrogen chloride (HCl) | 3.03 | 1.61 | nm | nm | nm | nm | bdl | 77.9 | nm | 2.32(1.01) |
| Nitric Oxide (NO) | 1.43 | 1.61 | nm | nm | nm | nm | 2.02 | 2.36 | nm | 1.52(0.12) |
| Nitrogen Dioxide ($NO_2$) | 1.14 | 0.983 | nm | nm | nm | nm | 1.20 | 1.69 | nm | 1.06(0.11) |
| Carbonyl sulfide (OCS) | 0.133 | 2.71E-02 | 8.62E-02 | 8.03E-02 | 0.106 | 1.33E-02 | nm | nm | 2.03E-02 | 7.43E-2(4.60E-2) |
| DMS ($C_2H_6S$) | - | 1.27E-03 | 1.89E-03 | 2.70E-02 | 6.74E-03 | 4.71E-05 | nm | nm | 1.19E-02 | 7.39E-3(1.13E-2) |
| Chloromethane ($CH_3Cl$) | 0.895 | 5.05E-02 | 0.343 | 1.59 | 1.26 | 6.55E-02 | nm | nm | 5.72E-02 | 0.702(0.648) |
| Bromomethane ($CH_3Br$) | 6.71E-03 | 5.47E-04 | 2.93E-03 | 1.16E-03 | 1.41E-03 | 3.96E-04 | nm | nm | 5.53E-05 | 2.19E-3(2.39E-3) |
| Methyl iodide ($CH_3I$) | 3.26E-04 | - | 4.41E-04 | 4.81E-04 | 2.55E-04 | 1.21E-04 | nm | nm | 1.54E-05 | 3.25E-4(1.45E-4) |
| 1,2-Dichloroethene ($C_2H_2Cl_2$) | 0.260 | 1.44E-02 | 4.75E-03 | 2.70E-03 | 1.02E-02 | 4.92E-03 | nm | nm | 5.94E-04 | 4.96E-2(1.03E-1) |
| Methyl nitrate ($CH_3NO_3$) | 0.185 | 6.45E-02 | 2.21E-02 | 1.02E-02 | 7.61E-02 | 8.44E-04 | nm | nm | 7.99E-02 | 5.98E-2(6.84E-2) |
| Ethane ($C_2H_6$) | 5.64 | 6.09E-02 | 0.830 | 2.11 | 1.42 | 7.19E-02 | nm | nm | 3.04E-02 | 1.69(2.09) |
| Propane ($C_3H_8$) | 3.15 | 2.52E-02 | 0.388 | 0.913 | 0.920 | 3.01E-02 | nm | nm | 1.68E-02 | 0.904(1.169) |
| i-Butane ($C_4H_{10}$) | 0.445 | 1.25E-03 | 3.81E-02 | 5.79E-02 | 6.52E-02 | - | nm | nm | 0.002 | 0.122(0.183) |
| n-Butane ($C_4H_{10}$) | 1.87 | 1.41E-02 | 0.190 | 0.341 | 0.650 | 1.19E-02 | nm | nm | 1.86E-02 | 0.513(0.707) |
| 1-Butene ($C_4H_8$) | 3.89 | 8.36E-02 | 0.569 | 0.502 | 1.23 | 5.51E-02 | nm | nm | 6.45E-02 | 1.05(1.45) |
| i-Butene ($C_4H_8$) | 1.93 | 5.80E-02 | 0.508 | 0.400 | 0.829 | 2.62E-02 | nm | nm | 7.90E-03 | 0.625(0.705) |



| Species | | | | | | | | | | | | |
|---|---|---|---|---|---|---|---|---|---|---|---|---|
| trans-2-Butene (C₄H₈) | 0.630 | 7.09E-03 | 9.55E-02 | 0.135 | 0.160 | 6.89E-03 | nm | nm | nm | 1.28E-02 | nm | 0.172(0.233) |
| cis-2-Butene (C₄H₈) | 0.580 | 6.04E-03 | 7.27E-02 | 9.72E-02 | 0.102 | 4.91E-03 | nm | nm | nm | 9.46E-03 | nm | 0.144(0.218) |
| i-Pentane (C₅H₁₂) | 1.13 | - | 2.00E-02 | - | 2.43E-02 | - | nm | nm | nm | 3.00E-02 | nm | 0.391(0.639) |
| n-Pentane (C₅H₁₂) | 4.09 | 3.90E-02 | 0.435 | 0.698 | 1.21 | 1.69E-02 | nm | nm | nm | 1.85E-02 | nm | 1.08(1.54) |
| 1-Pentene (C₅H₁₀) | 2.53 | 4.19E-02 | 0.341 | 0.374 | 1.07 | 2.86E-02 | nm | nm | nm | 3.75E-02 | nm | 0.731(0.960) |
| trans-2-Pentene (C₅H₁₀) | 0.700 | 1.67E-02 | 0.108 | 0.126 | 0.270 | 6.63E-03 | nm | nm | nm | 9.65E-03 | nm | 0.205(0.260) |
| cis-2-Pentene (C₅H₁₀) | 0.320 | 7.43E-03 | 5.14E-02 | 5.73E-02 | 0.118 | 2.83E-03 | nm | nm | nm | 4.29E-03 | nm | 9.29E-2(1.19E-1) |
| 3-Methyl-1-butene (C₅H₁₀) | 0.129 | 3.80E-03 | 3.99E-02 | 2.63E-02 | 4.66E-02 | 1.98E-03 | nm | nm | nm | 2.79E-03 | nm | 4.12E-2(4.65E-2) |
| 1,2-Propadiene (C₃H₄) | 0.198 | 1.74E-02 | 2.84E-02 | 3.92E-03 | 6.76E-02 | 1.25E-02 | nm | nm | nm | 5.38E-03 | nm | 5.47E-2(7.39E-2) |
| Propyne (C₃H₄) | 0.315 | 3.27E-02 | 5.41E-02 | 1.18E-02 | 9.60E-02 | 2.92E-02 | nm | nm | nm | 1.10E-02 | nm | 8.99E-2(1.14E-1) |
| 1-Butyne (C₄H₆) | 3.61E-02 | 2.08E-03 | - | 1.84E-03 | 1.12E-02 | 1.17E-03 | nm | nm | nm | 8.69E-04 | nm | 1.05E-2(1.49E-2) |
| 2-Butyne (C₄H₆) | 2.46E-02 | 1.07E-03 | - | 1.47E-03 | 8.79E-03 | 7.65E-04 | nm | nm | nm | 4.00E-04 | nm | 7.34E-3(1.02E-2) |
| n-Hexane (C₆H₁₄) | 0.761 | - | 0.101 | 0.126 | 0.417 | 5.05E-03 | nm | nm | nm | 1.54E-02 | nm | 0.282(0.309) |
| n-Heptane (C₇H₁₆) | 0.707 | 9.86E-03 | 9.61E-02 | 0.154 | 0.413 | 5.41E-03 | nm | nm | nm | 5.10E-03 | nm | 0.231(0.277) |
| n-Octane (C₈H₁₈) | 0.411 | 1.24E-02 | 6.53E-02 | 0.078 | 0.313 | 1.36E-03 | nm | nm | nm | 1.24E-02 | nm | 0.147(0.172) |
| n-Nonane (C₉H₂₀) | 0.134 | 3.81E-03 | 5.94E-02 | 0.076 | 0.158 | 3.68E-03 | nm | nm | nm | 2.77E-02 | nm | 7.24E-2(6.43E-2) |
| n-Decane (C₁₀H₂₂) | 0.266 | 1.00E-02 | 7.99E-02 | 0.153 | 0.224 | 2.36E-02 | nm | nm | nm | bdl | nm | 0.126(0.106) |
| 2,3-Dimethylbutane (C₆H₁₄) | 3.73E-02 | - | 3.11E-03 | 8.79E-04 | 3.62E-03 | - | nm | nm | nm | 2.70E-03 | nm | 1.12E-2(1.74E-2) |
| 2-Methylpentane (C₆H₁₄) | 0.342 | 3.36E-03 | 4.32E-02 | 6.59E-02 | 9.48E-02 | - | nm | nm | nm | 4.35E-03 | nm | 0.110(0.134) |
| 3-Methylpentane (C₆H₁₄) | 8.60E-02 | 0.228 | bdl | bdl | bdl | bdl | nm | nm | nm | 1.64E-03 | nm | 0.157(0.100) |
| 2,2,4-Trimethylpentane (C₈H₁₈) | bdl | bdl | bdl | bdl | bdl | bdl | nm | nm | nm | bdl | nm | bdl |
| Cyclopentane (C₅H₁₀) | 5.84E-02 | 1.63E-04 | 1.00E-02 | 4.35E-03 | 1.41E-02 | 3.05E-04 | nm | nm | nm | 7.39E-04 | nm | 1.46E-2(2.22E-2) |
| Cyclohexane (C₆H₁₂) | bdl | 9.80E-03 | bdl | bdl | bdl | bdl | nm | nm | nm | 2.85E-03 | nm | 9.80E-3(-) |
| Methylcyclohexane (C₇H₁₄) | 0.100 | 1.71E-03 | 3.61E-03 | - | 5.23E-03 | bdl | nm | nm | nm | - | nm | 2.76E-2(4.81E-2) |
| Benzene (C₆H₆) | 5.66 | 0.389 | 2.74 | 1.59 | 3.60 | 1.68 | nm | nm | nm | 0.285 | nm | 2.61(1.85) |
| Toluene (C₇H₈) | 2.68 | 5.74E-02 | 0.574 | 0.645 | 0.802 | 0.139 | nm | nm | nm | 3.23E-02 | nm | 0.817(0.960) |
| Ethylbenzene (C₈H₁₀) | 2.18 | 2.11E-02 | 0.232 | 0.239 | 0.289 | 2.75E-02 | nm | nm | nm | 1.61E-02 | nm | 0.498(0.831) |
| m/p-Xylene (C₈H₁₀) | 1.14 | 3.42E-02 | 0.279 | 0.329 | 0.228 | 3.55E-02 | nm | nm | nm | 1.41E-02 | nm | 0.342(0.412) |
| o-Xylene (C₈H₁₀) | 0.657 | 1.78E-02 | 0.153 | 0.195 | 0.296 | 1.92E-02 | nm | nm | nm | 7.75E-03 | nm | 0.223(0.238) |
| Styrene (C₈H₈) | 0.347 | 3.33E-03 | 0.493 | 0.811 | 0.349 | 0.199 | nm | nm | nm | 2.00E-03 | nm | 0.367(0.274) |
| i-Propylbenzene (C₉H₁₂) | 6.80E-02 | bdl | - | 9.97E-03 | 5.58E-03 | 1.20E-03 | nm | nm | nm | 1.19E-03 | nm | 2.12E-2(3.14E-2) |
| n-Propylbenzene (C₉H₁₂) | 7.19E-02 | 3.29E-03 | 2.45E-02 | 2.43E-02 | 5.79E-02 | 3.73E-03 | nm | nm | nm | 2.35E-03 | nm | 3.09E-2(2.83E-2) |
| 3-Ethyltoluene (C₉H₁₂) | 0.128 | 4.97E-03 | 2.95E-02 | 2.67E-02 | 2.18E-02 | 2.50E-03 | nm | nm | nm | 1.84E-03 | nm | 3.55E-2(2.46E-2) |
| 4-Ethyltoluene (C₉H₁₂) | 4.28E-02 | 2.59E-03 | 2.03E-02 | 2.26E-02 | 1.69E-02 | 1.22E-03 | nm | nm | nm | 9.27E-04 | nm | 1.77E-2(1.52E-2) |
| 2-Ethyltoluene (C₉H₁₂) | 6.49E-02 | 2.45E-03 | 1.79E-02 | 2.06E-02 | 2.95E-02 | 2.72E-03 | nm | nm | nm | 1.65E-03 | nm | 2.30E-2(2.31E-2) |
| 1,3,5-Trimethylbenzene (C₉H₁₂) | 6.33E-02 | 3.78E-03 | 3.73E-02 | 4.34E-02 | 5.17E-02 | 2.40E-03 | nm | nm | nm | 9.73E-04 | nm | 3.36E-2(2.52E-2) |
| 1,2,4-Trimethylbenzene (C₉H₁₂) | 7.24E-02 | 5.70E-03 | 2.49E-02 | 2.34E-02 | 2.29E-02 | 4.25E-03 | nm | nm | nm | 1.67E-03 | nm | 2.56E-2(2.47E-2) |
| 1,2,3-Trimethylbenzene (C₉H₁₂) | 2.15E-02 | 2.53E-03 | 2.21E-02 | 2.49E-02 | 1.55E-02 | 4.76E-03 | nm | nm | nm | 1.44E-03 | nm | 1.52E-2(9.49E-3) |
| alpha-Pinene (C₁₀H₁₆) | 1.66E-02 | bdl | 0.135 | 2.40E-02 | 2.48E-02 | bdl | nm | nm | nm | 7.81E-03 | nm | 5.00E-2(5.65E-2) |
| beta-Pinene (C₁₀H₁₆) | - | bdl | - | bdl | 8.27E-02 | 3.10E-04 | nm | nm | nm | - | nm | 4.15E-2(5.83E-2) |
| Ethanol (C₂H₆O) | 8.39 | 6.01E-02 | 0.103 | 0.147 | 0.117 | 1.06E-02 | nm | nm | nm | 0.143 | nm | 8.74E-2(5.31E-2) |
| Acetaldehyde (C₂H₄O) | 5.38 | 0.271 | 1.167 | 2.51 | 0.276 | 0.108 | nm | nm | nm | 0.950 | nm | 2.12(3.20) |
| Acetone (C₃H₆O) | 0.907 | 1.01 | 1.04 | 2.42 | 3.57 | 0.380 | nm | nm | nm | | nm | 2.30(1.90) |
| Butanal (C₄H₈O) | 0.907 | 4.22E-02 | 7.68E-02 | 0.102 | 0.415 | 1.40E-02 | nm | nm | nm | 6.21E-02 | nm | 0.259(0.349) |
| Butanone (C₄H₈O) | 0.755 | 5.37E-02 | 1.94E-03 | 0.419 | 1.89E-02 | 2.54E-02 | nm | nm | nm | 0.472 | nm | 0.212(0.310) |



| | | | | | | | | |
|---|---|---|---|---|---|---|---|---|
| EF Black Carbon (BC) | 0.561 | 6.04 | nm | nm | nm | 1.58 | 1.69 | 3.30(3.88) |
| EF Brown Carbon (BrC) | - | - | nm | nm | nm | - | - | - |
| EF $B_{abs}$ 405 (m²/kg) | 5.72 | 60.2 | nm | nm | nm | 16.1 | 17.3 | - |
| EF $B_{scat}$ 405 (m²/kg) | 197 | 52 | nm | nm | nm | 26.6 | 70.0 | - |
| EF $B_{abs}$ 870 (m²/kg) | nm | 28.6 | nm | nm | nm | nm | nm | - |
| EF $B_{scat}$ 870 (m²/kg) | nm | 14.1 | nm | nm | nm | nm | nm | - |
| SSA 405 nm | 0.972 | 0.463 | nm | nm | nm | 0.623 | 0.802 | - |
| SSA 870 nm | - | 0.329 | nm | nm | nm | - | - | - |
| AAE | - | 0.971 | nm | nm | nm | nm | nm | - |

Note: "bdl" indicates below the detection limit; "-" indicates concentrations were not greater than background; "nm" indicates not measured; See Table S1 for garbage compositions

C-fractions: mixed garbage (0.50)-source is *Stockwell et al.* (2014); plastics (0.74) & chip bags (0.63)- source is *USEPA*, 2010 (see Sect. 2.1.4 for details)



Table 4. Compiled emission factors (g/kg) and one standard deviation for open traditional cooking fires using dung and wood fuels. The NAMaSTE values include field measurements and adjusted laboratory measurements.

| Compound (Formula) | EF Hardwood cooking EF NAMaSTE avg (stdev)[a] | EF Dung cooking NAMaSTE avg (stdev) | EF wood open cooking Akagi et al. [2011] avg (stdev) | EF wood open cooking Stockwell et al. [2015] avg (stdev)[b] | EF Dung burning Akagi et al. [2011] avg (stdev) |
|---|---|---|---|---|---|
| MCE | 0.923 | 0.898 | 0.927 | 0.927 | 0.839 |
| PM | - | - | 6.73(1.61) | - | 22.9 |
| Carbon Dioxide ($CO_2$) | 1462(16) | 1129(80) | 1548(125) | 1548(125) | 859(15) |
| Carbon Monoxide (CO) | 77.2(13.5) | 80.9(13.8) | 77.4(26.2) | 77.4 (26) | 105(10) |
| Methane ($CH_4$) | 5.16(1.39) | 6.65(0.46) | 4.86(2.73) | 4.86(0.20) | 11.0(3.3) |
| Acetylene ($C_2H_2$) | 0.764(0.363) | 0.593(0.443) | 0.970(0.503) | 0.602(0.361) | nm |
| Ethylene ($C_2H_4$) | 2.70(1.17) | 4.23(1.39) | 1.53(0.66) | 2.21(1.40) | 1.12(0.23) |
| Propylene ($C_3H_6$) | 0.576(0.195) | 1.47(0.58) | 0.565(0.338) | 0.317(0.145) | 1.89(0.42) |
| Formaldehyde (HCHO) | 1.94(0.71) | 2.42(1.40) | 2.08(1.40) | 1.70(0.74) | nm |
| Methanol ($CH_3OH$) | 1.92(0.61) | 2.38(0.90) | 2.26(1.27) | 2.05(1.63) | 4.14(0.88) |
| Formic Acid (HCOOH) | 0.179(0.071) | 0.341(0.308) | 0.220(0.168) | 0.620(0.533) | 0.460(0.308) |
| Acetic Acid ($CH_3COOH$) | 3.14(1.11) | 7.32(6.59) | 4.97(3.32) | 8.90(9.27) | 11.7(5.1) |
| Glycolaldehyde ($C_2H_4O_2$) | 0.238(0.155) | 0.499(0.260) | 1.42(-) | 0.455(0.149) | nm |
| Furan ($C_4H_4O$) | 0.241(0.024) | 0.534(0.209) | 0.400(-) | 0.228(0.162) | 0.950(0.220) |
| Hydroxyacetone ($C_3H_6O_2$) | 1.26(0.09) | 3.19(2.24) | nm | 0.480(0.367) | 9.60(2.38) |
| Phenol ($C_6H_5OH$) | 0.496(0.159) | 1.008(0.348) | 3.32(-) | 0.264(0.085) | 2.16(0.36) |
| 1,3-Butadiene ($C_4H_6$) | 0.204(0.144) | 0.409(0.306) | nm | 3.37E-2(9.67E-3) | nm |
| Isoprene ($C_5H_8$) | 4.16E-2(2.23E-2) | 0.325(0.443) | nm | 0.145(0.077) | nm |
| Ammonia ($NH_3$) | 0.259(0.253) | 3.00(1.33) | 0.865(0.404) | 7.88E-2(6.90E-2) | 4.75(1.00) |
| Hydrogen Cyanide (HCN) | 0.557(0.247) | 2.01(1.25) | nm | 0.221(0.005) | 0.530(0.300) |
| Nitrous Acid (HONO) | 0.452(0.068) | 0.276(0.101) | nm | 0.291(0.169) | nm |
| Sulfur Dioxide ($SO_2$) | bdl | bdl | nm | 0.499 | 6.00E-2(-) |
| Hydrogen Fluoride (HF) | bdl | bdl | nm | bdl | nm |
| Hydrogen chloride (HCl) | 7.51E-2(7.99E-2) | 3.76E-2(3.59E-2) | nm | bdl | nm |
| Nitric Oxide (NO) | 1.62(1.30) | 2.22(1.02) | 1.72(0.75) | 0.319(0.089) | 0.500 |
| Nitrogen Dioxide ($NO_2$) | 0.577(0.348) | 0.898(0.444) | 0.490(0.330) | 1.11(0.28) | nm |
| Carbonyl sulfide (OCS) | 1.87E-2(1.15E-2) | 0.148(0.123) | nm | nm | nm |
| DMS ($C_2H_6S$) | 0.255(0.359) | 2.37E-2(7.67E-4) | nm | nm | nm |
| Chloromethane ($CH_3Cl$) | 2.36E-2(1.62E-2) | 1.60(1.53) | nm | nm | nm |
| Bromomethane ($CH_3Br$) | 5.61E-4(3.01E-4) | 5.34E-3(3.02E-3) | nm | nm | nm |
| Methyl iodide ($CH_3I$) | 1.23E-4(1.11E-4) | 4.39E-4(1.78E-4) | nm | nm | nm |
| 1,2-Dichloroethene ($C_2H_2Cl_2$) | 1.24E-4(3.00E-5) | 4.97E-3(-) | nm | nm | nm |
| Methyl nitrate ($CH_3NO_3$) | 6.96E-3(5.73E-3) | 1.46E-2(1.94E-2) | nm | nm | nm |
| Ethane ($C_2H_6$) | 0.160(0.122) | 1.075(0.300) | 1.50(0.50) | nm | nm |
| Propane ($C_3H_8$) | 0.202(0.140) | 0.457(0.137) | nm | nm | nm |
| i-Butane ($C_4H_{10}$) | 0.406(0.478) | 0.215(0.126) | nm | nm | nm |
| n-Butane ($C_4H_{10}$) | 1.11(1.48) | 0.29(0.09) | nm | nm | nm |
| 1-Butene ($C_4H_8$) | 0.726(0.904) | 0.399(0.331) | nm | 0.245(0.148) | nm |
| i-Butene ($C_4H_8$) | 0.846(1.113) | 0.281(0.091) | nm | nm | nm |
| trans-2-Butene ($C_4H_8$) | 6.78E-2(5.98E-2) | 0.151(0.010) | nm | nm | nm |
| cis-2-Butene ($C_4H_8$) | 5.51E-2(4.76E-2) | 0.102(0.016) | nm | nm | nm |
| i-Pentane ($C_5H_{12}$) | 8.58E-2(1.58E-2) | 0.811(0.387) | nm | nm | nm |
| n-Pentane ($C_5H_{12}$) | 2.18E-2(1.73E-2) | 0.190(0.254) | nm | nm | nm |
| 1-Pentene ($C_5H_{10}$) | 1.43E-2(9.36E-3) | 0.168(0.086) | nm | nm | nm |
| trans-2-Pentene ($C_5H_{10}$) | 1.05E-2(8.30E-3) | 0.115(0.035) | nm | nm | nm |
| cis-2-Pentene ($C_5H_{10}$) | 8.69E-3(-) | 5.14E-2(7.55E-3) | nm | nm | nm |
| 3-Methyl-1-butene ($C_5H_{10}$) | 7.43E-3(5.79E-3) | 5.58E-2(3.50E-2) | nm | nm | nm |
| 1,2-Propadiene ($C_3H_4$) | 2.33E-2(1.07E-2) | 7.15E-2(6.76E-2) | nm | nm | nm |
| Propyne ($C_3H_4$) | 6.39E-2(3.07E-2) | 0.172(0.156) | nm | nm | nm |
| 1-Butyne ($C_4H_6$) | 1.28E-2(4.73E-3) | 2.29E-2(1.38E-2) | nm | nm | nm |
| 2-Butyne ($C_4H_6$) | 1.02E-2(6.56E-3) | 1.86E-2(9.11E-3) | nm | nm | nm |
| n-Hexane ($C_6H_{14}$) | 1.85E-2(-) | 0.291(0.248) | nm | nm | nm |
| n-Heptane ($C_7H_{16}$) | 1.01E-2(1.35E-2) | 0.114(0.069) | nm | nm | nm |
| n-Octane ($C_8H_{18}$) | 1.75E-2(-) | 4.77E-2(9.85E-3) | nm | nm | nm |




| | | | | | |
|---|---|---|---|---|---|
| n-Nonane ($C_9H_{20}$) | 4.87E-2(6.40E-2) | 4.68E-2(2.55E-2) | nm | nm | nm |
| n-Decane ($C_{10}H_{22}$) | 6.90E-2(9.61E-2) | 4.71E-2(4.03E-2) | nm | nm | nm |
| 2,3-Dimethylbutane ($C_6H_{14}$) | 1.57E-2(1.16E-2) | 0.112(0.105) | nm | nm | nm |
| 2-Methylpentane ($C_6H_{14}$) | 9.93E-3(1.29E-2) | 0.231(0.192) | nm | nm | nm |
| 3-Methylpentane ($C_6H_{14}$) | 6.79E-3(6.63E-3) | 0.155(0.137) | nm | nm | nm |
| 2,2,4-Trimethylpentane ($C_8H_{18}$) | -(-) | 0.100(0.080) | nm | nm | nm |
| Cyclopentane ($C_5H_{10}$) | 4.06E-3(-) | 0.146(0.178) | nm | nm | nm |
| Cyclohexane ($C_6H_{12}$) | 1.16E-2(-) | 0.224(0.255) | nm | nm | nm |
| Methylcyclohexane ($C_7H_{14}$) | 1.62E-2(-) | 4.76E-2(3.96E-2) | nm | nm | nm |
| Benzene ($C_6H_6$) | 1.05(0.19) | 1.96(0.45) | nm | 2.58(2.68) | nm |
| Toluene ($C_7H_8$) | 0.241(0.160) | 1.26(0.05) | nm | 0.290(0.311) | nm |
| Ethylbenzene ($C_8H_{10}$) | 4.19E-2(4.25E-2) | 0.366(0.085) | nm | nm | nm |
| m/p-Xylene ($C_8H_{10}$) | 9.57E-2(7.99E-2) | 0.601(0.294) | nm | 0.265(0.380) | nm |
| o-Xylene ($C_8H_{10}$) | 3.93E-2(4.31E-2) | 0.228(0.083) | nm | nm | nm |
| Styrene ($C_8H_8$) | 8.71E-2(6.69E-2) | 0.255(0.091) | nm | 0.234(0.306) | nm |
| i-Propylbenzene ($C_9H_{12}$) | 1.70E-2(1.67E-2) | 1.87E-2(1.40E-2) | nm | nm | nm |
| n-Propylbenzene ($C_9H_{12}$) | 1.78E-2(1.58E-2) | 3.10E-1(1.45E-2) | nm | nm | nm |
| 3-Ethyltoluene ($C_9H_{12}$) | 2.62E-2(5.41E-3) | 5.61E-2(2.38E-2) | nm | nm | nm |
| 4-Ethyltoluene ($C_9H_{12}$) | 2.07E-2(1.19E-2) | 3.57E-2(1.74E-2) | nm | nm | nm |
| 2-Ethyltoluene ($C_9H_{12}$) | 2.10E-2(1.16E-2) | 3.39E-2(1.34E-2) | nm | nm | nm |
| 1,3,5-Trimethylbenzene ($C_9H_{12}$) | 2.14E-2(-) | 1.79E-2(8.32E-3) | nm | 7.01E-2(9.27E-2) | nm |
| 1,2,4-Trimethylbenzene ($C_9H_{12}$) | 1.74E-2(2.35E-2) | 3.91E-2(1.65E-2) | nm | nm | nm |
| 1,2,3-Trimethylbenzene ($C_9H_{12}$) | 2.16E-2(-) | 2.34E-2(4.30E-3) | nm | nm | nm |
| alpha-Pinene ($C_{10}H_{16}$) | 2.02E-2(2.33E-2) | 0.348(0.487)[c] | nm | 0.197(0.257) | nm |
| beta-Pinene ($C_{10}H_{16}$) | 4.67E-2(-) | 0.471(-)[c] | nm | nm | nm |
| Ethanol ($C_2H_6O$) | 0.128(0.017) | 0.563(0.589) | nm | nm | nm |
| Acetaldehyde ($C_2H_4O$) | 0.541(0.362) | 1.88(1.63) | nm | 0.792(0.439) | nm |
| Acetone ($C_3H_6O$) | 0.524(0.256) | 1.63(0.38) | nm | nm | nm |
| Butanal ($C_4H_8O$) | 8.28E-3(6.27E-3) | 5.40E-2(2.19E-2) | nm | nm | nm |
| Butanone ($C_4H_8O$) | 0.232(0.286) | 0.262(0.109) | nm | 8.04E-2(4.98E-2) | nm |
| EF Black Carbon (BC) | 0.221(0.127) | 4.15E-2(3.18E-2) | 0.833(0.453) | nm | nm |
| EF Brown Carbon (BrC) | 8.59(5.62) | 5.54(1.66) | nm | nm | nm |
| EF $B_{abs}$ 405 ($m^2$/kg) | 10.6(6.8) | 5.85(1.95) | nm | nm | nm |
| EF $B_{scat}$ 405 ($m^2$/kg) | 40.4(23.8) | 49.5(5.8) | nm | nm | nm |
| EF $B_{abs}$ 870 ($m^2$/kg) | 1.04(0.60) | 0.197(0.151) | nm | nm | nm |
| EF $B_{scat}$ 870 ($m^2$/kg) | 1.51(0.52) | 0.922(0.324) | nm | nm | nm |
| EF $B_{abs}$ 405 just BrC ($m^2$/kg) | 8.40(5.48) | 5.43(1.62) | nm | nm | nm |
| EF $B_{abs}$ 405 just BC ($m^2$/kg) | 2.24(1.28) | 0.423(0.324) | nm | nm | nm |
| SSA 405 nm | 0.605(0.061) | 0.811(0.164) | nm | nm | nm |
| SSA 870 nm | 0.794(0.009) | 0.893(0.043) | nm | nm | nm |
| AAE | 3.01(0.10) | 4.63(0.68) | nm | nm | nm |

Note: "bdl" indicates below the detection limit; "-" indicates concentrations were not greater than background; "nm" indicates not measured

[a] NAMaSTE gas-phase data include adjusted laboratory and unadjusted field values. Aerosol values include field measurements only (see Sect. 3.6)

[b] This includes laboratory adjusted values (see *Stockwell* et al., (2014,2015)); additional gas-phase compounds are reported therein

[c] High monoterpene values likely due to wood kindling



Table 5. Summary of emission factors (g/kg) and one standard deviation for crop residue burns from this study and others.

| Compound (Formula) | EF Crop Residue Yokelson et al. [2011] avg (stdev)[a] | EF Crop Residue (food fuels) Stockwell et al. [2015] avg (stdev) | EF Crop Residue NAMaSTE avg (stdev)[b,c] |
|---|---|---|---|
| MCE | 0.925 | 0.925 | 0.925 |
| Carbon Dioxide ($CO_2$) | 1398(55) | 1353(80) | 1401(68) |
| Carbon Monoxide (CO) | 71.9(28.4) | 68.7(25.2) | 72.3(23.9) |
| Methane ($CH_4$) | 4.21(3.53) | 3.49(2.19) | 2.79(0.85) |
| Acetylene ($C_2H_2$) | 0.193(0.059) | 0.331(0.277) | 0.216(0.063) |
| Ethylene ($C_2H_4$) | 0.974(0.470) | 1.34(0.80) | 0.890(0.230) |
| Propylene ($C_3H_6$) | 0.417(0.224) | 0.576(0.415) | 0.492(0.094) |
| Formaldehyde (HCHO) | 1.55(0.78) | 1.93(1.32) | 0.865(0.298) |
| Methanol ($CH_3OH$) | 2.24(1.33) | 1.87(1.53) | 1.01(0.37) |
| Formic Acid (HCOOH) | 0.840(0.571) | 0.633(0.846) | 0.119(0.055) |
| Acetic Acid ($CH_3COOH$) | 3.80(2.35) | 3.88(3.64) | 0.871(0.719) |
| Glycolaldehyde ($C_2H_4O_2$) | - | 2.29(3.04) | 4.07(4.03) |
| Furan ($C_4H_4O$) | - | 0.355(0.445) | 0.116(0.049) |
| Hydroxyacetone ($C_3H_6O_2$) | - | 1.69(2.03) | 1.48(0.62) |
| Phenol ($C_6H_5OH$) | - | 0.494(0.480) | 0.341(0.170) |
| 1,3-Butadiene ($C_4H_6$) | 0.127(0.060) | 3.63E-3(4.51E-3) | 0.180(0.068) |
| Isoprene ($C_5H_8$) | - | 0.220(0.170) | 1.97E-2(1.57E-2) |
| Ammonia ($NH_3$) | 1.48(1.13) | 1.10(1.05) | 1.32(1.10) |
| Hydrogen Cyanide (HCN) | 0.134(0.252) | 0.381(0.259) | 0.630(0.463) |
| Nitrous Acid (HONO) | - | 0.395(0.221) | 0.377(0.084) |
| Sulfur Dioxide ($SO_2$) | - | 1.06(0.36) | 2.54(1.09) |
| Hydrogen Fluoride (HF) | - | - | bdl |
| Hydrogen chloride (HCl) | - | 0.472(0.320) | 2.65E-2(-) |
| Nitric Oxide (NO) | 1.73(0.66) | 1.44(0.42) | 1.72(0.93) |
| Nitrogen Dioxide ($NO_2$) | 2.92(1.77) | 1.65(0.47) | 0.630(0.203) |
| Ethane (C2H6) | 0.764(0.414) | - | 0.566(-) |
| Propane (C3H8) | 0.237(0.126) | - | 0.186(-) |
| 1-Butene ($C_4H_8$) | 0.113(0.050) | 0.134(0.100) | 0.119(0.007) |
| Benzene (C6H6) | - | 0.301(0.177) | 0.379(0.091) |
| Toluene (C7H8) | - | 0.296(0.228) | 0.224(0.041) |
| Ethylbenzene (C8H10) | - | - | 6.24E-2(4.05E-3) |
| m/p-Xylene ($C_8H_{10}$) | - | 0.107(0.088) | 0.297(0.319) |
| PM | 5.26(1.98) | - | - |
| EF Black Carbon (BC) | - | - | 0.831(0.497) |
| EF Brown Carbon (BrC) | - | - | 10.9(6.5) |
| EF $B_{abs}$ 405 ($m^2$/kg) | - | - | 19.2(8.0) |
| EF $B_{scat}$ 405 ($m^2$/kg) | - | - | 116(80) |
| EF $B_{abs}$ 870 ($m^2$/kg) | - | - | 3.94(2.36) |
| EF $B_{scat}$ 870 ($m^2$/kg) | - | - | 33.1(29.5) |
| EF $B_{abs}$ 405 just BrC ($m^2$/kg) | - | - | 10.7(6.3) |
| EF $B_{abs}$ 405 just BC ($m^2$/kg) | - | - | 8.47(5.06) |
| SSA 405 nm | - | - | 0.818(0.146) |
| SSA 870 nm | - | - | 0.825(0.082) |
| AAE | - | - | 2.15(0.79) |

[a]*Yokelson* et al. (2011) data are adjusted to a lower carbon fraction (0.42)

[b]NAMaSTE gas-phase EF values are adjusted to MCE 0.925 (see Sect. 3.7)

[c]Additional gas-phase compounds are in Table S9





Table 6. Emission factors (g/kg) for a single clamp kiln, zig-zag kiln, and stoke holes on the zig-zag kiln.

| Compound (Formula) | EF clamp kiln | EF zig - zag kiln | EF coal stoke holes at zig-zag kiln |
|---|---|---|---|
| Method | FTIR+WAS | FTIR+WAS | FTIR |
| MCE | 0.950 | 0.994 | 0.861 |
| Carbon Dioxide ($CO_2$) | 2102 | 2620 | 2234 |
| Carbon Monoxide (CO) | 70.9 | 10.1 | 230 |
| Methane ($CH_4$) | 19.5 | 8.73E-02 | 4.59 |
| Acetylene ($C_2H_2$) | 5.58E-02 | 1.65E-02 | 1.87E-02 |
| Ethylene ($C_2H_4$) | 1.27 | 4.32E-02 | 0.445 |
| Propylene ($C_3H_6$) | 1.49 | 6.58E-02 | 0.808 |
| Formaldehyde (HCHO) | 8.21E-02 | bdl | bdl |
| Methanol ($CH_3OH$) | 1.77 | 0.112 | 0.437 |
| Formic Acid (HCOOH) | 0.241 | 5.84E-02 | 0.180 |
| Acetic Acid ($CH_3COOH$) | 0.430 | 0.471 | 11.3 |
| Glycolaldehyde ($C_2H_4O_2$) | bdl | bdl | bdl |
| Furan ($C_4H_4O$) | 0.383 | bdl | bdl |
| Hydroxyacetone ($C_3H_6O_2$) | 1.81 | bdl | 1.61 |
| Phenol ($C_6H_5OH$) | 0.429 | 1.54E-02 | bdl |
| 1,3-Butadiene ($C_4H_6$) | 0.103 | 1.51E-02 | bdl |
| Isoprene ($C_5H_8$) | 8.66E-02 | 2.46E-02 | 1.47 |
| Ammonia ($NH_3$) | 0.317 | bdl | bdl |
| Hydrogen Cyanide (HCN) | 1.39 | 0.446 | 2.28 |
| Nitrous Acid (HONO) | 0.320 | 4.45E-02 | 1.33 |
| Sulfur Dioxide ($SO_2$) | 13.0 | 12.7 | 28.5 |
| Hydrogen Fluoride (HF) | bdl | 0.629 | 0.888 |
| Hydrogen chloride (HCl) | bdl | 1.24 | 1.86 |
| Nitric Oxide (NO) | bdl | 1.28 | 10.4 |
| Nitrogen Dioxide ($NO_2$) | 0.297 | 8.21E-02 | 1.36 |
| Carbonyl sulfide (OCS) | - | 3.42E-03 | nm |
| DMS ($C_2H_6S$) | - | 3.68E-05 | nm |
| Chloromethane ($CH_3Cl$) | - | 2.22E-02 | nm |
| Bromomethane ($CH_3Br$) | 2.62E-03 | 2.59E-03 | nm |
| Methyl iodide ($CH_3I$) | bdl | 2.01E-03 | nm |
| 1,2-Dichloroethene ($C_2H_2Cl_2$) | - | 4.45E-05 | nm |
| Methyl nitrate ($CH_3NO_3$) | 2.36E-05 | 2.92E-03 | nm |
| Ethane ($C_2H_6$) | 5.37 | 2.06E-03 | nm |
| Propane ($C_3H_8$) | 3.00 | 1.97E-03 | nm |
| i-Butane ($C_4H_{10}$) | 0.342 | 1.60E-03 | nm |
| n-Butane ($C_4H_{10}$) | 1.16 | 1.92E-03 | nm |
| 1-Butene ($C_4H_8$) | 0.347 | 1.68E-03 | nm |
| i-Butene ($C_4H_8$) | 0.428 | 1.47E-03 | nm |
| trans-2-Butene ($C_4H_8$) | 0.346 | 1.44E-03 | nm |
| cis-2-Butene ($C_4H_8$) | 0.214 | 9.65E-04 | nm |
| i-Pentane ($C_5H_{12}$) | 0.349 | 3.70E-02 | nm |
| n-Pentane ($C_5H_{12}$) | 0.811 | 3.26E-02 | nm |
| 1-Pentene ($C_5H_{10}$) | 0.233 | 1.60E-03 | nm |
| trans-2-Pentene ($C_5H_{10}$) | 0.249 | 2.64E-03 | nm |
| cis-2-Pentene ($C_5H_{10}$) | 0.093 | 9.01E-04 | nm |
| 3-Methyl-1-butene ($C_5H_{10}$) | 5.72E-02 | 3.32E-04 | nm |
| 1,2-Propadiene ($C_3H_4$) | 4.97E-04 | 2.15E-05 | nm |
| Propyne ($C_3H_4$) | 1.80E-03 | bdl | nm |
| 1-Butyne ($C_4H_6$) | bdl | bdl | nm |
| 2-Butyne ($C_4H_6$) | bdl | bdl | nm |
| n-Hexane ($C_6H_{14}$) | 0.670 | 2.16E-02 | nm |
| n-Heptane ($C_7H_{16}$) | 0.617 | 3.04E-03 | nm |
| n-Octane ($C_8H_{18}$) | 0.549 | 1.58E-03 | nm |
| n-Nonane ($C_9H_{20}$) | 0.434 | 2.42E-03 | nm |
| n-Decane ($C_{10}H_{22}$) | 0.428 | 2.02E-03 | nm |



| | | | |
|---|---|---|---|
| 2,3-Dimethylbutane ($C_6H_{14}$) | 0.127 | 3.59E-03 | nm |
| 2-Methylpentane ($C_6H_{14}$) | 0.398 | 4.84E-03 | nm |
| 3-Methylpentane ($C_6H_{14}$) | 0.312 | 1.17E-02 | nm |
| 2,2,4-Trimethylpentane ($C_8H_{18}$) | bdl | 8.02E-04 | nm |
| Cyclopentane ($C_5H_{10}$) | 0.134 | 8.53E-04 | nm |
| Cyclohexane ($C_6H_{12}$) | 5.55E-02 | 2.98E-03 | nm |
| Methylcyclohexane ($C_7H_{14}$) | 5.84E-02 | bdl | nm |
| Benzene ($C_6H_6$) | 1.68 | 8.25E-03 | nm |
| Toluene ($C_7H_8$) | 1.05 | 2.80E-02 | nm |
| Ethylbenzene ($C_8H_{10}$) | 0.279 | 1.35E-02 | nm |
| m/p-Xylene ($C_8H_{10}$) | 1.06 | 5.74E-02 | nm |
| o-Xylene ($C_8H_{10}$) | 0.377 | 2.18E-02 | nm |
| Styrene ($C_8H_8$) | 2.62E-03 | 4.56E-03 | nm |
| i-Propylbenzene ($C_9H_{12}$) | 2.84E-02 | 4.07E-04 | nm |
| n-Propylbenzene ($C_9H_{12}$) | 3.82E-02 | 1.82E-03 | nm |
| 3-Ethyltoluene ($C_9H_{12}$) | 0.091 | 6.93E-03 | nm |
| 4-Ethyltoluene ($C_9H_{12}$) | 3.55E-02 | 3.69E-03 | nm |
| 2-Ethyltoluene ($C_9H_{12}$) | 2.76E-02 | 2.30E-03 | nm |
| 1,3,5-Trimethylbenzene ($C_9H_{12}$) | 5.88E-02 | 4.30E-03 | nm |
| 1,2,4-Trimethylbenzene ($C_9H_{12}$) | 8.46E-02 | 5.59E-03 | nm |
| 1,2,3-Trimethylbenzene ($C_9H_{12}$) | 2.76E-02 | 2.03E-03 | nm |
| alpha-Pinene ($C_{10}H_{16}$) | bdl | 1.49E-03 | nm |
| beta-Pinene ($C_{10}H_{16}$) | bdl | 1.31E-03 | nm |
| Ethanol ($C_2H_6O$) | - | 4.84E-03 | nm |
| Acetaldehyde ($C_2H_4O$) | 4.13E-02 | 6.94E-02 | nm |
| Acetone ($C_3H_6O$) | - | 1.46E-01 | nm |
| Butanal ($C_4H_8O$) | bdl | 2.19E-03 | nm |
| Butanone ($C_4H_8O$) | - | 2.29E-03 | nm |
| EF Black Carbon (BC) | 1.72E-2(7.50E-3) | 0.112(0.063) | nm |
| EF Brown Carbon (BrC) | 1.74(0.34) | 0.913(0.278) | nm |
| EF $B_{abs}$ 405 ($m^2$/kg) | 1.86(0.24) | 2.03(0.70) | nm |
| EF $B_{scat}$ 405 ($m^2$/kg) | 32.8(2.1) | 21.2(12.8) | nm |
| EF $B_{abs}$ 870 ($m^2$/kg) | 8.16E-2(3.56E-2) | 0.530(0.300) | nm |
| EF $B_{scat}$ 870 ($m^2$/kg) | 0.670(0.129) | 1.75(0.25) | nm |
| EF $B_{abs}$ 405 just BrC ($m^2$/kg) | 1.70(0.33) | 0.895(0.273) | nm |
| EF $B_{abs}$ 405 just BC ($m^2$/kg) | 0.155(0.102) | 1.14(0.64) | nm |
| SSA 405 nm | 0.946(0.007) | 0.881(0.098) | nm |
| SSA 870 nm | 0.895(0.029) | 0.779(0.103) | nm |
| AAE | 4.19(0.73) | 1.92(0.50) | nm |

Note: "bdl" indicates below the detection limit; "-" indicates concentrations were not greater than background; "nm" indicates not measured; C-fractions: zigzag kiln (0.722), clamp kiln (0.644) (see Sect. 2.4)





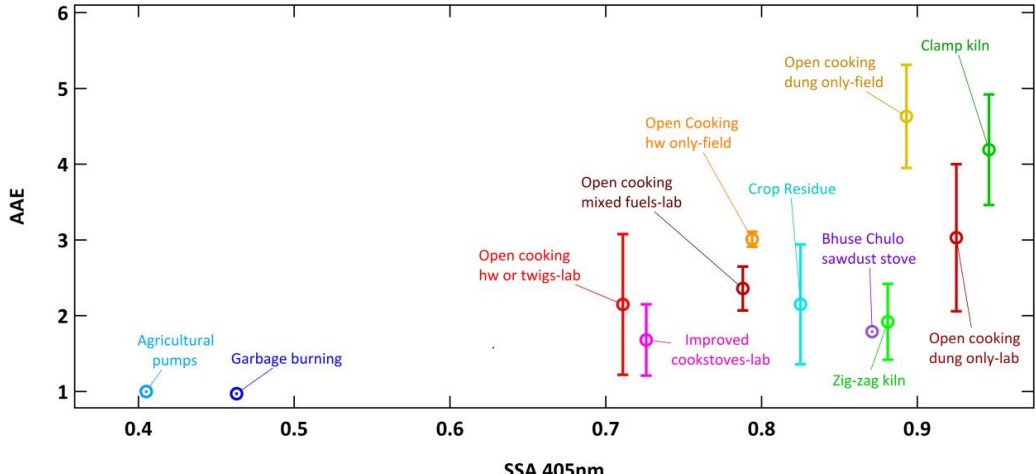

**Figure 1.** The absorption Ångström exponent (AAE) calculated at 405 and 870 nm as a function of single scattering albedo (SSA) at 405 nm for fuel types measured during the NAMaSTE campaign. The error bars represent ±1 standard deviation of AAE measured for different burns (or different samples as is the case for brick kilns). Note: "hw" indicates hardwood fuels.



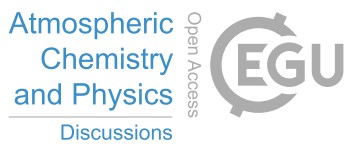

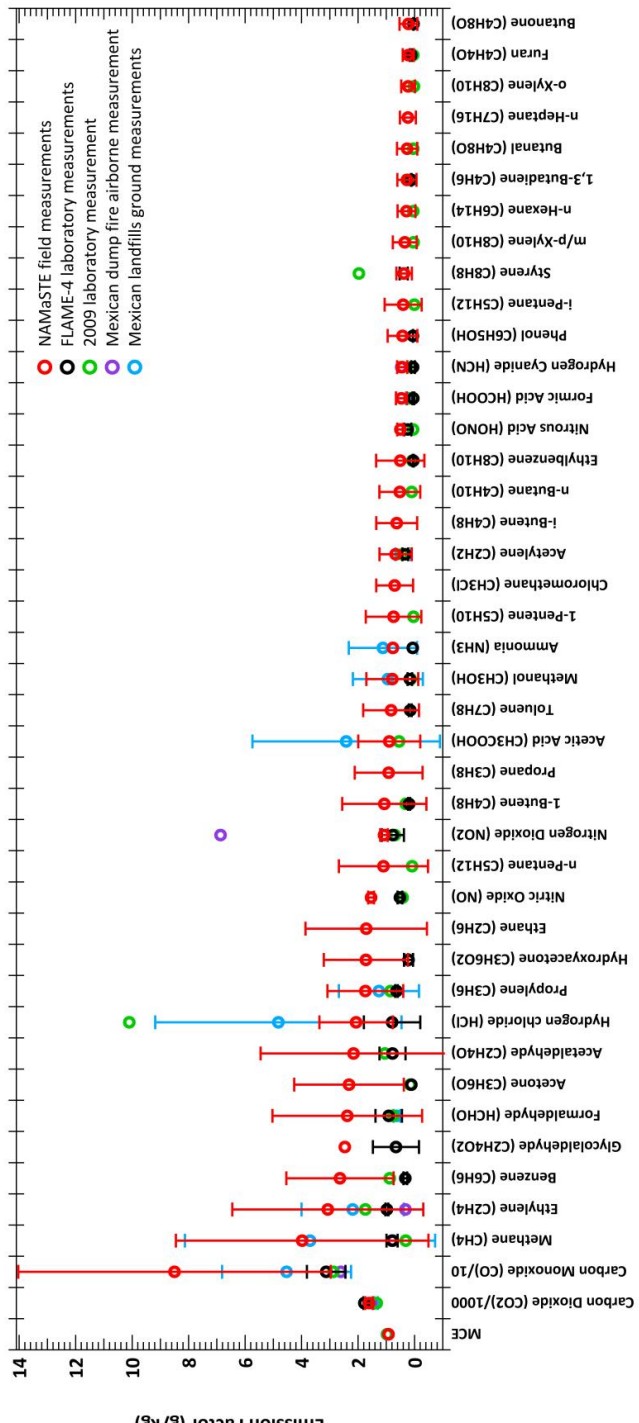

**Figure 2.** Garbage burning emission factors (g/kg) compiled for laboratory measurements (Yokelson et al., 2013; Stockwell et al., 2015) (green, black), field measurements of open burning in Mexican landfills (Christian et al., 2010) (blue), a single airborne measurement from a Mexican dump fire (Yokelson et al., 2011) (purple) and our current study of mixed garbage (red). Error bars indicate one standard deviation of the EF for each study where available.



**Figure 3.** The modified combustion efficiency (MCE) shown in descending order for each cookstove/fuel combination measured in this study. The stove-type is listed followed by the main fuel constituents and an indication whether the source was a lab or field measurement. Note: "hw" indicates hardwood fuels; "d" indicates dung; "cc" indicates charcoal; "t" indicates twigs; and "sd" indicates sawdust.





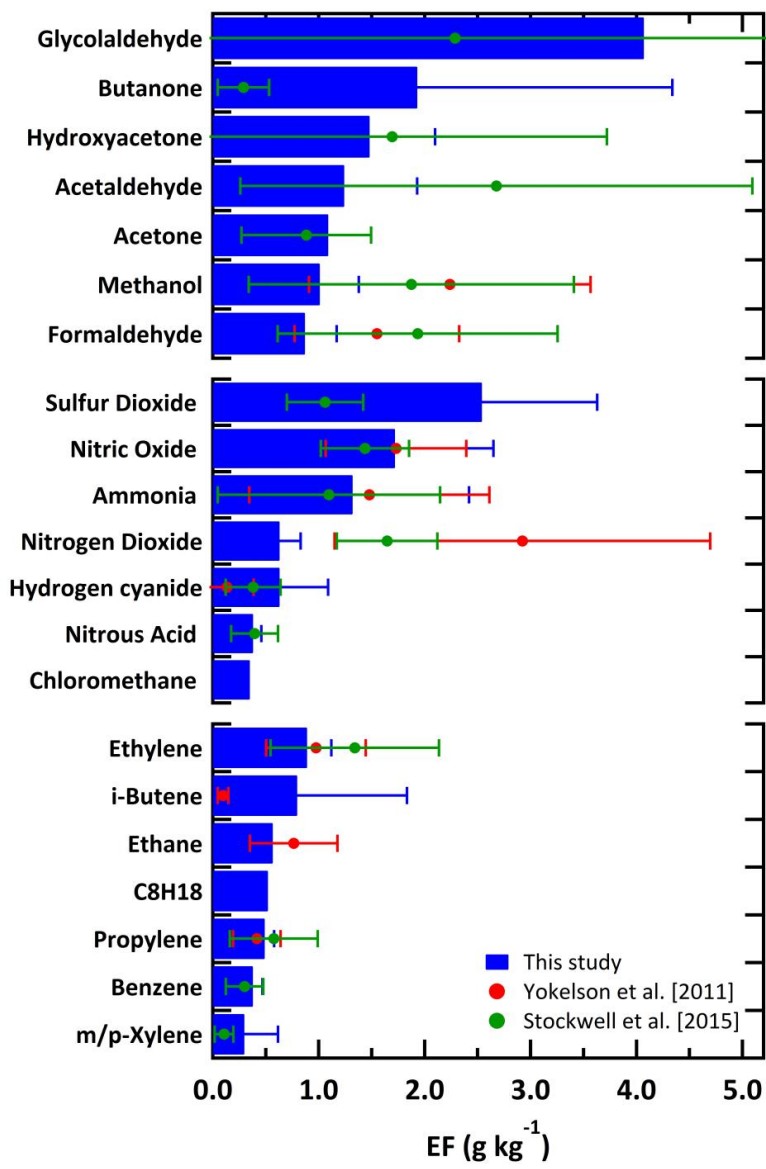

**Figure 4.** The emission factors (g/kg) and ± one standard deviation for the most abundant OVOCs, NMHCs, and S-/N- containing compounds emitted from crop residue burns. The crop residue fires from other studies (Yokelson et al., 2011; Stockwell et al., 2015) are shown in red and green.



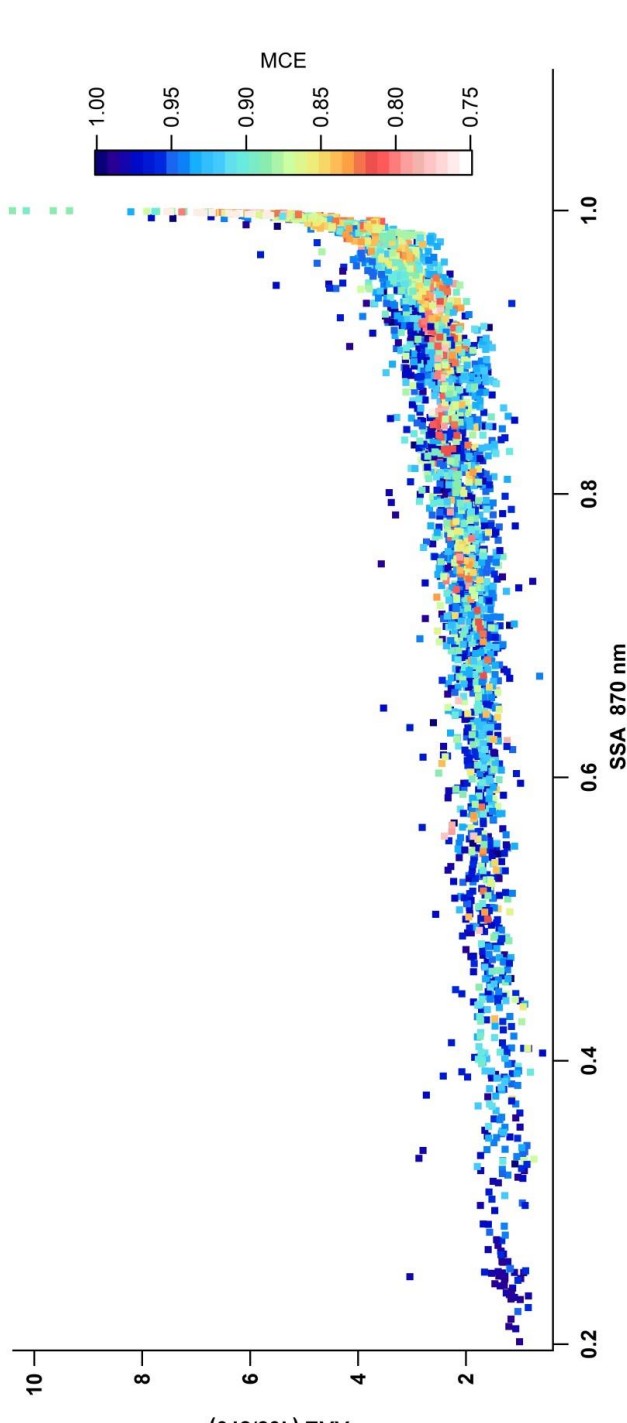

**Figure 5.** The AAE calculated at 405 and 870 nm versus SSA at 870 nm for all crop residue burn samples measured every second during emissions collection. Each data point is colored by MCE. The AAE increases sharply at high SSA, while the MCE distinctly decreases at increasing SSA. BC emissions are associated mostly with high MCE flaming and BrC emissions are associated mostly with low MCE smoldering. Most source-types demonstrated a similar trend.



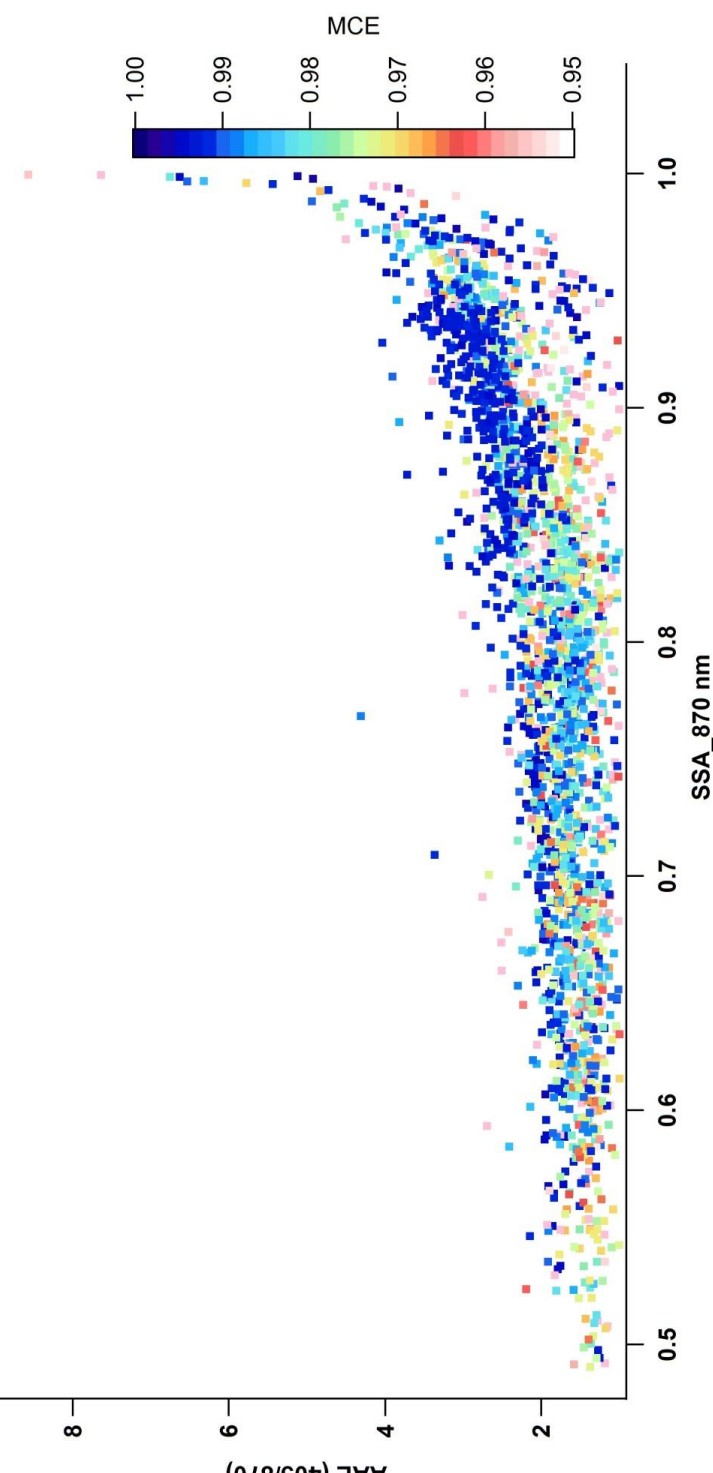

**Figure 6.** The AAE calculated at 405 and 870 nm versus SSA at 870 nm for the zig-zag kiln measured every second during emissions collection. Each data point is colored by MCE. This deviates from the typical trend in that the highest MCEs are not clustered at the lowest SSA/AAEs. Some BrC is emitted at a variety of "higher" MCEs.