# Peer review of "Nepal Ambient Monitoring and Source Testing Experiment (NAMaSTE): Emissions of trace gases and light-absorbing carbon from wood and dung cooking fires, garbage and crop residue burning, brick kilns, and other sources"

_Atmospheric Chemistry and Physics, 2016_

## Short Comment (SC1) · 22 Apr 2016

Thank you very much for this useful work. In the brick kiln section (2.1.7), you describe a technique to measure at kilns with shorter stacks and some of the "real" emissions benefits of such measurements. I would suggest that you also add language that makes it even clearer that such a technique could not be used for taller stacks, which

are common on zig-zag kilns. You may want to include some statement about how there are some instruments and protocols, at least for aerosols and some (but fewer) gases, for measuring at the port of the taller stacks.

---

## Author Comment (AC1) · 18 May 2016

Thank you for the comment highlighting an important issue. Brick kilns can have large air quality impacts and emissions testing could help mitigate those effects. Port sampling is the usual emissions testing approach for regulatory purposes, but high temperatures and high concentrations in stacks lead to a host of sampling issues and

uncertainty in the actual atmospheric impacts. These issues are dynamically corrected shortly after emission by dilution and cooling. Thus, post emission sampling is more atmospherically relevant. However, comparing the results between stacks that were sampled through ports should provide useful guidance. We were aware of the difficulties others had experienced in measuring low flow rates in brick kiln stacks (with affordable flow meters) and the general need for a state-of-the-art dilution system for port sampling. We brought the expensive equipment to Nepal that is needed to implement port sampling thinking that port sampling would be our only option. Fortunately, we had the opportunity to measure the real emissions from shorter stacks. However, we only had time to implement this one approach due in part to the earthquake. While measuring the emissions after they exit the top of the stack is not impossible for taller kilns, it would require repeated costs in the form of e.g. scaffolding. On the other hand, port sampling requires an upfront investment in a heated dilution system and expensive flow meter. Ideally, the results and cost of both approaches could be compared on a tall stack and a preferred method could be selected taking into account the amount of sampling needed and the science or regulatory goals. However, this is outside the scope of this work. Certainly, emissions testing by either approach on brick kilns has value. We will modify the text to clarify that port sampling approaches also have benefits and gladly cite any recommended protocol.

---

## Referee Comment (RC1) · Anonymous Referee #1 · 24 May 2016

This paper presents a wealth of information on emissions of a wide range of gas phase species and particulate from most of the important sources in the Indo-Gangetic plain region of Nepal. These are important measurements and greatly extend the information on many of the emissions from a number of sources. I therefore recommend publication of the paper in ACP.

[Figure]

The paper is very long. This is, to a large extent necessary as the authors present a large amount of information and provide a significant amount of important and detailed discussion. In addition, the supplementary material contains all the basic data used to produce the average data presented in the main paper and a methodology of how the averages have been developed. The authors should be applauded for this.

However, the data available from the NAMASTE sampling varies in extent and detail quite considerably from one source to another. Clearly the earthquake also had a disruptive effect on the sampling. This, in itself is not a problem, some of the sources are sampled in great detail and make significant advances on what is currently available (such as the field sampling of cook stoves) and may be used to develop emissions inventories across the region. In other cases only a couple of measurements are made (e.g. brick kilns), whilst these may be important in that they identify whether or not the source may be important they are more preliminary in nature. Neither of these is a problem and I welcome seeing all the results, however, as presented the long discussion of data from different sources has the effect of burying some of the major implications in a forest of less important results. I suggest that to gain maximum impact from the paper the authors consider a more informative summary in the conclusions section that compares their data to present literature, highlights where the authors recommend changes to current thinking about emissions inventories and the estimated size of any effect of such consideration (eg crop residue burning), and points out where further work is needed and how their data shows what this entails (eg brick kilns and garbage burns). The authors may also wish to amend the abstract in a similar fashion. I would like to see the importance of any recommendations quantified in the abstract and conclusion and not only in the main body of text.

The paper reports NOx emission ratios for a wide range of sources. However, despite the FTIR measurements making independent determinations of NO and NO2, there is no real discussion of their relevant contribution to primary NOx at all. Given the toxicity of NO2 and the widespread regulatory framework for NO2 that exists this is somewhat

surprising since the data could be used to inform air quality management decisions in this area. I would very much recommend that the NO2/NOx is discussed for each of the sources wherever possible.

The paper reports high relative emissions of NOx and absorbing aerosol from diesel generators and diesel agricultural pumps (page 18 line 1) and these figures should be included in revised emission inventories for the region. However, there are very few measurements of these sources in this study, only 2 diesel generators were studied (and only one of these sampled for NOx by the FTIR) and two irrigation diesel pumps were studied. How confident are the authors that these samples are representative enough of these emission classes across the region to argue that revised emission inventories should include these sources?

Page 8 line 33: It is incorrect to say that SSA and the absorption Angstrom exponent were measured. The measured properties from the PAX are the absorption and scattering coefficients. The SSA and absorption Angstrom exponent are derived.

Page 10, lines 33-36: Were the ERs derived from the WAS and the FTIR for the same species within uncertainty and if not why was the FTIR data chosen, was this simply on S/N, could there have been biases?

Page 11, line 6: You need to say that n is the total number of measured species.

Page 11, lines 11-14: How are FTIR, WAS and filter data combined into the denominator of equation 1? I can imagine that different averaging times could lead to sampling very different source strengths and hence biasing the denominator relative to the numerator for a given species.

Page 11, line 20: are all the CO and CO2 data sufficiently above the background in all the samples? If not taking the average of the ratios will introduce uncertainty (and possibly bias) and it may be better to take the average of deltaCO and delta CO2 data and then calculate a ratio.

Page 12 line 4: The MAC used to calculate BC is an average value over a wide range of source types. There is some variation of MAC between different sources of BC, particularly some of those in this paper. The authors should comment on this uncertainty and variability.

Page 12 section 2.4: There is considerable uncertainty in the emission factors for the mixed fuels. The authors provide a detailed summary of their assumption but don't provide an estimate of the uncertainty in their estimate. This would be good to see.

Page 13 line 16: As before, SSA is derived, not measured.

Page 13, lines 19-20: "the shape of the absorption cross section". Better to write "the wavelength dependence of the absorption cross section"

Page 13 lines 21-22: The bars in figure 1 seem to represent the variations in AAE seen for different burns of the same type. There will also be some burn to burn variation in AAE as well, it would be good to show this also.

Page 13, line 23: should read "upper right hand corner"

Page 13 line 26: I am not sure I understand what is said here correctly. If there were no PAX measurements at 870 nm for the agricultural pumps and all but one of the garbage burns, how were the AAEs in figure calculated? I can understand that based on the SSA they can be assumed to be almost pure BC and hence an appropriate MAC is chosen to derive a BC mass but unless an AAE measurement derived value is possible they should not be included in figure 1.

Page 14 section 3.2: I am somewhat surprised that NO2 is below detection limit. However, the detection limits don't appear to be available anywhere in the paper to know whether the NO2/NOx ratios are simply too low to be measured (<2%) or are abnormally low for the vehicle type.

Page 16 line23-24: it isn't clear why this isn't provided in the earlier section

Page 20 line 8-10: It is stated that the scaling of lab emission factors is carried out according to Stockwell et al (2014) but this uses emission ratios to CO for smouldering and not methane. Furthermore, the full discussion of the method is not really presented in detailed in Stockwell et al (2014) but is in Yokelson et al (2008). It might be good to cite this paper directly.

Page 26 line 18: define the subscripts CK and ZZ and use elsewhere, otherwise write out what is meant.

---

## Referee Comment (RC2) · Anonymous Referee #2 · 12 Jul 2016

Excellent work and well presented.

Recommended for publication in ACP.

---

## Referee Comment (RC3) · Anonymous Referee #3 · 29 Jul 2016

Summary: This paper describes the measurements of trace gases and carbonaceous (BC, and BrC) aerosols (mass and optical properties) from a variety of bio and fossil fuel burning sources in an around Kathmandu valley of Nepal in the eastern Himalayas. These measurements formed part of a field campaign using a mobile laboratory and a set of state-of-the-art instruments. These measurements provide important and valuable data on light absorbing aerosols and trace gases over this part of the Himalayas,

and is done so extensively perhaps for the first time and thus are, useful for environmental and climate impact assessment studies. Having said the above, the data should also be taken with the pinch of salt that the measurements have been for a very short duration (about a fortnight), the number of samples used to evolve the means are statistically not very high. How well these sample represent the regional population of these sources remains to be seen. In other words, the data emanates from a highly under-sampled source. Of course, the authors also seem to be aware of this. Moreover, the reported values correspond only to a particular period of time of the year (not even a full month). They also do not represent the strong seasonal variations in some of these sources and also changes caused by changes in synoptic meteorology and long-range transport; which are very important over this region. Their spatial representativeness also is highly limited to the urban region of Nepal and its southern slopes, which would be impact by transport of emission from the adjoining north Indian plains. Nonetheless, it remains that these are the first comprehensive set of measurements from several sources that are specific to this region (on which field measurements are not possible in other parts of the world), and provide information that could be used as inputs for regional climate and environment impact assessment and probably would lead to more such measurements. In other words, they provide some ball-park figure on these parameters, which were otherwise not available. In view of the above, the paper may be accepted for publication in ACP, provided the authors take care of the following specific concerns in their revised version 1. Make a clear statement of the limitation of the data (under sampling in time, space and sources) explicitly in the abstract and conclusions 2. Provide the fractional figures (to the extent accurate) of the sources sampled, against the total number of such sources present in Kathmandu (for eg % ge of brick kilns sampled vs the total no. available in Kathmandu, the no. of two-wheelers sampled vs the total two-wheelers plying in the city, the no. of gensets samples vs the total etc in a table or in the respective sub-sections. 3. For FTIR – please provide the actual averaging time used (for grab sampling), the best and worst S/N for the species being reported in this study and the resulting uncertainty in the derived concentrations.

4. In the ER calculation, state explicitly how the average ER estimated from the FTIR around a given source, compared with the values obtained from WAS data, for completeness of information and also in view that WAS value represents the 'bulk'. 5. It is highly appreciable the way authors have described all the assumptions used in their estimates. Yet, it would be better if they can give an upper and lower bound for the estimates due to the specific assumption they have made (like 50% wood and dunk in mixed fuels – suppose it is 80-20 or 30-70 )

---

## Author Comment (AC2) · 22 Aug 2016

We thank Referee #1 for their constructive comments, which will improve the paper. We reproduce the exact comments, followed by our response and include exact changes to be made in the text. At the end we note a few additional voluntary improvements.

Anonymous Referee #1

This paper presents a wealth of information on emissions of a wide range of gas phase species and particulate from most of the important sources in the Indo-Gangetic plain region of Nepal. These are important measurements and greatly extend the information on many of the emissions from a number of sources. I therefore recommend publication of the paper in ACP.

The paper is very long. This is, to a large extent necessary as the authors present a large amount of information and provide a significant amount of important and detailed discussion. In addition, the supplementary material contains all the basic data used to produce the average data presented in the main paper and a methodology of how the averages have been developed. The authors should be applauded for this.

However, the data available from the NAMASTE sampling varies in extent and detail quite considerably from one source to another. Clearly the earthquake also had a disruptive effect on the sampling. This, in itself is not a problem, some of the sources are sampled in great detail and make significant advances on what is currently available (such as the field sampling of cook stoves) and may be used to develop emissions inventories across the region. In other cases only a couple of measurements are made (e.g. brick kilns), whilst these may be important in that they identify whether or not the source may be important they are more preliminary in nature. Neither of these is a problem and I welcome seeing all the results, however, as presented the long discussion of data from different sources has the effect of burying some of the major implications in a forest of less important results. I suggest that to gain maximum impact from the paper the authors consider a more informative summary in the conclusions section that compares their data to present literature, highlights where the authors recommend changes to current thinking about emissions inventories and the estimated size of any effect of such consideration (eg crop residue burning), and points out where further work is needed and how their data shows what this entails (eg brick kilns and garbage burns). The authors may also wish to amend the abstract in a similar fashion. I would like to see the importance of any recommendations quantified in the abstract

and conclusion and not only in the main body of text.

**Authors:** The idea to assess the overall significance of the results is excellent, but we feel perhaps a bit premature mostly because this paper is already long and the first in a series. Two more source sampling papers that expand the coverage of the sources are in preparation and some follow-up measurements have been initiated. For instance, as noted in the text, the paper on the filter-based results has more complete sampling of garbage burning. We also have ambient data from the super-site and plan model evaluation of all the data. In addition, the assessment is less straightforward (and would be more involved) than might appear initially. E.g., we sampled more cooking fires than irrigation pumps, but the former probably employs a wider variety of fuels and devices. Thus, we prefer a stepwise approach where we focus on presenting the data initially and then summarize and synthesize it more in upcoming publications that will have the advantage of more complete information. However, the Referee's point is well taken and it makes sense to try to clarify better in the abstract and conclusions that some sources are more prolific and/or probably better sampled than others. For instance, for cooking fires and crop residue burning, which are major South Asian and global sources, it's important that BrC is significant in addition to BC, and this is even more pronounced for dung compared to wood. It's also the first extensive suite of gases for most of the sources. On the other hand, though we have begun to address these undersampled sources, given the diversity of the sources, much more work is needed, especially for gensets, pumps, traffic, and brick kilns.

To add this emphasis we make the following changes:

At the end of the abstract (Page (P) 2, Line (L) 14): We added: "These measurements begin to address the critical data gap for these important, undersampled sources, but due to their diversity and abundance, more work is needed."

At the end of the introduction (P4, L7) changed: "Taken together, the NAMaSTE efforts reduce the information gap for these important undersampled sources." to "Taken to-

gether, the NAMaSTE efforts begin to reduce the information gap for these important undersampled sources."

Near the end of the conclusions (P28, L19): We added: "In summary, we have provided the first extensive suite of gases for most of the sources. For cooking fires and crop residue burning, which are major South Asian and global sources, we have shown that absorption by both BrC and BC is significant with BrC absorption even more pronounced for dung fuel compared to wood fuel. On the other hand, though we have begun to address these undersampled sources, given the diversity and abundance of the sources, much more work is needed, especially for gensets, pumps, traffic, and brick kilns."

**R1:** The paper reports NOx emission ratios for a wide range of sources. However, despite the FTIR measurements making independent determinations of NO and NO2, there is no real discussion of their relevant contribution to primary NOx at all. Given the toxicity of NO2 and the widespread regulatory framework for NO2 that exists this is somewhat surprising since the data could be used to inform air quality management decisions in this area. I would very much recommend that the NO2/NOx is discussed for each of the sources wherever possible.

**Authors:** For most combustion sources NO is the primary emission (typically about 80% of $NO_x$ immediately after emission), but NO is converted rapidly to $NO_2$ on a fast time scale after emission (within 5-10 minutes) via titration mainly by $O_3$ and also $HO_2$. In addition, $NO_2$ is then converted to PAN and nitrate within an hour or two so exposure to $NO_2$ is a sensitive function of time and dilution rates since emission and our exact $NO_2$ levels in plumes only apply over a narrow range of distance from our specific sources. We have now added this explanatory sentence with references on page 16 line 29, right after we first mention high $NO_x$ emissions from a source.

Change: Added at P16, L29: "NO is the main form of fresh combustion $NO_x$, but it is converted to $NO_2$ within minutes and PAN and nitrate within a few hours (affecting

aerosol and $O_3$ levels) as discussed elsewhere (Akagi et al., 2013; Liu et al., 2016)."
(Liu et al reference was updated.)

**R1:** The paper reports high relative emissions of NOx and absorbing aerosol from
diesel generators and diesel agricultural pumps (page 18 line 1) and these figures
should be included in revised emission inventories for the region. However, there are
very few measurements of these sources in this study, only 2 diesel generators were
studied (and only one of these sampled for NOx by the FTIR) and two irrigation diesel
pumps were studied. How confident are the authors that these samples are repre-
sentative enough of these emission classes across the region to argue that revised
emission inventories should include these sources?

**Authors:** Referee #3 also noted the problem of sufficient sampling of complex, variable
sources. As with many pioneering, but necessarily limited measurements we can only
get the process started and hope that our data are an improvement over no data and
we hope future measurements will contribute to an evolution and improvement of these
first values. This is related to the above suggestion we addressed by listing the sources
where more sampling is most needed in the conclusions.

**R1:** Page 8 line 33: It is incorrect to say that SSA and the absorption Angstrom expo-
nent were measured. The measured properties from the PAX are the absorption and
scattering coefficients. The SSA and absorption Angstrom exponent are derived.

**Authors:** On page 9 L33-we have revised the text to say that SSA and AAE were
calculated from the direct measurements of absorption and scattering.

Change: "Particle absorption and scattering coefficients (Babs, Bscat), single scatter-
ing albedo (SSA), and absorption Ångström exponent (AAE) at 405 nm and 870 nm
were measured directly at 1 s time resolution using two photoacoustic extinctiometers
(PAX, Droplet Measurement Technologies, Inc., CO)."

To: "Particle absorption and scattering coefficients (Babs, Bscat) at 405 nm and 870 nm

were measured directly at 1 s time resolution using two photoacoustic extinctiometers (PAX, Droplet Measurement Technologies, Inc., CO) and the single scattering albedo (SSA) and absorption Ångström exponent (AAE) were calculated from these measurements."

This is further clarified on page 11, lines 25-27.

**R1:** Page 10, lines 33-36: Were the ERs derived from the WAS and the FTIR for the same species within uncertainty and if not why was the FTIR data chosen, was this simply on S/N, could there have been biases?

**Authors:** The WAS grab samples were not collected at the same exact time and location as the FTIR samples so any difference could be natural variability or uncertainty in either instrument with no way to know for certain the contribution of each. The vast majority of trace gases were measured by one or the other technique and not both. In the handful of cases where both techniques measured the same gas on the same source we used the FTIR data only, mainly because there was a much larger number of FTIR grab samples (5-30 by FTIR as opposed to one by WAS). We assumed that the time required to blend the WAS results into a weighted average would have little effect on a properly weighted average.

**R1:** Page 11, line 6: You need to say that n is the total number of measured species.

**Authors:** Changed: "where FC is the measured carbon mass fraction of the fuel; MMx is the molar mass of species X; AMC is the atomic mass of carbon (12 g mol-1); NCj is the number of carbon atoms in species j; $\Delta$Cj or $\Delta$X referenced to $\Delta$CO are the source-average molar emission ratios for the respective species."

To "where FC is the measured carbon mass fraction of the fuel; MMx is the molar mass of species X; AMC is the atomic mass of carbon (12 g mol-1); NCj is the number of carbon atoms in species j; n is the total number of measured species; $\Delta$Cj or $\Delta$X referenced to $\Delta$CO are the source-average molar emission ratios for the respective

species."

**R1:** Page 11, lines 11-14: How are FTIR, WAS and filter data combined into the denominator of equation 1? I can imagine that different averaging times could lead to sampling very different source strengths and hence biasing the denominator relative to the numerator for a given species.

**Authors:** The FTIR and WAS grab samples and the integrated filter data were all ratioed separately to the CO measured over the exact same time period as the indicated technique was employed. This serves to normalize all the data to a common reference and also to extend the amount of time that each source is sampled. Even though there are different sampling times this should not impart bias, but instead greater representativeness. For example, the filter data were collected as the FTIR continuously measured CO and ratioed to that CO. This occurred in between the grab samples used for CO and other gases.

**R1:** Page 11, line 20: are all the CO and CO2 data sufficiently above the background in all the samples? If not taking the average of the ratios will introduce uncertainty (and possibly bias) and it may be better to take the average of deltaCO and delta CO2 data and then calculate a ratio.

**Authors:** All the CO and $CO_2$ data were well-resolved from background since the background levels are way above the detection limit for these species. In addition, we used linear regressions of $\Delta CO$ vs $\Delta CO_2$ to obtain the CO/$CO_2$ ERs that are already weighted towards the larger excess values as detailed on P10, L26-33. The cited paper explores/justifies this in great detail.

**R1:** Page 12 line 4: The MAC used to calculate BC is an average value over a wide range of source types. There is some variation of MAC between different sources of BC, particularly some of those in this paper. The authors should comment on this uncertainty and variability.

**Authors:** We used MACs for both BC and BrC that are recommended "average" values as specified in the text, but the results are easily scaled if a user feels a different MAC is preferable for one or more sources. It's not well known what the total uncertainty in the MACs are due to rapid evolution in the field of absorbing aerosol, but the coefficient of variation recommended by the review that developed our BC MAC is 16% at 550 nm for fresh uncoated combustion aerosol (Bond and Bergstrom, 2006). However, some fresh BC may have some coating and the assumption of an AAE of one to calculate the MAC at 870 nm is not exact. We can guess that ∼25-30% is a reasonable "typical" uncertainty with the error being asymmetric in that we are more likely to overestimate BC mass due to coating-induced MAC increases. We have revised the text to clarify this uncertainty. Clearly, more work is needed in this area and is planned for upcoming experiments. The MAC of bulk OA also varies substantially and the BrC mass we calculate with the single average MAC we used is only qualitatively similar to bulk OA mass with ∼45% uncertainty for "average" biomass burning aerosol (Lack and Langridge 2013) and more uncertainty for non-average aerosol (Saleh et al., 2014). These ideas should be clear in the text, but more importantly, this is why we report EF Babs and EF Bscat as they are directly measured using the PAX. In the case where only a mass emission is reported a user has to calculate the absorption and scattering with uncertain MAC/MSC values while also retaining any systematic error in the mass measurement, though we note that mass measured by a PAX can always be converted back to absorption (using the same MAC) without adding error. Also, note the AAE and SSA are not impacted by MAC variability.

Starting at P12, L3, the new text reads (in context):

"We directly measured aerosol absorption ($B_{abs}$, $Mm^{-1}$) and used the manufacturer-recommended mass absorption coefficient (MAC) (4.74 $m^2$/g at 870 nm) to calculate the BC concentration ($\mu g/m^3$). Our BC mass values are easily scaled if a user feels a different MAC is preferable for one or more sources. The total uncertainty in the MAC is not well known, but the coefficient of variation recommended by the review our

BC MAC is based on is 16% at 550 nm for fresh, uncoated combustion aerosol (Bond and Bergstrom, 2006). However, some fresh BC may have some coating and the assumption of an AAE of one to calculate the MAC at 870 nm is not exact. $\sim$25-30% is probably a reasonable "typical" uncertainty with the error being asymmetric in that we are more likely to overestimate BC mass due to coating-induced MAC increases. To a good approximation, sp2-hybridized carbon has an AAE of $1.0 \pm 0.2$ and absorbs light proportional to frequency. Thus, $B_{abs}$ due only to BC at 405 nm would be expected to equal $2.148 \times B_{abs}$ at 870 nm. This assumes any coating effects are similar at both wavelengths and has other assumptions considered reasonably valid, especially in biomass burning plumes by Lack and Langridge (2013). Following these authors, we assumed that excess absorption at 405 nm, above the projected amount, is associated with BrC absorption and the BrC ($\mu$g/m$^3$) concentration was calculated using a literature-recommended brown carbon MAC of $0.98 \pm 0.45$ m$^2$/g at 404 nm (Lack and Langridge, 2013). The BrC mass calculated this way is considered roughly equivalent to the total organic aerosol (OA) mass, which as a whole weakly absorbs UV light, and not the mass of the actual chromophores. The MAC of bulk OA varies substantially and the BrC mass we calculate with the single average MAC we used is only qualitatively similar to bulk OA mass for "average" aerosol and even less similar to bulk OA for non-average aerosol (Saleh et al., 2014). The BrC mass estimated by PAX in this way was independently sampled and worth reporting, but the filters and mini-AMS provide additional samples of the mass of organic aerosol emissions that have lower per-sample uncertainty for mass. Most importantly, the optical properties from the PAX (SSA, AAE, and absorption EFs calculated as detailed below) are not impacted by MAC variability or filter artifacts. In the case where only a mass emission is reported a user has to calculate the absorption and scattering with uncertain MAC/MSC values while also retaining any systematic error in the mass measurement, though we note that mass measured by a PAX can always be converted back to absorption (using the same MAC) without adding error. As mentioned above, the PAXs were run in series or in parallel with a $CO_2$ monitor. The mass ratio of BC and BrC to the simultaneous

co-located $CO_2$, measured by either the Picarro or LI-COR, was multiplied by the FTIR-WAS grab sample EF for $CO_2$ to determine mass EFs for BC and BrC in g/kg. From the measured ratios of $B_{abs}$ and $B_{scat}$ to $CO_2$, the EFs for scattering and absorption at 870 and 405 nm (EF $B_{abs}$, EF $B_{scat}$) were calculated and reported in units of $m^2$ emitted per kg of dry fuel burned. We reiterate that the absorption and scattering EFs do not depend on assumptions about the AAE of BC or MAC values."

**R1:** Page 12 section 2.4: There is considerable uncertainty in the emission factors for the mixed fuels. The authors provide a detailed summary of their assumption but don't provide an estimate of the uncertainty in their estimate. This would be good to see.

**Authors:** The emissions data are not that uncertain, but which fuel gave rise to the emissions is uncertain in the cases we singled out, as is the real carbon fraction. Since EFs are proportional to assumed %C it is a simple task to adjust the EF to reflect different %C that would arise from different mixes. Thus, the total range could be given by the possible mixtures. However, the pure fuels have variable %C in practice and we had no way of measuring the real mix of fuels that was burning at the instant of our grab samples, or the range of mixtures commonly used in practice. Thus, we took a reasonable guess, but did not have enough information for a quantitative estimate of uncertainty. We do now provide more carbon data in the text (see response to Referee #3) and remind the reader how to adjust mixed fuel EF for different overall carbon fractions.

P12, L39: Added: "For mixtures differing from those we used, the EF scale with the assumed carbon fraction."

**R1:** Page 13 line 16: As before, SSA is derived, not measured.

**Authors:** Changed: "As mentioned above, we measured absorption and scattering coefficients as well as single scattering albedo directly at 405 and 870 nm."

To: "As mentioned above, we measured absorption and scattering coefficients directly

and calculated single scattering albedo at 405 and 870 nm."

**R1:** Page 13, lines 19-20: "the shape of the absorption cross section". Better to write "the wavelength dependence of the absorption cross section"

**Authors:** changed.

**R1:** Page 13 lines 21-22: The bars in figure 1 seem to represent the variations in AAE seen for different burns of the same type. There will also be some burn to burn variation in AAE as well, it would be good to show this also.

**Authors:** We don't understand the comment. The burn to burn variation for each source type sampled more than once is what is represented by the error bars.

P13, L23: We added: "The error bars are one standard deviation of the average for each source type sampled more than once."

**R1:** Page 13, line 23: should read "upper right hand corner"

**Authors:** Changed: "upper left-hand corner" To: "upper right-hand corner"

**R1:** Page 13 line 26: I am not sure I understand what is said here correctly. If there were no PAX measurements at 870 nm for the agricultural pumps and all but one of the garbage burns, how were the AAEs in figure calculated? I can understand that based on the SSA they can be assumed to be almost pure BC and hence an appropriate MAC is chosen to derive a BC mass but unless an AAE measurement derived value is possible they should not be included in figure 1.

**Authors:** Given the low SSA for both ag-pumps, as the reviewer points out, they can be assumed to be almost pure BC, which has an AAE of ∼1 (Bond and Bergstrom, 2006). One garbage burning fire had a measured AAE of one, but we now note that one other garbage burning fire had a high SSA at 405 nm, which is likely incompatible with an AAE of one. We think it is good to include these sources in the figure as a very useful way to illustrate the spectrum of possible AAEs. However, we agree it is absolutely

necessary to highlight the caveats above in the figure caption. We have modified the figure caption to read as follows: "Figure 1. The absorption Ångstr?m exponent (AAE) calculated at 405 and 870 nm as a function of single scattering albedo (SSA) at 405 nm for fuel types measured during the NAMaSTE campaign. The error bars represent $\pm 1$ standard deviation of the AAE measured for different burns (or different samples in the case of brick kilns). Note: "hw" indicates hardwood fuels. The AAE for agricultural pumps was not measured, but was assumed to be one because the SSA at 405 nm is indicative of pure BC. AAE was only measured on one garbage burning fire (value of 0.971) though the SSA at 405 nm on another garbage burning fire indicates that its AAE was larger than one."

**R1:** Page 14 section 3.2: I am somewhat surprised that NO2 is below detection limit. However, the detection limits don't appear to be available anywhere in the paper to know whether the NO2/NOx ratios are simply too low to be measured ($<$2%) or are abnormally low for the vehicle type.

**Authors:** NO2 is one of the more difficult compounds to measure by FTIR due to interference by water lines. Hence the experimental section (P9, L5) mentions the higher uncertainty for $NO_x$ species and we appended "with the detection limits being near the upper end of the reported range." (e.g. $\sim$100 ppb)

**R1:** Page 16 line23-24: it isn't clear why this isn't provided in the earlier section.

**Authors:** The high EF CO for gasoline powered motorbikes was discussed in the earlier section (P14, L35-39). The point of repeating this information here was to point out the recurring tendency for inefficient gas-powered engines to emit high CO. To clarify this we changed the beginning of the sentence to "This is similar to the gasoline-powered motorcycles discussed in Sect. 3.2 that also had high EFs for CO . . . "

**R1:** Page 20 line 8-10: It is stated that the scaling of lab emission factors is carried out according to Stockwell et al (2014) but this uses emission ratios to CO for smouldering and not methane. Furthermore, the full discussion of the method is not really presented

in detailed in Stockwell et al (2014) but is in Yokelson et al (2008). It might be good to cite this paper directly.

**Authors:** We thank the Referee for careful reading and we changed the text to explain that any widely-measured smoldering compound (e.g. CO or $CH_4$) can be used as a reference. We also included more citations to provide more examples.

P20, L8: Change

Old: "As in Stockwell et al. (2014, 2015) we obtained field representative values from the lab data . . . "

New: "A frequently measured smoldering compound (e.g. CO or $CH_4$) can be used as a reference for other smoldering compounds and $CO_2$ is a good reference for other flaming compounds. Similar to previous work (e.g. Yokelson et al., 2008; 2013; Stockwell et al., 2014; 2015) we obtained field representative values from the lab data . . . "

**R1:** Page 26 line 18: define the subscripts CK and ZZ and use elsewhere, otherwise write out what is meant.

**Authors:** The text here now reads: "We simply list some common pollutants/precursors of concern and include the approximate clamp kiln to zig-zag kiln EF ratio in parentheses after each:"

**Additional voluntary changes**

Table S3: "RETS coal" corrected to "RETS charcoal"

P7, L25: We replied to comment by Ellen Baum "We will modify the text to clarify that port sampling approaches also have benefits"

Old: The tall stacks are most easily sampled from a port on the side, but this raises uncertainties due to possible condensation after sampling hot/moist exhaust or losses on stack walls past the sampling point.

New: The tall stacks have been sampled from a port on the side, which is useful but raises uncertainties due to possible condensation after sampling hot/moist exhaust or losses on stack walls past the sampling point.

P8, L25-29: The original text here may have incorrectly implied that Teflon allowed losses of sticky species in the White cell, but the actual problem with the previous cell was that the mirrors were glued to the cell walls and the glue did not stick well to Teflon. Thus, we mounted the mirrors independently and then refreshed the cell coating ourselves with halocarbon wax (to save time and money). We had earlier shown that the halocarbon wax has similar passing efficiency to Teflon (Yokelson et al., 2003). Minor editing clarified this point.

Old text: "Since the last report of the use of this system (Akagi et al., 2013), several upgrades were made: (1) addition of a retroreflector to the White cell mirrors increased the optical pathlength from 11 m to 17.2 m, lowering previous instrument detection limits, (2) replacing the Teflon cell coating with halocarbon wax for better measurements of ammonia ($NH_3$), hydrogen chloride (HCl), hydrogen fluoride (HF), and other species prone to absorption on surfaces,"

New text: "Since the last report of the use of this system (Akagi et al., 2013), several upgrades/changes were made: (1) addition of a retroreflector to the White cell mirrors increased the optical pathlength from 11 m to 17.2 m, lowering previous instrument detection limits, (2) renewing the Teflon cell coating with halocarbon wax to maintain good measurements of ammonia ($NH_3$), hydrogen chloride (HCl), hydrogen fluoride (HF), and other species prone to absorption on surfaces,"

P10, L15: Added the following text: "The 405 nm laser in the PAX has a common nominal wavelength that is usually not measured precisely. After the mission a factory absorption calibration was performed with $NO_2$ gas that was within 1% of the expected result (Nakayama et al., 2015). As part of this calibration, the laser wavelength was precisely measured as 401 nm. This difference from the nominal 405 nm wavelength

adds 1% or less uncertainty to the AAE and absorption attribution (Sect. 2.3). We have continued to refer to the wavelength as 405 nm since this is a standard nominal wavelength for aerosol optical measurements."

P15, L9: added citation (and reference) to Moussa et al., (2016) on vehicle emissions of HCN.

P21, L37: changed ∼5.5 to 4.5 g/kg.

P26, L 34: change from "slightly greater" to "much greater"

P28, Acknowledgements: We added information, new text reads: "C. E. S., T. J. C., R. J. Y., purchase of the PAXs, WAS analyses, and many other NAMaSTE-associated expenses were supported by NSF grant AGS-1349976. R. J. Y., D. R. B., and I. J. S. were also supported by NASA Earth Science Division Award NNX14AP45G. T. J. and E. A. S. were supported by NSF grant AGS-1351616. J. D. G. and P. F. D. were supported by NSF grant AGS-1461458. P. V. B., P. S. P., S. A., R. M., and A. K. P. were partially supported by core funds of ICIMOD contributed by the governments of Afghanistan, Australia, Austria, Bangladesh, Bhutan, China, India, Myanmar, Nepal, Norway, Pakistan, Switzerland, and the United Kingdom. E. S. was supported by NSF grant AGS-1350021. We thank G. McMeeking, J. Walker, and S. Murphy for helpful discussions on the PAX instruments and data; S. B. Dangol, S. Dhungel, S. Ghimire, and M. Rai for identifying and arranging access to the field sampling sites; B.R. Khanal for assisting with the lab-based cooking tests; and Nawraj and K. Sherpa for logistic support."

P30, L22-25: Flipped Bond et al 2006 and Bond et al 2004

P32, L18-21: Flipped Franco et al and Fullerton et al to comply with alphabetical order

P34: Updated the Jayarathne et al. 2016 in prep reference (to add authors)

Table 5: fixed some subscripts.

---

## Author Comment (AC3) · 22 Aug 2016

We thank Referee #2 for their positive assessment of our work.

---

## Author Comment (AC4) · 22 Aug 2016

We thank Referee #3 for their overall positive assessment and suggestions, which will improve the paper. Next we reproduce the exact comments followed by a detailed response:

**Anonymous Referee #3**

[Figure]

Summary: This paper describes the measurements of trace gases and carbonaceous (BC, and BrC) aerosols (mass and optical properties) from a variety of bio and fossil fuel burning sources in an around Kathmandu valley of Nepal in the eastern Himalayas. These measurements formed part of a field campaign using a mobile laboratory and a set of state-of-the-art instruments. These measurements provide important and valuable data on light absorbing aerosols and trace gases over this part of the Himalayas, and is done so extensively perhaps for the first time and thus are, useful for environmental and climate impact assessment studies. Having said the above, the data should also be taken with the pinch of salt that the measurements have been for a very short duration (about a fortnight), the number of samples used to evolve the means are statistically not very high. How well these sample represent the regional population of these sources remains to be seen. In other words, the data emanates from a highly under-sampled source. Of course, the authors also seem to be aware of this. Moreover, the reported values correspond only to a particular period of time of the year (not even a full month). They also do not represent the strong seasonal variations in some of these sources and also changes caused by changes in synoptic meteorology and long-range transport; which are very important over this region. Their spatial representativeness also is highly limited to the urban region of Nepal and its southern slopes, which would be impact by transport of emission from the adjoining north Indian plains. Nonetheless, it remains that these are the first comprehensive set of measurements from several sources that are specific to this region (on which field measurements are not possible in other parts of the world), and provide information that could be used as inputs for regional climate and environment impact assessment and probably would lead to more such measurements. In other words, they provide some ball-park figure on these parameters, which were otherwise not available. In view of the above, the paper may be accepted for publication in ACP, provided the authors take care of the following specific concerns in their revised version:

[Figure]

**R1.** Make a clear statement of the limitation of the data (under sampling in time, space and sources) explicitly in the abstract and conclusions

**Authors:** Two Referees suggested clarifying the need for more sampling of these sources and we have added that to the abstract, introduction, and conclusions as described in the response to Referee #1.

**R2.** Provide the fractional figures (to the extent accurate) of the sources sampled, against the total number of such sources present in Kathmandu (for e.g. % of brick kilns sampled vs the total no. available in Kathmandu, the no. of two-wheelers sampled vs the total two-wheelers plying in the city, the no. of gensets samples vs the total etc in a table or in the respective sub-sections.

**Authors:** We hesitate to try to add an estimate of the fraction of each source sampled. We certainly understand the gist of the comment in that not nearly enough sampling has been done! Our work just begins a daunting task. We already made this point in several ways: 1) we clearly stated that we began to address the data shortage (and added emphasis as above), 2) we said fleet-average values for traffic requires a bigger study, 3) we estimated that there are 1000 kilns in Nepal and noted that we measured only two, 4) we gave the total fuel use as opposed to number of devices in most cases, but that makes it clear our sampling is just a beginning. According to statistical theory, it's the size of a random sample and not the fraction of the population sampled that determines representativeness and uncertainty. The more common target types are more likely to be selected. Our sampling was not purely random, but was targeted towards common local practice by in-country experts. The variability of these sources is not well-known and there are seasonal trends we could not measure, but our measurements were in the dry season, which is when pollution problems are the highest. Note transport and meteorology issues are not relevant to source measurements. As is typical for emerging issues, we use what data we have in the beginning and the available data, as a whole, are generally improved as more measurements are made. We do however agree with both Referees #1 and #3 that a reminder that more sampling

is needed (especially for gensets, ag-pumps, brick kilns, and transportation) would be useful and we have added that to the conclusions and abstract as described above.

**R3.** For FTIR – please provide the actual averaging time used (for grab sampling), the best and worst S/N for the species being reported in this study and the resulting uncertainty in the derived concentrations.

**Authors:** We are not completely sure what the comment refers to, but there are several important issues raised and we try to address all of them. For both FTIR and WAS it takes about 10 s to acquire a grab sample so they are spot measurements rather than time-averaged. If the question is about the storage time in the FTIR cell for signal averaging that is 2-3 minutes and on P8, L21 we changed "several minutes" to "two to three minutes" to be more specific.

The FTIR S/N and the uncertainty, which is impacted by multiple factors, is different for every mixing ratio we retrieve. The uncertainty in mixing ratios ranges from very high near the detection limit to very low in most cases because of the extremely high concentrations in source plumes. The ER and EF for each source are based on the regression using all the samples of the source in an ER plot as described in the text. For less concentrated samples the uncertainty is higher but their weight is reduced in the regression with the intercept forced as explained in text. For the extremely high concentrations that dominate the ER plots, the uncertainty is mainly the uncertainty in the reference spectra given in Sect. 2.2.1. The natural variability in the ERs or EFs from source to source of the same nominal type (∼40%, seen in the tables) tends to much larger than the uncertainties in the slopes for each source (∼10%). Thus, as a reasonable estimate, the uncertainty for the mean EF for each source type is given as one standard deviation of the mean. We've clarified the relatively minor role of the uncertainty in the individual mixing ratios with the following change in the text:

P10, L31-32: Now reads: "Forcing the intercept effectively weights the points obtained at higher concentrations that reflect more emissions and have greater signal to noise

so that error is dominated by calibration uncertainty."

**R4.** In the ER calculation, state explicitly how the average ER estimated from the FTIR around a given source, compared with the values obtained from WAS data, for completeness of information and also in view that WAS value represents the 'bulk'.

**Authors:** The samples by WAS and FTIR were at different times so the differences that may occur between them could be due both to uncertainty and natural variability. The number of FTIR samples is much larger as the cell can be refilled after the spectra are collected for two-three minutes, but the WAS samples add unique species and add sampling of the plume at more times overall. Despite the lack of exact overlap in timing, the single WAS samples for each source are in, or close to, the range of FTIR values. We don't understand the comment about WAS representing the "bulk," but think both approaches give a good overview of the emissions.

**R5.** It is highly appreciable the way authors have described all the assumptions used in their estimates. Yet, it would be better if they can give an upper and lower bound for the estimates due to the specific assumption they have made (like 50% wood and dunk in mixed fuels – suppose it is 80-20 or 30-70 )

**Authors:** EFs are proportional to assumed %C and we discussed how to adjust the EF to reflect different %C that would arise from different mixes in the response to Referee #1. We had no way of measuring the real mix of fuels that was burning at the instant of our grab samples, nor do we know the range of mixtures commonly used so we took an illustrative guess. We have added the adjustment procedure in the mixed fuel EF section.

P12, L39: Add: "For mixtures differing from those we used, the EF scale with the assumed carbon fraction."

We also added the assumed carbon fraction of the pure fuels to the text so the impact of arbitrary mixtures can be estimated directly from the information in the text.

P12, L31-32: "Thus for the mixed-fuel cooking fires, we simply assumed an equal amount of wood (0.45 C) and dung (0.35 C) burned and used the average carbon fraction for the two fuels (0.40)"